# Herbicide leakage into seawater impacts primary productivity and zooplankton globally

Liqiang Yang [1,2], Xiaotong He [1], Shaoguo Ru [1] ✉ & Yongyu Zhang [2,3,4] ✉

Predicting the magnitude of herbicide impacts on marine primary productivity remains challenging because the extent of worldwide herbicide pollution in coastal waters and the concentration-response relationships of phytoplankton communities to multiple herbicides are unclear. By analyzing the spatio-temporal distribution of herbicides at 661 bay and gulf stations worldwide from 1990 to 2022, we determined median, third quartile and maximum concentrations of 12 triazine herbicides of 0.18 nmol L$^{-1}$, 1.27 nmol L$^{-1}$ and 29.50 nmol L$^{-1}$ (95%Confidence Interval: CI 1.06, 1.47), respectively. Under current herbicide stress, phytoplankton primary productivity was inhibited by more than 5% at 25% of the sites and by more than 10% at 10% of the sites (95% CI 3.67, 4.34), due to the inhibition of highly abundant sensitive species, community structure/particle size succession (from Bacillariophyta to Dino-phyceae and from nano-phytoplankton to micro-phytoplankton), and result-ing growth rate reduction. Concurrently, due to food chain cascade effects, the dominant micro-zooplankton population shifted from larger copepod larvae to smaller unicellular ciliates, which might prolong the transmission process in marine food chain and reduce the primary productivity transmission effi-ciency. As herbicide application rates on farmlands worldwide are correlated with residues in their adjacent seas, a continued future increase in herbicide input may seriously affect the stability of coastal waters.

Millions of tonnes of herbicides are used annually worldwide[1,2], 70% of which eventually enter the ocean through runoff[3], and various herbi-cides are frequently detected at concentrations that pose risks in seawaters throughout the world[4,5]. More than half of these herbicides are photosynthetic inhibitors that can significantly inhibit the photo-synthesis of phytoplankton at very low concentrations (ppb)[6]. The inhibition of photosynthesis in coral symbiotic algae (zooxanthellae) by herbicides is thought to be an important factor contributing to the bleaching of coral reefs in Australia's Great Barrier Refs. 7,8. Our pre-vious study found that some dominant offshore species (e.g., *Phaeo-dactylum tricornutum* and *Chaetoceros* sp.) were extremely sensitive to

triazine herbicides, and atrazine at ambient concentrations (1 μg L$^{-1}$) significantly inhibited their photosynthesis (approximately 57% of photosynthetic genes were significantly downregulated) and wea-kened their chlorophyll *a* fluorescence intensity[9,10]. Marine phyto-plankton contribute approximately 50% of the total global primary productivity and play a vital role in global carbon cycling[11]. Whether the inhibition of sensitive phytoplankton communities by herbicides will seriously affect offshore primary productivity and thus lead to offshore desertification is an important potential problem.

Many studies on the toxicity of herbicides to microalgae have been carried out, and the results showed that herbicides have

[1]College of Marine Life Sciences, Ocean University of China, Qingdao, China. [2]Qingdao Institute of Bioenergy and Bioprocess Technology, Chinese Academy of Sciences, No. 189 Songling Road, Qingdao, Shandong 266101, China. [3]Shandong Energy Institute, No. 189 Songling Road, Qingdao, Shandong 266101, China. [4]Qingdao New Energy Shandong Laboratory, Qingdao, Shandong 266101, China. ✉e-mail: rusg@ouc.edu.cn; zhangyy@qibebt.ac.cn

significant effects on the photosynthetic physiology, nutrient uptake rate, and the expression of key carbon sequestration enzyme-encoding genes of planktonic algae[12–14]. These findings are very helpful for understanding the sensitivity differences of different phytoplankton species to herbicides. However, it is difficult to truly understand the ecological effects of herbicides in situ from the individual or population response level of algae alone, and the impact of herbicides on the structure of the phytoplankton community cannot be based on changes in photosynthetic physiological indicators alone. Marine primary productivity is the aggregate of various phytoplankton groups, and the community structure and particle size of phytoplankton are two key indicators of primary productivity[15,16]. Different phytoplankton groups have different preferences for substrates (such as $NO_3^-$ and $NH_4^+$) and different growth rates, leading to different production levels and ecological significance[17,18]. In addition, the productivity and energy flow pathways formed by phytoplankton with different particle sizes are also different. The smaller pico-phytoplankton mainly enter the complex microbial loop through ingestion by heterotrophic protists[19], while the larger nano- and micro-phytoplankton mainly enter the classic food chain (phytoplankton->heterotrophic protists->zooplankton), with higher energy conversion efficiency[19]. Therefore, the inhibitory effects of herbicides on high-abundance and sensitive groups (e.g., *P. tricornutum* and *Chaetoceros* sp.) may not only lead to a decrease in overall primary productivity but may increase the proportion of phytoplankton groups that are resistant to herbicides, which may trigger succession of the community structure and changes in the algal cell particle size composition, thus changing the growth rate and energy conversion efficiency. Therefore, exploring the effects of the suppression of sensitive algae on the abundance, community composition, and particle size of phytoplankton is expected to explain the effects of modern intensive agriculture on marine primary production.

However, it is not enough to simply establish the concentration-response relationship between herbicides and phytoplankton primary productivity to assess the extent of herbicide effects on primary productivity in coastal waters, which is closely related to the residual status of herbicides in the coastal waters. Rather, determining the scope and degree of offshore herbicide pollution is critical. However, most of the current research on herbicide toxicity is based on risk assessments of a single herbicide in a small region[20,21], and a comprehensive understanding of the current status of herbicide pollution in global coastal waters is lacking, which severely limits the assessment of herbicide effects on primary productivity. To quantify the impact of current herbicide pollution on offshore primary productivity on a larger scale, we first collected survey data published from 1995 to 2022 on herbicide pollution in bay and gulf around the world. The temporal and spatial distribution patterns and background values of herbicides in the coastal waters of typical bay and gulf areas on all continents were determined. This approach aimed to help answer key scientific questions such as the scope of herbicide pollution in offshore waters and how large of an area of phytoplankton primary productivity is affected.

In fact, even after understanding the distribution of various herbicides in coastal waters, exploring the effects of these herbicides on coastal primary productivity at current environmental concentrations is still challenging. In contrast to the single herbicide types typically found in farmland soils, the ocean is the final sink for nearly all herbicides, and the composition of herbicides in seawater is extremely complex[10,21–23]. There is an additive toxicity effect among herbicides with the same mode of action[24]. Thus, assessing the ecotoxicity of individual herbicides does not reflect in situ conditions. However, few studies have evaluated the cumulative toxicity of herbicides to phytoplankton. In addition, there are differences in the types, concentrations, and proportions of herbicides in different sea areas and even at different locations within the same area. It is unrealistic to simulate the distribution of herbicides at various sites and to test their ecological

effects through experiments. To truly reflect the stress effects on phytoplankton under the current herbicide pollution levels, we first established the concentration-effect relationship curve and a toxicity equivalence database for each herbicide to a representative population of phytoplankton. Then, these were combined with the concentration addition model, in which the total concentration of various herbicides in a specific sea area/station that was obtained from a marine survey was expressed by the toxic equivalent quantity (TEQ) of a typical herbicide to normalize the various herbicide homologs remaining in the water body. On this basis, a comprehensive analysis of the responses of key indicators of phytoplankton primary productivity to herbicide stress within an equivalent TEQ range could be used to quantify the impact of compound pollution from multiple herbicides on the primary productivity of large sea areas.

Agricultural production is the main source of herbicides in coastal waters, and the types and dosages of herbicides used on farmlands largely determine the herbicide pollution status in the adjacent seas[22,25]. At present, there are more than 1500 herbicides on the market worldwide, with more than 300 active ingredients[26]. There are differences in the types of herbicides applied in different regions and on different crops. If a geographic, quantitative database of global herbicide usage and residues can be established, it will be possible to predict the types and even concentrations of herbicides in coastal sea areas based on the nearby herbicide usage, which may guide marine monitoring of herbicides. Therefore, we first tried to retrieve the average usage data of 52 herbicides that are currently being widely used worldwide from the PEST-CHEMGRIDSv1 global database[27] established by Maggi et al., and reanalyzed them in combination with the Food and Agriculture Organization (FAO) Enterprise Statistics Database[28] and the National Integrated Pesticide Project Database[29] of the United States Geological Survey to draw the geographic distribution map and ecological risk level map of global herbicide usage. Combined with marine surveys, the indication effects of the types and amounts of herbicides used on farmlands on the pollution status of herbicides in adjacent sea areas were tested, and the correlation was verified between the inhibition degree of primary productivity in each bay and gulf area and the ecological risk level of the herbicide residues in its surrounding farmlands. This work will provide suggestions for reducing emissions, preventing herbicide overuse on the premise of ensuring food security, and ensuring coastal ecological health.

This study aims to reveal the current global status of marine herbicide pollution and evaluate its impacts on marine primary productivity and secondary effects on higher trophic levels (Fig. S1). By analyzing the spatiotemporal distribution of herbicides at 661 bay and gulf stations worldwide from 1990 to 2022, an overall picture of the current status of herbicide pollution in global coastal waters was obtained; by establishing the toxicity equivalent database of each herbicide and the dose-response relationship between the concentration of atrazine and chlorophyll a in seawater at the phytoplankton community level, the overall inhibition effect of 12 triazine herbicides on phytoplankton primary productivity was quantified; by analyzing the effects of herbicides on phytoplankton community structure, particle size composition, production cycle, and energy transfer process (Fig. S2), the potential mechanism of herbicides inhibiting phytoplankton primary productivity was elucidated; Moreover, the effect of herbicides on higher trophic levels and the possibility of predicting marine herbicide pollution through indicators of herbicide use on land were explored.

## Results

### Temporal and spatial distribution of herbicides in global bay and gulf waters

Among 32 herbicides in 5 categories, triazine herbicides (12) accounted for 37.5% of the total types, and their detection frequency reached 95% among the 661 stations within the 7 sea areas of 5 continents

(Fig. 1). The Gulf of Mexico, East Asia and Vilaine Bay were areas of high concentrations of triazine herbicides, with median concentrations of 3.84 nmol $L^{-1}$ (95%Confidence Interval: CI 1.82, 9.93), 2.28 (95%CI 1.89, 2.61) nmol $L^{-1}$ and 1.64 (95%CI 0.52, 2.76) nmol $L^{-1}$ and the highest detection concentrations of up to 13.67 nmol $L^{-1}$, 12.07 nmol $L^{-1}$ and 11.68 nmol $L^{-1}$, respectively (Fig. 1, Table S1). The total concentration of triazine herbicides in water bodies such as East Asia (one-way ANOVA, $p < 0.001$), the US East Coast (one-way ANOVA, $p < 0.001$), and South Africa (one-way ANOVA, $p = 0.042$) showed a gradual upward trend over time (Fig. 1).

## Prediction of the effects of herbicide pollution on phytoplankton primary productivity

Here and elsewhere, uncertainties were reported as mean ± standard deviation (SD). The toxicity of 12 triazine herbicides to *P. tricornutum* Pt-1 were significantly different (EC50 values were between 4.3 ± 0.3 and 849.1 ± 21.7 nmol $L^{-1}$) (Fig. S3). Cybertron and terbutryn showed high toxicity values, 30 and 16 times higher than that of atrazine, respectively (Table S2). Therefore, the overall toxicity of these 12 herbicides cannot be expressed simply by adding up the concentrations of each herbicide. The concentration-response curves of the 12 triazine herbicides all conformed to the logistic or Weibull equation (Fig. S3, Table S2), and the fitting coefficients ($R^2 = 0.989–0.999$) indicated that the toxicity of each herbicide to phytoplankton had a good dose correlation, indicating that the precondition for equivalent conversion was met. The concentrations of atrazine that had an equivalent toxic effect to that of the in situ concentrations (equi-effective concentrations) of all 12 detected triazine residues were

calculated to be 0–47.58 nmol $L^{-1}$ (Fig. S3, Table S2), and the corresponding median, third quartile, and maximum concentration values were 0.54 nmol $L^{-1}$, 5.09 nmol $L^{-1}$ and 47.58 nmol $L^{-1}$ (95%CI 3.33, 4.33), respectively (Fig. 2). Camps Bay and Maputo Bay in South Africa, the Yellow Sea and the Bohai Sea in Asia, and the Gulf of Mexico in North America were high-value areas, with median equi-effective concentrations of 9.47 nmol $L^{-1}$ (95%CI 3.33, 14.33), 8.74 (95%CI 7.90, 9.58) nmol $L^{-1}$, and 7.31 nmol $L^{-1}$ (95%CI 2.32, 11.28), respectively. The highest equivalent concentrations in Maputo Bay, the Yellow River estuary, and the coastal waters of the Gulf of Mexico even reached 47.58 nmol $L^{-1}$, 23.16 nmol $L^{-1}$, and 22.91 nmol $L^{-1}$, respectively (Fig. 2, Table S1). With reference to the equi-effective concentrations, three concentrations that were similar to the in situ concentrations of triazine herbicides, that is, a low dose (0.5 nmol $L^{-1}$), an intermediate dose (5 nmol $L^{-1}$), and a high dose (50 nmol $L^{-1}$), were established to study the impacts of triazine herbicides on phytoplankton primary productivity.

High-throughput sequencing showed that *Bacillariophyta* (*Chaetoceros tenuissimus*, nano-phytoplankton) and *Dinophyceae* (*Gyrodinium jinhaense*, *Adenoides eludens*, *Ankistrodinium semilunatum* and *Euduboscquella* sp., micro-phytoplankton) were the dominant groups of phytoplankton in the natural seawater, with relative proportions (OTU count data) of 77.4 ± 5.87% and 17.8 ± 2.16%, respectively (Fig. 3, Table S4). Atrazine exposure had significant impacts on the phytoplankton community, which was indicated by both the significant variation in alpha diversity ($p = 0.041$) (Table S3) and the principal coordinate analysis (PCoA) profile ($p = 0.01$; $R = 0.81$) (Fig. S4a, Fig. 3). Under the stress of low, medium and high doses of atrazine, the

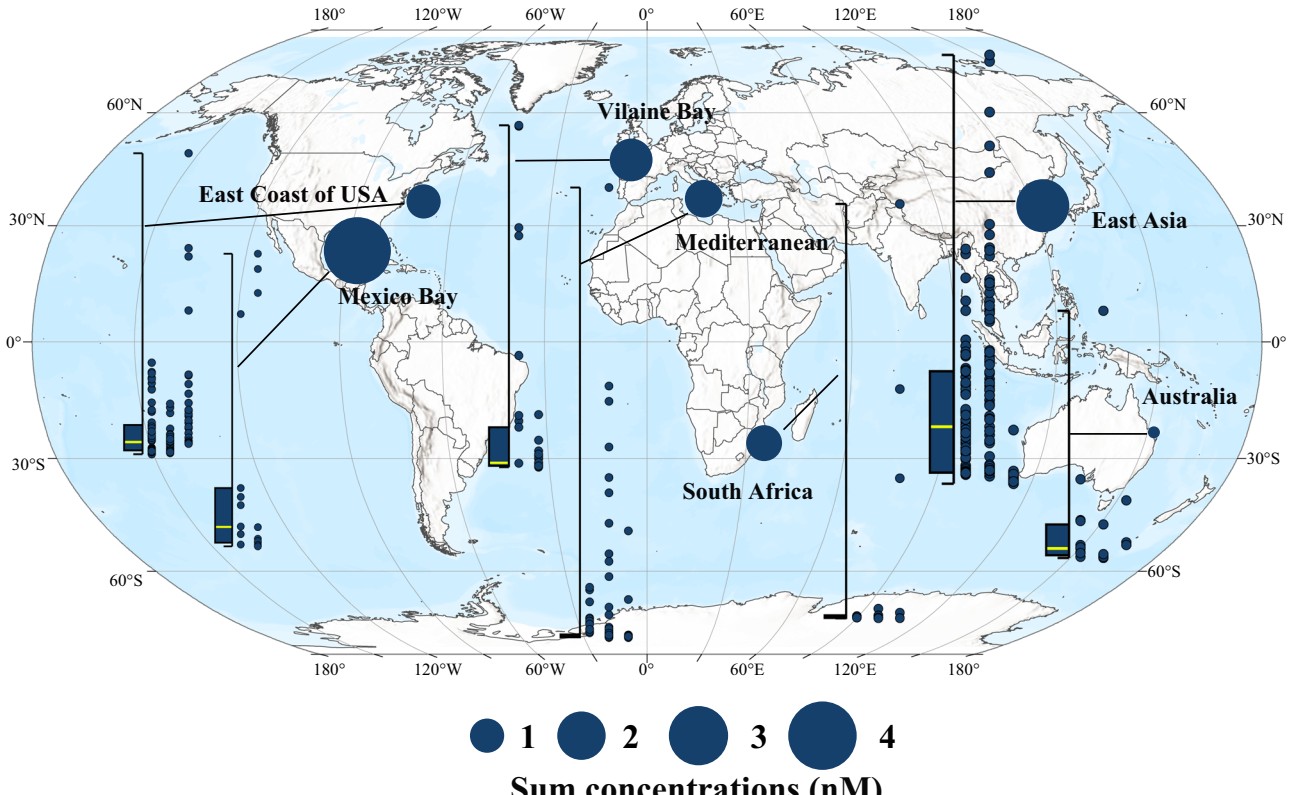

**Fig. 1 | Geographical distribution of the total concentration (nM) of 12 triazine herbicides in typical bay and gulf worldwide.** The box-scatter plots show the residual concentrations of herbicides. Each point in the scatter plots corresponds to a survey site, arranged from left to right in chronological order (1990–2000, 2001–2011, and 2012–2022). Significant differences among chronological stages were calculated using one-way analysis of variance (ANOVA). Box plot whiskers of the box chart on the left represent the minimum and maximum values, with bounds of the box representing the first quartile, median (marked by yellow line), and third quartile of 12 triazine herbicides at all sites in the corresponding sea area ($n = 16$, 128, 34, 271, 21, 168, and 22 independent survey sites, from left to right). The solid circle in each sea area represents the median value of the total concentration of the 12 triazine herbicides at each station in that area, with a larger circle size representing a larger median value. Source data are provided as a Source Data file.

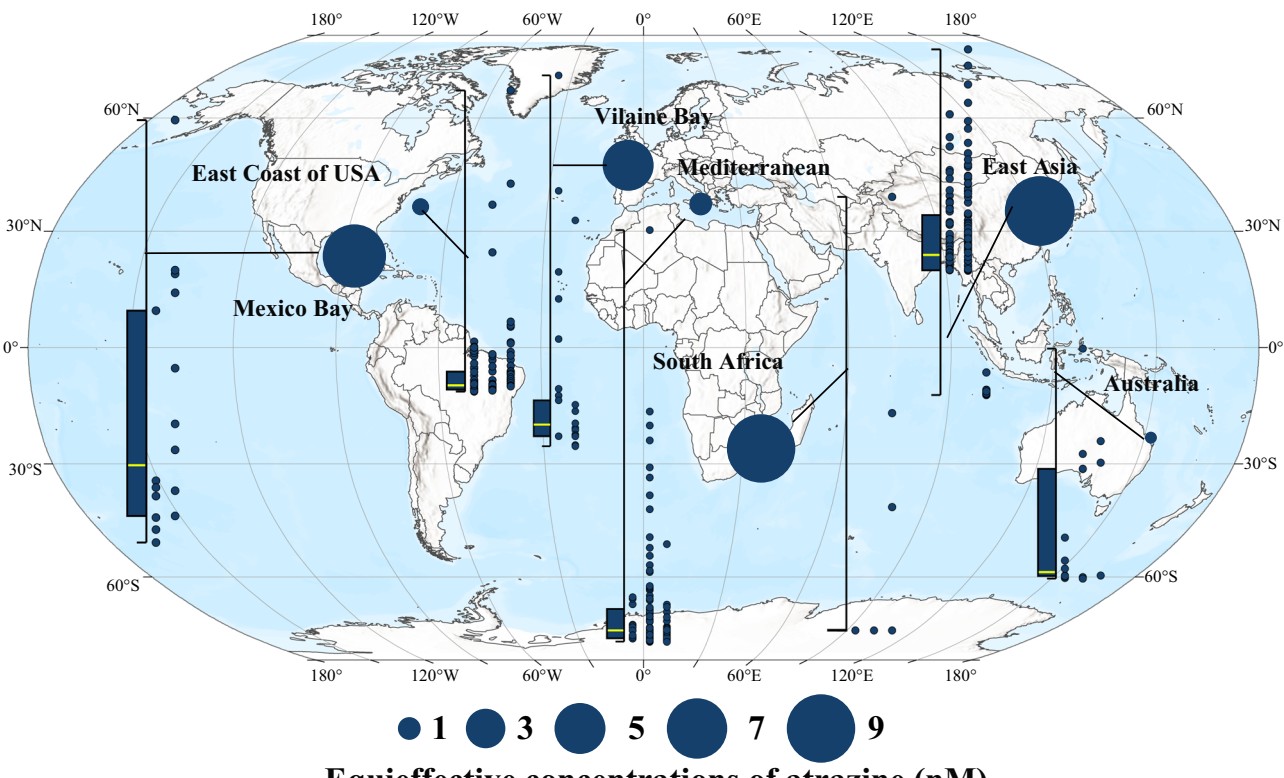

**Equieffective concentrations of atrazine (nM)**

**Fig. 2 | Geographical distribution of equi-effective concentrations of atrazine in typical bay and gulf worldwide.** The box-scatter plots show the equi-effective concentrations of atrazine. Each point in the scatter plot on the right corresponds to a survey site, arranged from left to right in chronological order. Box plot whiskers of the box chart on the left represent the minimum and maximum values, with bounds of the box representing The box chart on the left shows the first quartile, median (marked by yellow line) and third quartile of the equi-effective concentrations of atrazine at all sites in the corresponding sea area ($n$ = 16, 128, 34, 271, 21, 168, and 22 independent survey sites, from left to right). The solid circle in each sea area represents the median value of the equi-effective concentrations of atrazine at each station in that area, with a larger circle size representing a larger median value. Source data are provided as a Source Data file.

proportion of *Bacillariophyta* decreased from 77.4 ± 5.87% to 15.5 ± 5.21%, 10.7 ± 3.35% and 0.7 ± 0.23%, respectively. At the same time, *Dinophyceae* increased from 17.8 ± 2.16% to 47.5 ± 6.87%, 53.6 ± 10.06% and 79.2 ± 11.17%, respectively, and the phytoplankton community changed from *Bacillariophyta*-dominated to *Dinophyceae*-dominated (Table S3). At genus or species level (Table S4), the genus *Chaetoceros*, which was the most common representative of the Bacillariophyta, suffered the most significant inhibitory effects. Under the stress of low, medium, and high doses of atrazine, the proportion of *Chaetoceros* decreased from 73.0% to 5.2% ($p$ = 0.0007), 2.9% ($p$ = 0.00029) and 0.3% ($p$ = 0.00013), respectively. Moreover, most members of the Dinophyceae (*Gyrodinium jinhaense, Adenoides eludens, Ankistrodinium semilunatum, and Euduboscquella* sp.,) were more resistant to atrazine stress. The relative abundances of *Gyrodinium jinhaense* significantly increased from 9.8% to 17.2% ($p$ = 0.00075), 14.0% ($p$ = 0.00046), and 60.5% ($p$ = 0.00028), respectively, under exposure to low, medium and high doses of atrazine.

In terms of particle size, based on high-throughput sequencing data, the control group was dominated by nano-phytoplankton (*Chaetoceros tenuissimus* and *Goniomonas avonlea*, ≥73.1%), while the low-dose atrazine treatment group was dominated by micro-phytoplankton (*Gyrodinium jinhaense, Adenoides eludens, Euduboscquella* sp. JMC-2019a, and *Ankistrodinium semilunatum*, 31.9%) and pico-phytoplankton (*Chrysochromulina rotalis* and *Chrysochromulina leadbeateri*, 27.3%) (Fig. 3, Table S4). The dominance of micro-phytoplankton (*Gyrodinium jinhaense, Euduboscquella* sp. JMC-2019a, *Woloszynskia halophila* and *Fibrocapsa japonica*, 69.4%) was more obvious in the treatment group that received a high dose of atrazine.

The phytoplankton community changed from nano-phytoplankton-dominated to micro-phytoplankton-dominated. Even within the class *Bacillariophyta*, the dominant taxa showed a transition from small-sized *Chaetoceros* (from 73% to 1%) to larger-sized *Thalassiosira* (from 2.7% to 8.5%) sp. (Table S4). Atrazine exposure led to a decrease in the total chlorophyll $a$ concentration (indicating primary productivity). In comparison to that in the control groups, the chlorophyll $a$ concentration of phytoplankton decreased by 9.1% ($p$ = 0.048), 8.8% ($p$ = 0.047), and 18.8% ($p$ = 0.004) after 2 days of exposure to 0.5, 5, and 50 nmol L$^{-1}$ of atrazine, respectively, and decreased by 16.9% ($p$ = 0.00053) and 24.5% ($p$ = 0.000071) after 30 days of exposure to 5 and 50 nmol L$^{-1}$ of atrazine, respectively (Fig. 4, Table S4). By comparing the changes in the Chl $a$ concentration of each particle size (<2 μm, Pico-; 2–20 μm, Nano-; 20–200 μm, Micro-) in the control group and the experimental group, the potential effects of environmental concentrations of triazine herbicides on the particle size structure of phytoplankton were characterized. We revealed that the contribution of nano-phytoplankton decreased significantly after two days of atrazine exposure. On the 30th day, the contribution of nano-phytoplankton to the primary productivity decreased from 75.3 ± 3.52% in the control group to 44.9 ± 4.79%, 23.7 ± 2.87% and 16.7 ± 3.91% in the treatment groups dosed with 0.5, 5 and 50 nmol L$^{-1}$ of atrazine, while the proportion of micro-phytoplankton increased from 21.7 ± 2.17% to 52.7 ± 6.32%, 66.7 ± 9.57% and 74.1 ± 11.21%, respectively. This indicated that nano-phytoplankton were more sensitive than micro-phytoplankton to atrazine (Fig. 4, Table S4), which was consistent with the results of the high-throughput sequencing. It is worth noting that although the total concentration of Chl a under a low

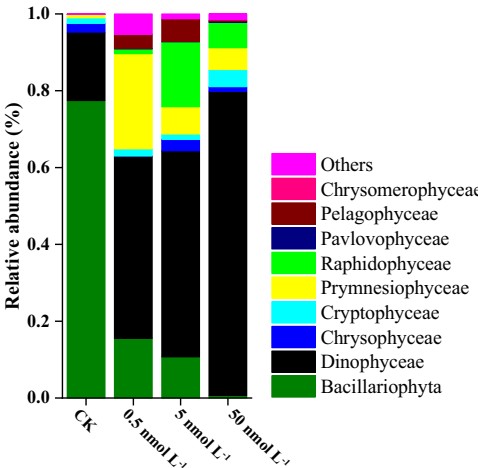

**Fig. 3 | Comparison of phytoplankton community structure between the control and atrazine-treated groups at the phylum/class level.** CK, 0.5, 5, and 50 nmol L$^{-1}$ represent the control and the treatment groups dosed with 0.5–50 nmol L$^{-1}$ of atrazine, respectively, on the 21st day (with three replicates). A total of 566,933,791 high-quality sequences (average length of 464) from 12 samples were obtained and clustered into 99 OTUs (97% cutoff). Phyla or classes that did not represent at least 1% of the sequences in at least one sample were regrouped as "Others." $n = 3$ samples per group. Uncertainties were reported as mean ± standard deviation (SD). Statistical significance was determined at $p < 0.05$ and comparisons used Kruskal–Wallis tests with FDR adjusted for multiple comparisons using the Benjamini and Hochberg method. Exact $p$ values and $H$ values can be found in the Source Data file. Source data are provided as a Source Data file.

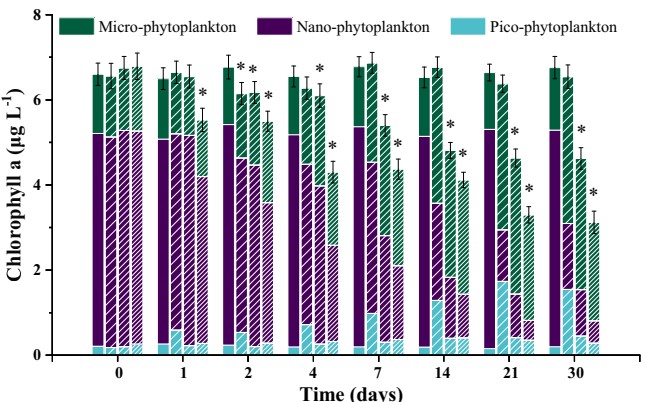

**Fig. 4 | Size-fractionated chlorophyll $a$ concentrations of the phytoplankton communities in the control and atrazine-treated groups.** Within each time period the four bars depict respectively the values for control, 0.5, 5, and 50 nmol/L of atrazine. $n = 3$ samples per group. Uncertainties were reported as mean ± standard deviation (SD). $P$ values were calculated using a two-sided $t$-test. The exact $p$ values were provided in the Source data. The asterisk represents a significant difference ($p < 0.05$) in the chlorophyll $a$ concentration of phytoplankton between the treatment group and the control group. Source data are provided as a Source Data file.

atrazine dose basically recovered to the control level on the fourth day, the inhibition of nano-phytoplankton was not relieved even by the end of the experiment, indicating that atrazine had a more persistent effect on the size fraction of phytoplankton. The intrinsic growth rates ($\mu$) of micro-, nano- and pico-phytoplankton in the control group were 0.65 d$^{-1}$, 0.74 d$^{-1}$ and 1.14 d$^{-1}$ (Table S5), respectively, which indicated that the larger the size fraction was, the slower the growth rate. Atrazine significantly reduced the intrinsic growth rate of phytoplankton (Table S5). Especially in the groups treated with intermediate and high doses of atrazine, the intrinsic growth rates of micro, nano- and pico-phytoplankton decreased by 18.5% ($p = 0.014$) and 32.3% ($p = 0.0072$), 36.5% ($p = 0.00081$) and 52.7% ($p = 0.00047$), and 14.9% ($p = 0.029$) and 71.9% ($p = 0.0028$), respectively, which meant that atrazine slowed the growth rate of phytoplankton. Compared with micro- and pico-phytoplankton, the intrinsic growth rate of nano-phytoplankton was more inhibited under low, medium, and high atrazine stress (Table S5). In addition, since the net growth rate (NGR) of nano- and pico-phytoplankton was negative under atrazine stress, while the micro-phytoplankton still showed a positive growth trend, the proportion of micro-phytoplankton increased. Therefore, the increase in micro-phytoplankton caused by atrazine exposure might slow the growth rate of phytoplankton primary productivity.

In addition to slowing the growth rate, atrazine interfered with the energy transfer process of primary productivity (Table S5). Although the grazing rate (g) of zooplankton on phytoplankton did not change under low-dose atrazine treatment, it decreased significantly ($P \leq 0.01$) with increasing atrazine dosage. On the fourth day, the grazing rates of zooplankton on micro, nano- and pico-phytoplankton under intermediate and high doses of atrazine decreased by 18.1% ($p = 0.0026$) and 37.7% ($p = 0.00041$), 16.9% ($p = 0.0019$) and 11.9% ($p = 0.00027$), and 2.9% ($p = 0.029$) and 24.5% ($p = 0.0044$) (Table S5), respectively, indicating that atrazine impeded the energy transfer process of phytoplankton primary productivity and might have triggered a series of secondary effects by altering the grazing activities of zooplankton.

## Secondary effects of environmental concentrations of triazine herbicides due to structural changes in phytoplankton community

A total of 45–78% of the fixed carbon of phytoplankton enters the food chain through ingestion by zooplankton, especially micro-zooplankton[30]. Therefore, the impact of triazine herbicides on phytoplankton must have secondary effects on micro-zooplankton.

The results of high-throughput sequencing showed that metazoans such as copepod larvae were the dominant group (97%) of micro-zooplankton in the control group, while the proportion of heterotrophic protists was only 3%. The alpha ($p = 0.035$) and beta diversity ($p = 0.01$; $R = 0.54$) (Table S3, Fig. S4b) of the micro-zooplankton community were both significantly changed under low and intermediate doses of atrazine exposure. The relative abundance of heterotrophic protists (Ciliophora) significantly ($p = 0.011$) increased to 10% under exposure to 5 nmol L$^{-1}$ atrazine, although metazoan larvae were still predominant (Fig. 5, Table S4). With a further increase in atrazine concentration (50 nmol L$^{-1}$), the dominant phyla of microplankton changed drastically, and the proportion of heterotrophic protists (*Holosticha diademata*) significantly ($p = 0.00058$) increased to 40.0% (Table S4, S6), becoming the most dominant group. In contrast, the relative abundance of arthropod larvae decreased from 74.4% in the intermediate-concentration group to 21.1% in the high-concentration group. Concomitantly, the proportion of Platyhelminthes (*Paraplehnia seisuiae*) significantly increased, from <1% in the intermediate-concentration group to 19.9% in the high-concentration group (Fig. 5, Table S4, S6).

The morphological identification results were consistent with the trend of the high-throughput sequencing (Table S6): The dominant groups of micro-zooplankton in the control group were mainly copepod larvae (Table S6). Under atrazine exposure, the abundance of copepod larvae showed a downward trend. Especially in the treatment groups dosed with medium and high concentrations of atrazine, the abundance of copepod larvae decreased by an order of magnitude; the abundance of ciliates such as *Tintinnopsis* and *Euplotes* showed an upward trend and became the new dominant group, indicating that atrazine exposure had a greater impact on the population of copepod larvae. The micro-zooplankton community succeeded from large copepod larvae-dominated to small ciliate-dominated under atrazine stress. In addition, there were also community changes in ciliates,

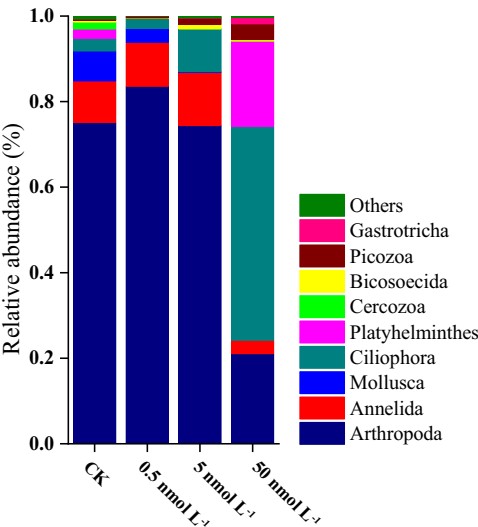

**Fig. 5 | Comparison of the micro-zooplankton community structure between the control and atrazine-treated groups at the phylum/class level.** CK, 0.5, 5, and 50 nmol L$^{-1}$ represent the control and treatment groups dosed with 0.5–50 nmol L$^{-1}$ of atrazine, respectively, on the 21st day (with three replicates). A total of 566,933,791 high-quality sequences (average length of 464) from 12 samples were obtained and clustered into 99 OTUs (97% cutoff). Phyla or classes that did not represent at least 1% of the sequences in at least one sample were regrouped as "Others." Statistical significance was determined at $p < 0.05$ and comparisons used Kruskal–Wallis tests with FDR adjusted for multiple comparisons using the Benjamini and Hochberg method. Exact $p$ values can be found in the Source Data file. $n = 3$ samples per group. Uncertainties were reported as mean ± standard deviation (SD). Source data are provided as a Source Data file.

showing a transition from smaller *Choreotrichida* (such as *Strombidium* sp. and *Pelagostrobilidium* sp.) to larger *Tintinnida* sp. Changes in micro-zooplankton community structure in response to atrazine exposure revealed that atrazine at environmental concentrations can cause significant responses in some copepod larvae and ciliates (Fig. 5).

To distinguish whether these responses are caused by the direct toxicity of atrazine to microplankton, or whether they are caused indirectly by atrazine altering the phytoplankton community structure and particle size composition, thereby affecting microplankton foraging, the larvae of two copepods (*Oithona similis* and *Paracalanus parvus*) and two ciliates (*Euplotes* sp. and *Strombidium* sp.) were employed to evaluate their susceptibility to atrazine. Toxicity experiments at the species level showed that the EC50 values of atrazine in the four micro-zooplankton ranged from 0.35 to 3.26 µmol L$^{-1}$ (Fig. S5). The two copepod larvae (LC50 values of 0.35 µmol L$^{-1}$ and 0.73 µmol L$^{-1}$) were more sensitive to atrazine than were the two ciliates (LC50 values of 1.82 µmol L$^{-1}$ and 3.26 µmol L$^{-1}$). However, even the most sensitive zooplankton to atrazine, *Oithona similis*, had a mortality rate of only 14% after 24 h of exposure to 50 nmol L$^{-1}$ of atrazine. The other three examined species barely experienced a significant change at this dose, indicating that the direct toxicity of atrazine was not the main factor causing the change in the zooplankton community structure.

The results of correlation network analysis indicated that the *Bacillariophyta* (*Chaetoceros* and *Thalassiosira*) phytoplankton showed strong positive correlations with the micro-zooplankton taxa *Oithonidae* (*Oithona*), *Chrysopetalidae* (*Paleanotus*), *Ostreidae* (*Crassostrea*), and *Veneridae* (*Ruditapes*) (Fig. 6). Correspondingly, the *Dinophyceae* (*Getia*, *Woloszynskia*, *Adenoides*, and *Heterocapsa*) phytoplankton were significantly positively correlated with *Hypotrichia*, *Aegisthidae*, *Plehniidae* and *Euplotia*; *Haptista* (*Chrysochromulina*)

showed strong positive correlations with the micro-zooplankton taxa Arthropoda (*Acartia* and *Pyrgoma*) and Annelida (*Pseudopolydora*).

## Quantifying the ecological effects of triazine herbicides on phytoplankton primary productivity in global bay and gulf waters

The fitting results showed that the concentration-response curve of atrazine on the inhibition rate of phytoplankton chlorophyll *a* concentration at the community level conforms to the logistic functional equation (Fig. S3b, Table S2), indicating that the toxicity of atrazine to phytoplankton in seawater showed a good concentration correlation in situ, and the degree of inhibition of primary productivity can be estimated by the equivalent concentration of atrazine remaining in the water body at each station. The inhibition rates of the chlorophyll *a* concentration corresponding to 5.1 nmol L$^{-1}$, 11.9 nmol L$^{-1}$ and 35.2 nmol L$^{-1}$ of atrazine were 5%, 10%, and 25%, respectively (Fig. S3b, Table S2). According to this functional equation, among the 661 stations within the 7 sea areas of 5 continents, the primary productivity of 25% (167) of the stations was inhibited by more than 5% (Fig. 7) and that of 10% (67) of the stations was inhibited by more than 10% (95%CI 3.67, 4.34). The primary productivity in Camps Bay and Maputo Bay in South Africa, the Yellow Sea and Bohai Sea in Asia (including Jiaozhou Bay and Xiangshan Bay), and the Gulf of Mexico experienced the highest degree of impact, with median inhibition rates of 11.59%, 7.72%, and 6.60%, respectively. Some stations near estuaries reached 48.97%, 17.86%, and 12.23% inhibition rates (Fig. 7), respectively.

## Herbicide risk in global agricultural lands

The residue data of 59 widely used herbicides in the soils/water bodies of global agricultural areas showed that approximately 65.02% of the global agricultural land (approximately 23.63 million km²) was at some risk of herbicide pollution (i.e., RS > 0, Fig. 8); remarkably, 16.84% (approximately 6.12 million km²) of that was considered high risk (i.e., RS > 3). Regional analysis showed that Latin America, Europe, Asia, and North America had the highest proportions of high-risk areas, with proportions of 29% (approximately 1.76 million km²), 25% (approximately 0.94 million km²), 22% (approximately 2.49 million km²) and 11% (approximately 0.4 million km²), respectively (Fig. 8). Among them, Brazil, Ukraine, China and the United States were the countries with the largest land area at high risk in Latin America, Europe, Asia and North America, respectively. As the major grain-producing areas of China, the North China Plain, the Yangtze Plain, and the Songnen Plain were all within the high-risk class. Through runoff (Yangtze River, Yellow River, etc.) and rainwater transport, the adjacent Bohai and Yellow Seas faced potentially high herbicide stress risks.

## Analysis of herbicide residues in waters adjacent to high-risk agricultural areas

There is a good correlation between the use of herbicides in various agricultural areas worldwide and the residues of herbicides in their adjacent sea areas (Fig. 9). Brazil, Ukraine, China, and the United States had the highest proportion of high-risk areas. Atrazine is widely used in northern China at a dose of 0.5 kg km² (Fig. S6). The marine survey results (Fig. S7) also confirmed that atrazine is one of the most widely distributed herbicides (detected at all 64 stations) with the highest concentration (up to 7.26 nmol L$^{-1}$), which is closely related to its high application rate. In addition, the residues of 22 herbicides in 3 categories, which are widely used in agricultural areas, were detected in the waters of the Bohai and Yellow Seas (Fig. 1).

The detection rate of 10 triazine herbicides (Fig. S7a, d), 4 phenylurea herbicides (Fig. S7b, e), and 3 amide herbicides (Fig. S7c, f) in spring and autumn was 100%. The maximum total concentrations of the three herbicide categories reached 12.07 nmol L$^{-1}$, 6.97 nmol L$^{-1}$ and 2.79 nmol L$^{-1}$, respectively, which is several times higher than the

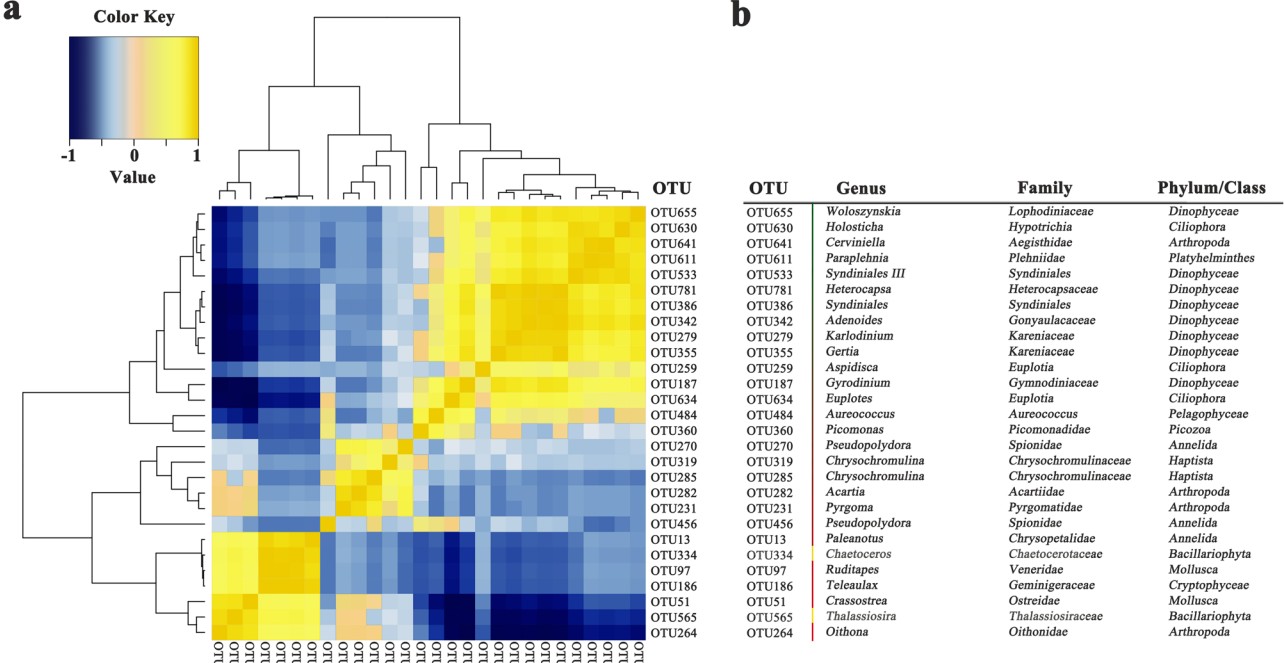

**Fig. 6 | Correlation patterns between phytoplankton and their associated micro-zooplankton at the genus level. a** The Spearman rank correlation coefficients between the phytoplankton and micro-zooplankton taxa. **b** The taxonomy of the phytoplankton and micro-zooplankton at the genus level. Correlations were calculated for phytoplankton (*Bacillariophyta*, *Dinophyceae*, and *Cryptophyceae*) and zooplankton groups with an abundance ≥ 1% in at least one sample. The groups marked unclassified in the high-throughput sequencing results were manually

blast-searched against the NCBI nucleotide collection and EzBioCloud Database. Sequences with the same taxonomic assignment at the genus level were combined. The Spearman rank correlation coefficients between the phytoplankton and micro-zooplankton taxa were calculated in R (version 3.5.3). Yellow represents a positive correlation, blue represents a negative correlation; and the darker the color, the greater the correlation coefficient (r). Source data are provided as a Source Data file.

EU standard for water safety (2.5 nmol L$^{-1}$). The spatial distribution of the herbicides at the 64 survey sites showed typical characteristics of terrestrial input, which was a greater distribution in the estuaries and coastal waters than that in the bay and gulf mouths and open sea. In particular, the estuaries of the Yellow River and the Yangtze River were areas with high herbicide concentrations. The residual concentration of herbicides in the Bohai Sea was significantly higher ($P < 0.01$, two-sided $t$-test) than that in the Yellow Sea. From the time series, the herbicide concentration in spring 2018 was generally higher than that in autumn 2017, which is closely related to agricultural production schedules.

## Discussion

Agricultural practices determine levels of food production and largely determine the state of the global environment[31,32]. In recent years, water pollution caused by herbicides has been spreading rapidly from freshwater to seawater, due to the increasing food demand caused by population growth and the acceleration of global urbanization[33–35]. Various herbicide residues have been detected frequently in coastal waters worldwide. Residual risk analysis of 59 herbicides in agricultural areas based on the PEST-CHEMGRIDSv1 global database[27] showed that 65.02% of the world's agricultural areas are at risk of pollution, 16.84% of which are at high-risk levels, which are concentrated in food production areas such as southeastern Brazil, western Europe, eastern China and the eastern United States (Fig. 8). The bay and gulf adjacent to these areas, such as Vilaine Bay, Mediterranean Sea, East Asia and the Gulf of Mexico, also have high herbicide residues (Fig. 2, Fig. 9). More than half of the herbicides are photosynthesis inhibitors[6]. Previous studies[9,10] have shown that herbicides such as triazines have obvious inhibitory effects on the photosynthesis of sensitive groups (e.g., *P. tricornutum* and *Chaetoceros* sp.). Based on this, scientists predict that such widespread herbicide contamination could have

immeasurable effects on oceanic productivity[36,37]. However, due to the inability to conduct in situ photosynthesis inhibition experiments on marine algae, there have been no research reports or clear conclusions on the impact of herbicide pollution on marine primary productivity. Therefore, by comprehensively analyzing the spatiotemporal distribution of herbicides at 661 bay and gulf stations worldwide from 1990 to 2022, we determined the median, third quartile, and maximum concentrations of 12 triazine herbicides of 0.54 nmol L$^{-1}$, 5.09 nmol L$^{-1}$ and 47.58 nmol L$^{-1}$, respectively (Table S1). Coastal investigations revealed that the concentrations of triazine herbicide in coastal waters, although relatively low, were also within this range. The results of microcosmic experiment confirmed that the inhibition degree of off-shore primary productivity under the three abovementioned atrazine concentrations reached 9.1%, 8.8%, and 18.8% (Fig. 7), indicating that herbicides had a great impact on primary productivity under environmental background concentrations. The composition of offshore herbicides is extremely complex, including 12 kinds of triazine herbicides alone (Fig. 1). The ecotoxicity of herbicides is largely determined by their types and residual concentrations. However, due to the large differences in the types and concentrations of herbicides between different sites (Fig. 1), there is no unified prediction scale or comparison benchmark for the effects of herbicides on primary productivity. To compare the herbicide contamination between stations, according to the relative toxicity of the typical atrazine herbicide and its homologs (Fig. S3a, Fig. 2), the concentrations of 12 triazine herbicides were converted into equi-effective concentrations of atrazine (Fig. 2), which were uniformly used to measure the degree of herbicide pollution at each site in this experiment. By establishing an in situ concentration-response relationship between atrazine and the chlorophyll $a$ concentration at the level of the phytoplankton community in seawater (Fig. S3b), the overall inhibition of 12 triazine herbicides on phytoplankton primary productivity was quantified for the first time. The

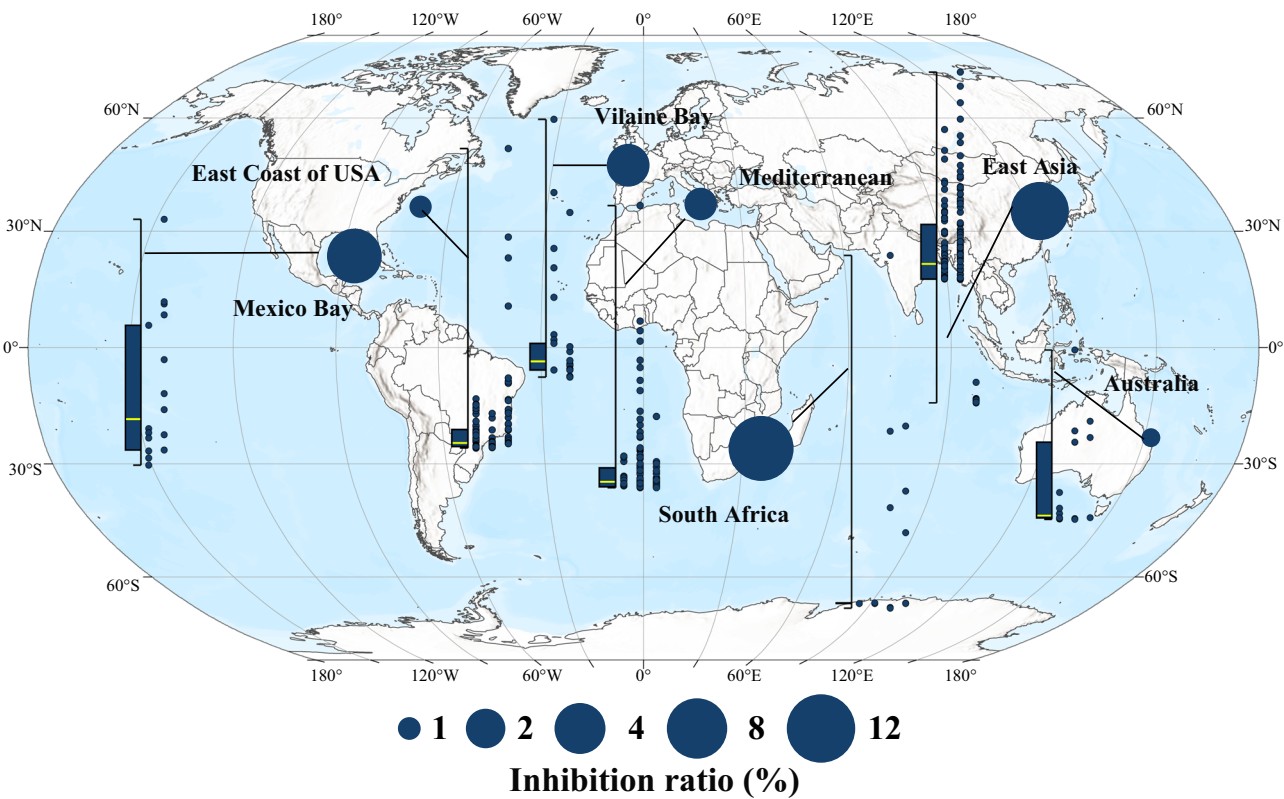

**Fig. 7 | Inhibition levels of environmental concentrations of triazine herbicides on phytoplankton primary productivity in typical bay and gulf worldwide.** The box-scatter plots show the inhibition levels. Each point in the scatter plot on the right corresponds to a survey site, which is arranged from left to right in chronological order. Box plot whiskers of the box chart on the left represent the minimum and maximum values, with bounds of the box representing The box chart on the left shows the first quartile, median (marked by yellow line), and third quartile of triazine herbicides on phytoplankton primary productivity at all sites in the corresponding sea area (*n* = 16, 128, 34, 271, 21, 168, and 22 independent survey sites, from left to right). The solid circle in each sea area represents the median value of the inhibition level at each station in that area, with a larger circle size indicating a larger median value. Source data are provided as a Source Data file.

results showed that among the 661 sites with herbicide residue data, primary productivity was inhibited by more than 5% at 25% of the stations and by more than 10% at 10% of the stations (Fig. 7). The primary productivity near the estuaries of Camps Bay and Maputo Bay in South Africa, the Yellow Sea and Bohai Sea in Asia and the Gulf of Mexico was affected by as much as 48.97%, 17.86% and 12.23%, respectively (Fig. 7). A 30-year time series of monitoring data indicated that the biomass of phytoplankton in the Bohai Sea showed an overall downward trend from 2005 to 2018[38], especially of dominant diatom groups such as *Chaetoceros* and *Coscinodiscus* sp., which showed a significant declining trend reaching up to 87% in some years. The period of rapid growth in herbicide use in China began in the mid-1990s[39], and there is a temporal correlation between these two findings. The primary production of marine phytoplankton worldwide is estimated to be $3-7 \times 10^{10}$ tons of carbon/year[11,40,41]. Although the coastal ocean accounts for only 7–8% of the ocean area, it contributes more than 25–28% of the global ocean primary productivity[42]. If herbicide pollution causes a 5% drop in coastal primary productivity, the annual carbon fixation amounts of phytoplankton will decrease by $3.75-8.75 \times 10^8$ tons, which is equivalent to the carbon fixation by the Amazon rainforest. These losses will be difficult to measure if the secondary effects on marine organisms at different trophic levels throughout the food chain are considered. Marine primary productivity is determined by phytoplankton abundance, community structure, and particle size, but the mechanisms by which herbicides affect phytoplankton primary productivity are currently unclear.

At present, most studies on the impacts of herbicides on marine primary productivity are based on single species or artificially mixed species[9,43,44]. Our experiment employed natural seawater and

identified as many as 117 OTUs, so it could more objectively reflect the in situ response trends of plankton in water bodies to herbicide stress at the community level. Under the stresses of low, medium, and high concentrations of atrazine, the proportion of Bacillariophyta (groups with the highest relative abundance) decreased from 77.4% to 15.5%, 10.7%, and 0.7%, respectively, while the proportion of Dinophyceae increased from 17.8% to 47.5%, 53.6%, and 79.2%, respectively (Fig. 3, Table S4). The dominant species changed from *C. tenuissimus* (Bacillariophyta) to *G. jinhaense* (Dinophyceae). While herbicides inhibited the growth of sensitive Bacillariophyta, the proportion of tolerant "opportunistic" groups, such as Dinophyceae, greatly increased, resulting in significant changes in the phytoplankton community structure and abundance. Diatoms are the most dominant phytoplankton in coastal oceans, contributing up to 40% of the total marine primary production. Moreover, it is well known that the production originating from larger phytoplankton, such as diatoms, is the portion most efficiently transferred to higher levels of the food web[45]. The inhibitory effect of herbicides on diatoms and the changes in the phytoplankton community will inevitably have a profound impact on the higher trophic levels of natural ecosystems. Changes in the community structure inevitably lead to changes in the particle size structure. Tiny cells have a higher surface area to volume ratio than larger cells[46,47]. Thus, for the same biomass, nano-phytoplankton contributes more to primary productivity than micro-phytoplankton[48]. However, it is still unclear how herbicides change primary productivity by changing the particle size of phytoplankton. As the reads from high-throughput sequencing techniques are relatively short, it is difficult to obtain information such as the full-length 18 S/ITS rDNA sequences in phytoplankton, and these techniques cannot provide particle size

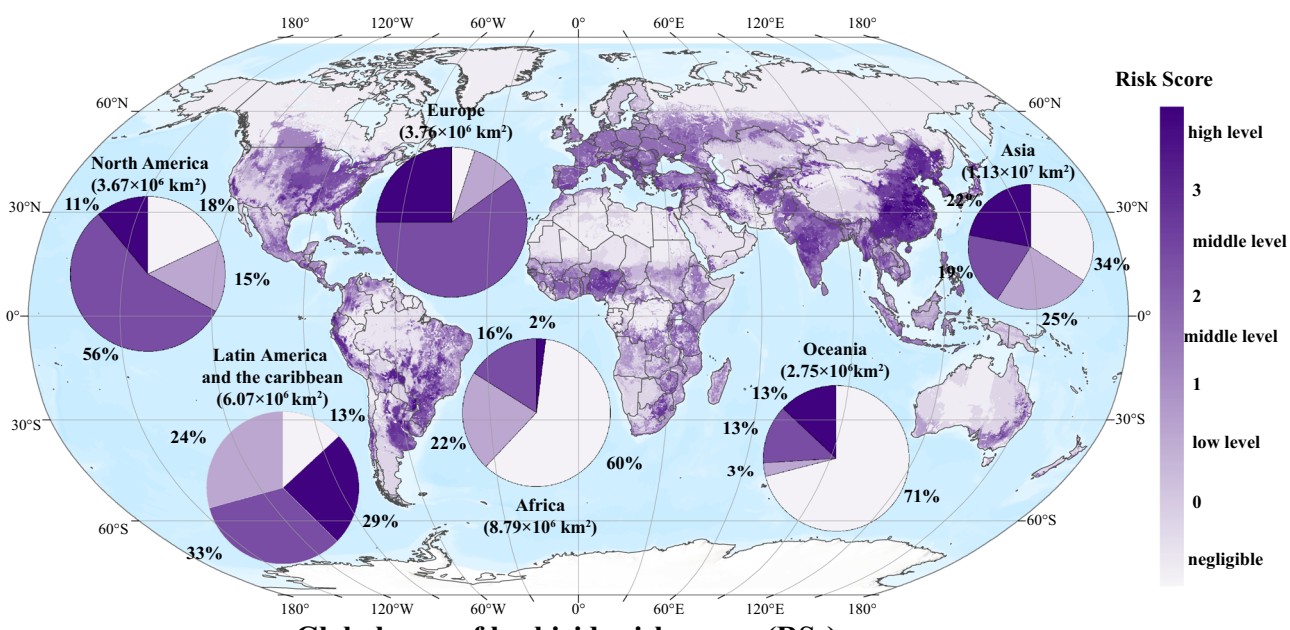

**Global map of herbicide risk scores (RSs)**

**Fig. 8 | Global map of herbicide risk scores (RSs).** The map has a spatial resolution of 5 arcmin, which is approximately 10 km × 10 km at the equator. The pie charts represent the fraction of agricultural land classed under different RSs in each region, and the values in parentheses above the pie charts denote the total amount of agricultural land in those regions. Herbicides data in each grid cell was based on the usage of 52 herbicides that are currently being widely used worldwide from the PEST-CHEMGRIDSv1 global database. Source data are provided as a Source Data file. we first tried to retrieve the average usage data of 52 herbicides that are currently being widely used worldwide from the PEST-CHEMGRIDSv1 global database.

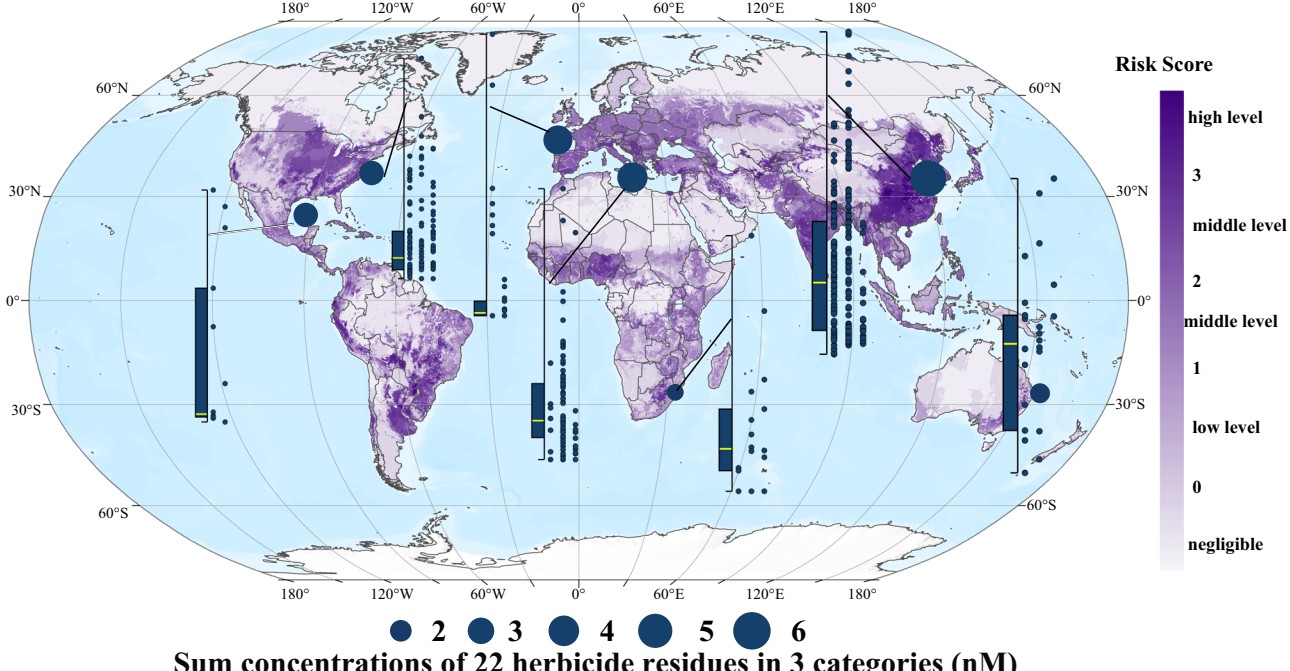

**Sum concentrations of 22 herbicide residues in 3 categories (nM)**

**Fig. 9 | Global map of herbicide risk scores in agricultural areas and 22 herbicide residues in 3 categories in adjacent sea areas.** The map has a spatial resolution of 5 arcmin, which is approximately 10 km × 10 km at the equator. The box-scatter plots show the residual concentration of herbicides. Each point in the scatter plots corresponds to a survey site, which are arranged from left to right in chronological order. Box plot whiskers of the box chart on the left represent the minimum and maximum values, with bounds of the box representing the first quartile, median (marked by yellow line), and third quartile of 22 triazine herbicides at all sites in the corresponding sea area ($n$ = 16, 128, 34, 271, 21, 168, and 22 independent survey sites, from left to right). The solid circle in each sea area represents the median value of the total concentration of 22 the triazine herbicides at each station in that area, with a larger circle size representing a larger median value. Herbicides data in each grid cell was based on the usage of 52 herbicides that are currently being widely used worldwide from the PEST-CHEMGRIDSv1 global database. Source data are provided as a Source Data file.

information at the species level. Via BLAST alignment of the sequenced nucleic acid sequences on the NCBI database, the sequences were linked with the particle size information of the species (Fig. 2). The results showed that under the stresses of low, medium, and high concentrations of atrazine, the proportion of nano-phytoplankton decreased from 74.2% to 37.5%, 48.5% and 22.1%, respectively, while that of micro-phytoplankton increased from 9.8% to 33.1%, 33.4% and 69.4%, respectively. This trend was also confirmed by the results of the size-fractionated chlorophyll a concentration (Fig. 4, Table S4). The proportion of nano-phytoplankton decreased from 60.02% to 23.68%, and the particle size composition of chlorophyll a transitioned from nano- to micro-sized. We revealed that the difference in the growth rate sensitivity of phytoplankton of different particle sizes to herbicide stress was an important reason for the change in phytoplankton particle size (Table S5). The intrinsic growth rates of nano-phytoplankton decreased by 18–53% under atrazine stress compared with a maximum of 18% for micro-phytoplankton, which increased even in the low-dose atrazine exposure. In summary, our experimental results elucidated, for the first time, the underlying mechanism by which herbicides inhibit phytoplankton primary productivity from three levels, including the inhibition of highly abundant-sensitive species, succession of community structure/particle size, and resulting growth rate reduction. Under the influence of these factors, the primary productivity of the groups treated with low, medium, and high doses of atrazine decreased by 9.1%, 8.8%, and 18.8%, respectively. Since the concentration range of atrazine (0.5–50 nmol L$^{-1}$) used in this experiment corresponded to the median and maximum concentrations of the equivalent triazine concentrations at all 661 stations, the results obtained here can characterize the degree of primary productivity inhibition at half of the stations and support the accuracy of the results estimated by the abovementioned quantitative model. Phytoplankton are the main food source of zooplankton and control the zooplankton community in a bottom-up manner through changes in phytoplankton community structure and particle size[49,50]. However, the secondary effects of herbicides on zooplankton communities due to phytoplankton changes are still unclear.

Our molecular experiment showed that under the stress of low, medium, and high concentrations of atrazine, the proportion of copepod larvae (the most abundant zooplankton group in natural waters) changed from 90.4% to 95.5%, 85.1%, and 31.1%, respectively, while the ciliate groups (such as Tintinnopsis and Euplotes) increased from less than 1% to 1.3%, 9.7%, and 50.7%, respectively (Fig. 5, Table S4, S6). This result is unexpected considering the results of a single species-based micro-zooplankton toxicity test. Herbicides mainly target the photosynthesis system, which does not exist in micro-zooplankton, and therefore are generally considered to have low toxicity to micro-zooplankton. According to existing literature reports, the herbicide concentration needs to reach the milligram level to have a significant impact on some micro-zooplankton, under pure culture conditions[51,52]. It should be noted that large differences in growth rate and taxon size across species do introduce some bias in DNA-sequencing-based results for micro-zooplankton communities, which is an inevitable challenge of the current metabarcoding approach. Even so, due to the advantages of time savings, high throughput, low cost, high sensitivity, and minimal destructiveness of this method, it is increasingly being used to determine the taxonomic composition of higher trophic levels[53–55]. Here, to reduce the bias caused by this method, we also included the method of microscopic morphological observation (Table S6), which can reflect the number of species of different sizes at different growth stages. The results of our toxicity experiments and correlation network analysis also showed that the reduction in copepods was not caused by the toxicity of the

herbicides but was significantly positively correlated with the sharp decline in nano-phytoplankton (e.g., *Chaetoceros* sp.) (Fig. S5, Fig. 6). Although low and medium doses of atrazine can inhibit the intrinsic and net growth rates of nano-phytoplankton, their relative abundance remained high (37.5% and 48.5%, respectively), so atrazine did not cause a sharp decline of copepod larvae. However, at high doses of atrazine, the abundance of nano-phytoplankton rapidly dropped to 22.1%, resulting in a serious lack of food for copepod larvae, whose abundance then plummeted to 31.1%. In contrast, the proportion of ciliates increased because nano-phytoplankton were not their main food source[56,57], which resulted in a zooplankton community transition from larger copepod larvae to smaller unicellular ciliates. Marine organisms of different particle sizes and different nutrient levels form a complex network structure through predatory relationships, and changes in any one population may affect the entire offshore ecosystem[58]. The change in particle size might modify the flux of matter from the main and traditional food chain to the microbial loop, which might prolong the transmission process of the marine food chain and reduce the transmission efficiency of primary productivity since the microbial loop was the most inefficient among multiple routes for algal primary production to transfer to higher trophic levels in terms of respiratory carbon losses[19,59,60]. As herbicides are considered a typical non-point-source agricultural pollutant, previous studies on the ecological risks of herbicides mainly focused on farmland and freshwater ecosystems. However, the global coastal pollution status of herbicides and their negative impact on marine life (especially phytoplankton) in natural environmental concentrations are poorly understood except for few special environments (e.g. the Great Barrier Refs. 61,62, Australia). Due to the hydrodynamic diffusion and self-purification effects of the oceans[63,64], the in situ impact of herbicides on primary productivity in seawater may be lower than that estimated by microcosmic experiments. To clarify this issue, we conducted a large-scale investigation of herbicide pollution in the Bohai Sea and the Yellow Sea, observing the spread of herbicides along the estuary, bay and gulf, and open sea. The results confirmed that the concentration of triazine herbicide at each sampling station was negatively correlated with its distance from the estuaries. However, all ten triazine herbicides were detected at nearly all the 64 stations. Even in the sea areas more than 50 nautical miles offshore (S11, for example), the total concentration of 10 triazine herbicides was still as high as 1.99 nmol L$^{-1}$, which was equivalent to or higher than the concentrations at some sites in the Bohai Sea. These indicate that even in coastal areas, herbicide pollution is also quite serious. Moreover, it should be noted that our estimate considers only triazine herbicides with serious pollution levels, and other herbicide types in the seawater have not been assessed, so these results may still underestimate the global impact of herbicides on marine primary productivity.

In the past 20 years, the global population has increased by 21%, while the area of arable land per capita has declined by more than 10%[65]. According to the latest global population forecast of the United Nations[66,67], the world population will peak at 10.4 billion by 2080. Due to the limited cultivated land, the trend of ensuring food security through herbicide application may be inevitable. Therefore, if herbicide reduction measures are not taken in the future, herbicide residues in global bay and gulf and coastal waters will show an overall increasing trend. At that time, the impact of herbicides on marine primary productivity will further expand. Whether this will lead to a serious decline in the potential of ocean carbon sequestration is a global environmental issue that deserves attention. In addition, terrestrial nutrients (such as overused fertilizers) will also be imported into the ocean together with herbicides, and the trade-off effect between the promotion of phytoplankton primary productivity by terrestrial input of

nutrients and the weakening effect of herbicides also deserves further study.

## Methods

### Data collection of the spatiotemporal distribution of herbicide residues in typical bay and gulf worldwide

Based on the Web of Science (WoS) Core Collection database, a total of 568 papers related to herbicides in bay and gulf were retrieved with the following search string: "Title = (bay* OR gulf *) AND Topic = herbicides* AND Published Year = 1990–2022". Data on the types and concentrations of herbicides in each bay and gulf were collected in each article, as well as the survey dates and geographical location. When the data in the papers were presented in graphical form, Plot Digitizer 3.3 software was used to extract the data from the figures. A dataset was compiled of the spatial distribution of herbicide residues in the coastal waters of typical bay gulf worldwide, which included 760 water samples and 4253 herbicide concentration data from 15 bays and gulfs (Fig. S8). The 15 bays and gulfs were clustered into seven sea areas, including the east coast of the United States, the Gulf of Mexico, France, the Mediterranean Sea, South Africa, East Asia, and Australia. The mean value of all survey data corresponding to each bay and gulf was used as the background value of herbicides in the coastal waters of the bay and gulf. In addition, from 1990 to 2022, the agricultural consumption of herbicides worldwide showed an overall increasing trend: the level was relatively flat from 1990 to 2000, increased rapidly from 2001 to 2012, and then stabilized (FAOSTAT (Jun 16, 2023) https://www.fao.org/faostat/en/#data/RP/visualize). The survey data of each sea area were roughly divided into three chronological stages (1990–2000, 2001–2011, and 2012–2022) to analyze (one-way ANOVA) the changes in herbicide residue concentration in each sea area over time.

### Potential effects of environmental concentrations of triazine herbicides on phytoplankton primary productivity

The results in the above section showed that triazine herbicides were the most widely distributed herbicides in all coastal waters (covering all 15 bays and gulfs), with the highest determined concentration (several times higher than the water quality safety standard[68], which sets a maximum allowable concentration of 0.1 µg L$^{-1}$ for any single pesticide and 0.5 µg L$^{-1}$ for the total pesticide concentration) and targeted inhibitory effects on phytoplankton photosynthesis, which makes them most likely to have a significant impact on primary productivity in the ocean. Therefore, the 12 triazine herbicides in the dataset above were selected for evaluation of their overall ecological impact on coastal primary productivity.

Toxicity equivalent conversion of triazine herbicides and microcosmic system construction: due to the large differences in herbicide types and concentrations between locations (Table S1), there is no unified prediction scale and comparison benchmark for the effects of herbicides on primary productivity. To truly reflect the stress effects on phytoplankton under the current herbicide pollution levels, the dose-effect relationship curves and toxicity equivalent database of the 12 triazine herbicides on a representative population (i.e., Fig. S3) of phytoplankton were firstly established. Then, according to the relative toxicity of the typical atrazine herbicide and its homologs, the concentrations of 12 triazine herbicides were converted into equi-effective concentrations of atrazine, which is uniformly used to measure the degree of herbicide pollution at each site in this experiment.

Selection of model organism: *Phaeodactylum tricornutum* (CCMP 2561) was used in this study for the toxicity equivalent conversion of triazine herbicides for three reasons: (1) it is one of the best-known cosmopolitan diatom species[69], especially in nutrient-rich coastal ecosystems[70,71]; (2) it is a model organism among diatoms, widely used in research on global climate change[72,73], eutrophication[74,75], marine pollution[76,77], etc.; (3) among five dominant diatom species

(*Chaetoceros muelleri* CCMP1316, *Phaeodactylum tricornutum* Pt-1, *Thalassiosira pseudonana* CCMP1335, *Nitzschia closterium*, and *Skeletonema costatum*), it is moderately sensitive to atrazine[10] and is therefore suitable for the calculation of the inhibition rate of atrazine on diatoms in offshore waters.

Toxicity equivalent conversion of triazine herbicides: The *P. tricornutum* Pt-1 cellswere cultivated in f/2 medium[78] at 20 °C under 60 µmol photons m$^{-2}$ s$^{-1}$ irradiance and a 12 L:12D photoperiod until they reached the exponential phase. Cell concentrations were measured microscopically using a Sedgewick-Rafter (SR) counting chamber (Phycotech, MI, USA) to monitor the growth of these cultures[79]. Briefly, the sample was homogenized gently before analysis. One milliliter of fully mixed sample was carefully dispensed into the counting cells. The average cell number of *P. tricornutum* Pt-1 in 20 sub-cells of the Sedgewick-Rafter counting chamber was counted. The algae were harvested by centrifugation at 4 °C ($5000 \times g$, 10 min). The cell pellets were washed twice and suspended in the f/2 liquid medium to an OD$_{730}$ of 3.0.

The test of short-term toxicity effects of the twelve triazine herbicides on *P. tricornutum* Pt-1 was performed on the Infinite M200 Pro plate reader (Tecan, Zurich, Switzerland) using excitation and emission wavelengths of 440 and 680 nm[80], respectively. 5 µmol of each herbicide was weighed accurately and diluted in 5 mL methanol as stock solution (1 mmol L$^{-1}$). A series of two-fold dilution of the stock solution were performed. The concentration series from 0.1 nmol L$^{-1}$ to 32 µmol L$^{-1}$ were arranged in 48-well microtiter plates (NEST Biotech, Shanghai, China) with three replicates. The procedure was detailed as follows. 10 µL herbicide solution was added into the corresponding well and dried in a clean bench. Then 1 mL algae suspension prepared as described in section 2.1 was added. After being shaken for 10 min, the plates were incubated at 20 °C as described above. The fluorescence intensity was detected every 12 h and the data at the 72 h were selected for the determination of the inhibition effects of each herbicide on the chlorophyll *a* fluorescence of cell *P. tricornutum* Pt-1. Toxicity data were recorded for the description of the complete effect range (0–100%) for each individual triazine herbicides. The best-fitting model for each concentration-response relationship was chosen as described by Scholze et al.[81].

Based on the concentration addition (CA) model[82] and the single-substance concentration-response curves, the residual distribution data of the 12 triazine herbicides in each bay and gulf in the dataset mentioned above were normalized according to Eq. (1):

$$\sum_{i=1}^{n} \frac{c_i}{ECx_i} = 1 \tag{1}$$

In this equation, $n$ is the number of herbicide types contained in the water body; $ECx_i$ is the concentration of the single substance $i$ provoking x% effects; $c_i$ denotes the concentration of component $i$ in the mixture; and $c_i/ECx_i$ is called the toxicity unit (TU) of component $I$, i.e[83]. Combined with the best-fitting model mentioned above and the Eq. (1), the detected concentration of triazine herbicide $i$ ($c_i$) in a specific sea area/station that was obtained from a marine survey can be converted into the concentration of atrazine with the same TU. Then the total concentration of various herbicides was expressed by the TEQ of atrazine to normalize the various herbicide homologs remaining in the water body, which made it possible to assess accurately the joint effects of multiple herbicides in natural seawaters (Fig. 2). The joint toxicity of the twelve triazine was predicted based on the concentration-response curve of atrazine on the phytoplankton chlorophyll a concentration at the community level.

Although this toxicity equivalent conversion based on the concentration addition model[82] has been widely used in the study of various pollutants[84–86]and makes it possible to assess the combined toxicity of multiple herbicides in natural seawater, it is also necessary

to admit that it may introduce some bias due to the varying susceptibility of different species to herbicides. To minimize this effect, we selected the most representative species possible for the single-substance concentration-response experiments, as mentioned above.

Construction of the herbicide microcosm system: Through the establishment of a controllable herbicide microcosm system, the interference of other environmental factors was eliminated to accurately reflect the impact of the current environmental concentration of triazine herbicides on offshore primary productivity. On 5 July 2018, 1200 liters of clean seawater (with no detected herbicide residue) was collected from the estuary of Shilaoren Bay (120°49′E, 36°09′N), Qingdao, China, and filtered through a 200 μm-mesh net to remove large particles. In the laboratory, four atrazine doses (0, 0.5, 5, and 50 nmol L$^{-1}$) were used in the treatment groups: (1) control check (CK), consisting of uncontaminated seawater; (2) 0.5 nmol L$^{-1}$ atrazine; (3) 5 nmol L$^{-1}$ atrazine; and (4) 50 nmol L$^{-1}$ atrazine. Each treatment sample consisted of 80 L of seawater in a transparent polycarbonate bottle (100 L), and each treatment had three replicates (Fig. S2a). The mouth of each bottle was covered with a parafilm membrane to ensure gas exchange while preventing contamination. The bottles were incubated at room temperature (25 °C ± 3 °C) for 30 d in the laboratory under natural light conditions, and subsamples were collected from each bottle on days 0, 1, 2, 4, 7, 14, 21, and 30 for subsequent analyses.

Potential effects of environmental concentrations of triazine herbicides on the particle size structure of phytoplankton: chlorophyll *a* was extracted and measured with the method described by Yentsch and Mezel et al.[87]. Briefly, a 0.5-liter subsample from each bottle was sequentially filtered through 20, 2, and 0.2 μm pore-size polycarbonate filters. The phytoplankton retained on the 20, 2, and 0.2 μm pore-size filters were designated as micro-phytoplankton, nano-phytoplankton, and pico-phytoplankton, respectively. Following filtration, the filters were wrapped in foil and frozen at −80 °C for fluorometric analysis. Filters were extracted using acetone solution (acetone: water = 9:1, v/v) overnight in the dark, and the extract was centrifuged at 5000 × *g* for 10 min to remove cell debris. Chlorophyll was quantified by fluorometry analysis (Hitachi F-2500), using an excitation wavelength of 470 nm and an emission of 680 nm. By comparing the changes in the Chl *a* concentration of each particle size in the control group and the experimental group, the response trend of the particle size composition of phytoplankton to herbicide stress could be characterized. Statistical analyses were performed using SPSS Statistics (version 22; IBM Corporation, Armonk, NY). A two-sided *t*-test was performed to identify the difference of Chl *a* concentration between CK group and every atrazine-treated group, with two-sided 95% confidence interval and considered significant when $P \leq 0.05$.

Potential effects of environmental concentrations of triazine herbicides on the phytoplankton community structure by high-throughput sequencing: the samples of each treatment group that were collected on the 21st day were used for phytoplankton community structure analysis. For DNA extraction, the phytoplankton in 1 L of each water sample was filtered onto a 0.22 μm pore-size polycarbonate membrane (47 mm diameter, Millipore, MA, USA) and stored at −80 °C for analysis. Blank filters were treated in the same manner as the samples. Total DNA was extracted from each filter using a FastDNA Spin Kit for soil (MP Biomedicals, Solon, OH, USA) according to the manufacturer's instructions. A NanoDrop spectrophotometer (NanoDrop, Wilmington, USA) was used to measure the genomic DNA concentration and purity. The V1-V3 hypervariable region of the phytoplankton 18 S rRNA gene was amplified with the eukaryote-specific primers 18 S-82 F (5′-GAAACTGCGAATGGCTC-3′) and Ek-516 R (5′-ACCAGACTTGCCCTCC-3′)[88]. The coverage and specificity of this primer set for the major taxonomic groups of phytoplankton and micro-zooplankton were evaluated using SILVA TestPrime 1.0 with version 138.1 of the SILVA SSU Ref database[89], with no mismatches allowed. This primer set was previously shown to theoretically amplify the 18 S

rDNA region from all major taxonomic groups (Fig. S9). The PCR samples were sent to Shanghai Majorbio Bio-Pharm Technology Co., Ltd. (Shanghai, China) and sequenced (2 × 300) on an Illumina MiSeq platform. The DNA concentrations in parallel incubations of blank filters were below the detection limit, and no detectable amplification of 18 S rRNA gene products was observed.

Raw data were quality-filtered using QIIME software (v1.8), and the paired reads were merged using FLASH software (v1.2.7). Sequences meeting the following three criteria were included in the downstream analyses: 200<sequence length <600, mean quality >30, ambiguous bases <1, and homopolymer length <6. Erroneous and chimeric sequences were further eliminated by USEARCH. Then, the remaining sequences were clustered into operational taxonomic units (OTUs) according to a 97% sequence similarity threshold using UCLUST (v1.2.22). OTUs containing fewer than five sequences were eliminated. The same number of sequences (30,000) were randomly selected from each sample for standardization. Statistical analyses were performed using the R base package (R Foundation for Statistical Computing, Vienna Austria). The association between atrazine concentration and phytoplankton community was investigated by Spearman's rank correlation coefficient, and then, clustering analysis was carried out to clarify the impact of atrazine treatment.

In addition, the DNA sequences corresponding to OTUs with significant changes in relative abundance under herbicide stress were extracted. By aligning these sequences with the NCBI BLAST database, the phytoplankton species with the highest sequence similarity were identified, and the particle size information of the corresponding species of each taxon was obtained. The corresponding results were helpful to further analyze the relationship between the changes in the particle size structure and the community structure of phytoplankton under different environmental concentrations of triazine herbicides.

Potential effects of environmental concentrations of triazine herbicides on phytoplankton growth rate and energy flow transfer: the changes in community structure and particle size of phytoplankton can modify the overall growth rate and even the energy flow of phytoplankton to higher trophic levels. Based on the classical model (dilution method)[90–92] of energy flow transfer between phytoplankton and micro-zooplankton, the effect of atrazine on the net (NGR) or intrinsic ($\mu$) growth rate of phytoplankton and the grazing rate (*g*) of zooplankton in natural seawater were measured. Briefly, subsamples of each treatment group (mentioned above 2.2.1) that were collected on the fourth day (according to the changes in the Chl *a* concentration for each particle size in Fig. 4) were considered as the initial seawater (ISW) samples, and particle-free water (PFW) samples were prepared by filtering the ISW samples through Millipore filters (pore size: 0.2 μm). Then, the ISW was diluted with PFW to five target dilutions of 100%, 80%, 60%, 40%, and 20% in 2.8 L transparent polycarbonate bottles. The incubation volume was 2.5 L with triplicates. All the bottles were incubated in a water incubator for 24 h, and then the subsample of each dilution gradient was filtered sequentially through 20, 2, and 0.2 μm pore-size polycarbonate filters and stored in the dark at −20 °C for further analysis. The phytoplankton retained on the 20, 2, and 0.2 μm pore-size filters were designated as micro-phytoplankton, nano-phytoplankton, and pico-phytoplankton, respectively.

The dilution approach rests on two fundamental assumptions: (1) that dilution ratio has no effect on phytoplankton growth rate, and (2) that grazing impact is a linear function of the experimental dilution[93]. Rates of phytoplankton growth and grazing mortality can be inferred from observed changes in population density following incubations of different dilutions of natural seawater. The changes in phytoplankton density over time

for the above dilution series can be represented appropriately by the following exponential equations.

$$P_t = P_0 e^{(\mu - g)t} \qquad (2)$$

$$P_t = P_0 e^{(\mu - 0.8g)t} \qquad (3)$$

$$P_t = P_0 e^{(\mu - 0.6g)t} \qquad (4)$$

$$P_t = P_0 e^{(\mu - 0.4g)t} \qquad (5)$$

$$P_t = P_0 e^{(\mu - 0.2g)t} \qquad (6)$$

where $P_0$ and $P_t$ are the initial and post-24h incubation time (t, $d^{-1}$) phytoplankton biomass, respectively; $\mu$ ($d^{-1}$) and $g$ ($d^{-1}$) are values of phytoplankton intrinsic growth rate in the absence of grazing and the phytoplankton mortality rate due to herbivory, respectively. Phytoplankton net growth rate ($d^{-1}$) is related to grazing and mortality by Eq. 6 at each dilution level.

$$\left(\frac{1}{t}\right)\ln\frac{P_t}{P_0} = \mu - g \qquad (7)$$

The negative slope of this relationship is the grazing mortality rate ($g$); the Y-axis intercept is the intrinsic growth rate ($\mu$). Phytoplankton net growth rate (NGR) is subtracted by μ and g.

### Secondary effects of triazine herbicides at environmental concentrations due to structural changes in phytoplankton

Effects of triazine herbicides at environmental concentrations on micro-zooplankton community structure in natural seawater: with reference to the high-throughput sequencing of phytoplankton, the specific primer 82 F/ Ek-516 R was used to amplify the 18 S rRNA gene of micro-zooplankton in herbicide microcosm system mentioned above.

Morphological identification of micro-zooplankton: for each micro-zooplankton sample, 250 mL of seawater from each treatment group collected on the 21st day was fixed with formalin (final concentration 10%, v/v) and set aside to settle in the dark at 4 °C for 3–4 days. Then, the supernatant was removed, and the concentrate was carefully transferred to 1.5 mL microcentrifuge tubes. Micro-zooplankton in at least 20 random fields were identified and counted by using a Sedgwick-Rafter counting chamber under a light inverted microscope (magnification ×100). Ciliates, heterotrophic flagellates, and metazoan larvae were morphologically identified down to the genus or the lowest taxonomic level possible. Works used for identifications included Campbell[94], Dolan[95], and Bachy et al.[96].

Direct toxicity of atrazine to sensitive/tolerant microplankton groups: the four experimental micro-zooplankton were isolated from the coastal waters of Shilaoren Bay (120°49′E, 36°09′N) and cultured in artificial seawater medium on 17 July 2021. Before the toxicity test, the copepods and ciliates stored in the laboratory were transferred with a micropipette to a new culture medium. Individuals of a uniform size and with good activity levels were selected and precultured at 25 °C for 48 h.

Atrazine toxicity to copepods: a 96-h toxicity test was conducted on the two species of copepod in 50 mL beakers containing 20 mL of artificial seawater (S = 28–30‰; pH = 8.2 ± 0.1). A serial 2-fold dilution of atrazine concentrations (0.1 nmol $L^{-1}$ to 32 μmol $L^{-1}$) was prepared by adding appropriate amount of atrazine stock solution, including 20 copepods in 3 replicates for each concentration. Copepods were transferred into test solutions using disposable Pasteur pipettes in a minimum of seawater to reduce dilution. The two species of copepods were fed daily with *Isochrysis galbana* ($1.0 \times 10^5$ ind m $L^{-1}$). The stock

cultures were maintained in a climate-controlled room/chamber at 20 ± 1 °C and a 14:10 h light:dark cycle. After 96 h, the animal's mobility was examined by stereomicroscopic observation[90]. The observed immobility was determined by lack of movement when gently prodded or blown with water. By the end of the experiment, the survival rate of copepods in the control group should be greater than 80%[97,98]. The data at 96 h were recorded to calculate the mortality (0–100%) of each copepods across a range of atrazine exposures. The best-fitting model for each concentration-response relationship was chosen according to Scholze et al.[81].

Atrazine toxicity to ciliates: the toxicity tests to ciliates were conducted in a 24-well plate, and 1 mL of boiled rice and wheat grain culture solution was added to each well. No more food was added during the subsequent toxicity tests. A serial 2-fold dilution of atrazine concentrations (0.1 nmol $L^{-1}$ to 32 μmol $L^{-1}$) was prepared by adding appropriate amount of atrazine stock solution. Both the atrazine-treated and control groups were established with 3 replicates. Twenty precultured ciliates were added to each microwell with a micropipette and incubated in a constant-temperature incubator at 25 °C, oxygen saturation >45%, and photoperiod of 14:10 h light:dark. After 96 h, the number of surviving ciliates in each group was counted under a stereomicroscope and recorded. Individuals that were incapacitated or had significant morphological changes were considered dead.

The 96 h LC50 value of atrazine for each zooplankton species was obtained by performing a probit regression analysis and a chi-squared goodness of fit test on the experimental data with SPSS17.0[99]. Taking the logarithm of the atrazine concentration as the x value and the mortality of the test species as the y value, a regression curve was drawn and a regression equation was fitted to the data.

Correlation analysis of sensitive species of phytoplankton and microplankton: the phytoplankton and micro-zooplankton OTUs that were significantly ($p < 0.05$) affected by at least one atrazine level (0.5, 5, or 50 nmol $L^{-1}$) were used in co-occurrence network analysis. The relative abundances of phytoplankton and micro-zooplankton OTUs in the control and atrazine-treated groups were employed to construct Bray-Curtis similarity matrices. The Spearman rank correlation coefficients between the phytoplankton and micro-zooplankton taxa were calculated in R (version 3.5.3). To reduce noise and the occurrence of false-positive predictions, only strong correlations ($|r| \geq 0.9$, $p \leq 0.05$) were selected for Gephi network visualization (version 0.9.2).

### Quantifying the ecological effects of triazine herbicides on phytoplankton primary productivity in global bay and gulf waters

The atrazine stock solution (100 mmol $L^{-1}$) was prepared by accurately weighing 1 mmol of atrazine standard and dissolving it in 10 mL of methanol. Then, a series of 2-fold serial dilutions of the stock solution were performed to obtain atrazine standard solutions with concentrations of 1 nmol $L^{-1}$ to 320 μmol $L^{-1}$. One milliliter of each of the abovementioned standard solutions was added to a series of 2 L conical flasks in triplicate. To prevent methanol from interfering with the experimental results, the methanol in each flask was blown dry with $N_2$ in a fume hood, and then one liter of the clean seawater mentioned above was added to each of the flasks and fully mixed on a shaker for a final concentration range of atrazine ranging from 0.001 nmol $L^{-1}$ to 320 nmol $L^{-1}$. After 72 h of incubation at 20 °C under an irradiance of 60 μmol photons $m^{-2} s^{-1}$ and a 12 L:12 D photoperiod, the concentration of chlorophyll *a* (representing the phytoplankton primary productivity) in each treatment group was measured, and the inhibition rate (0–100%) was calculated. The best-fitting model for the concentration-response relationship between chlorophyll a and atrazine concentration was chosen as described by Scholze et al.[81].

By substituting the equi-effective concentrations of atrazine corresponding to each bay and gulf that were obtained in the above equation, the degree of inhibition of triazine herbicides on

phytoplankton primary productivity in global bay and gulf waters can be estimated.

## Herbicide risk on global agricultural land

Application rates (ARs) of herbicides. At a 5 arcmin resolution (approximately 10 km × 10 km at the Equator), the Earth's land area (excluding the North and South poles) is divided into 7.3 million grid cells. Herbicide usage in each grid cell was estimated with reference to the method established by Maggi et al.[27]. In short, 175 crops were classified into six dominant crops (alfalfa, corn, cotton, rice, soybean, and wheat) and four aggregated crop classes (fruit and vegetables, orchards and grapes, pasture and hay, and others). The top 20 herbicides (based on the CHENGRIDS database[27]) used for each crop class in 2015 were selected as the statistical range of the global herbicide geographical distribution (approximately 84% of the herbicide use in 2015). The global georeferenced, crop-specific, annual herbicide application rates were obtained from the PEST-CHENGRIDS database[27] and constrained against the country-specific pesticide use data reported in the FAOSTAT database[28] on chlorophyll *a* content and phytoplankton abundance.

The crop-specific risk quotient (RQ) of each herbicide was calculated as the ratio between the predicted environmental concentration (PEC) and the predicted no-effect concentration (PNEC) (that is, $RQ = PEC/PNEC$).

The PEC in this study refers to the noncumulative environmental concentration due to the lack of historical data on herbicide use. The spatially explicit approach of the Environmental Potential Risk Indicator for Pesticide version 2.1[100] was employed to estimate the PEC value of herbicide *i* on crop *j* ($PEC_{i,j}$) as follows:

$$S_{i,j} = VDT_j \frac{AR_{i,j}}{100} \tag{7}$$

$$PEC_{i,j} = \frac{S_{i,j}}{4500} \int_0^t \frac{C^{(t)}dt}{100} \tag{8}$$

Note: $S_{i,j}$ is the absorption amount of the herbicide *i* on crop *j*; $AR_{i,j}$ is the usage amount of herbicide *i* on crop *j*; $VDT_j$ is the percentage of soil coverage found in the coverage table (%)[28]; and $C^{(t)}dt$ is the residual proportion of herbicide after *t* days of application (based on $DT_{50}$ and DT90 values).

## PNECs

The PNEC values of herbicide i in soil ($PNEC_i^{SL}$) and water ($PNEC_i^{SW}$) were estimated from the LC50 values of earthworms and fish respectively as follows:

$$PNEC_i^{SL} = \frac{LC50_i^{earthworms}}{1000} \tag{9}$$

$$PNEC_i^{SW} = \frac{LC50_i^{fishes}}{1000} \tag{10}$$

Assuming that the number of herbicide residues in a grid cell is i, the risk point (RP) was calculated as the log-transformed sum of all RQs of each grid cell that is,

$$RP = \log \sum_i RQ_i. \tag{11}$$

According to the species sensitivity distribution data of 59 pesticides reported by Nagai et al.[101], RP values are divided into four grades: RP ≤ 0, negligible; 0 < RP ≤ 1, low risk; 1 < RP ≤ 3, medium risk; and RP > 3, high risk. RP ≤ 0 means that the probability of any species being affected is less than 5%; RP > 3 means that the probability of a randomly selected species being affected by the herbicide exceeds 90%[101].

The above work will provide a comprehensive understanding of herbicide residue status and risk levels in agricultural soils/runoff in various regions of the world. The maximum safety threshold of total herbicide usage in farmland ecosystems can be obtained by comparing the herbicide pollution in adjacent sea areas and its inhibitory effect on primary productivity.

## Analysis of herbicide residues in waters adjacent to high-risk agricultural areas

Based on the global geographic distribution map of herbicide risk described, the Bohai Sea and Yellow Sea adjacent to the North China Plain (RP value > 3) were selected as study sites to determine the correlation between the ecological risk level of herbicides in farmlands and the degree of inhibition of phytoplankton primary productivity in the adjacent sea areas. Environmental surveys were carried out in autumn (August 29–September 24) 2017 and spring (March 26–April 16) 2018 to determine the spatial distribution and seasonal variation in 22 herbicides that are widely used in upstream agricultural areas in the Bohai Sea and Yellow Sea[39]. The distribution of the 64 monitoring stations is shown (Fig. S10), covering the estuary areas of the Yangtze River and the Yellow River. Surface seawater (0–50 cm) samples were collected. For each sample, 5 L of water was collected using Niskin bottles and filtered through a 200-μm-mesh net to remove large zooplankton. Then, the water samples were immediately transported to the laboratory for further treatment. The concentrations of 10 triazine herbicides and their derivatives (i.e., atrazine, desethylatrazine, propazine, simazine, terbutryn, ametryn, dipropetryn, deisopropylatrazine, prometryn, and prometon), six phenylurea herbicides (chlortoluron, diuron, fluometuron, isoproturon, metobromuron, and linuron), and six amide herbicides (propachlor, alachlor, metolachlor, metazachlor, propanil, and napropamide) in the water samples were analyzed. 400 mL of water treatment was adjusted to pH 7 and filtered through a 0.7-μm GF/F filter (47 mm). Then, 10 μL of internal standard solution (acetonitrile) containing 1 μmol L⁻¹ atrazine D5, diuron D6, and metolachlor D6 was added. After successive activation with methanol (10 mL) and deionized water (10 mL), Oasis HLB cartridges (Milford) were employed to pre-concentrate the analytes from the water samples, with a vacuum maintained at 400 mmHg with a Vac Elut SPS 24 vacuum manifold (Agilent Technology). The cartridge was then dried with N₂ for 30 min and then eluted with 5 mL of methanol. The eluant was evaporated under N₂, dissolved in a 1 mL mixture (acetonitrile: water = 10: 90, v/v), and filtered through a 0.22-μm nylon filter (MoBio). The subsamples were analyzed with high-performance liquid chromatography (HPLC) coupled to electrospray ionization tandem mass spectrometry (ESI-MS/MS) as described by Mazzella et al. (2008). The limits of quantifications (LOQ) were from 0.004 to 0.016 nmol L⁻¹ for the triazines, from 0.007 to 0.023 ng L⁻¹ for the phenylureas, and from 0.005 to 0.01 nmol L⁻¹ for the amides (Supporting Information Table S1). Concentrations below these LOQ were not included for the calculation of summed herbicide concentrations, and the values were corrected to two decimal places. Both external and internal calibration procedures were performed and the concentration ranges for the calibration curves were from 0.01 to 100 nmol L⁻¹ for the triazines and from 0.1 to 100 nmol L⁻¹ for phenylureas and amide. The concentrations of atrazine D5, diuron D6, and metolachlor D6 were used for the respective internal quantifications of triazines, phenylureas, and amides. Blank controls were set during sample preparation and liquid chromatography-tandem mass spectrometry (LC−MS/MS) analysis to avoid any false-positive results.

The degree of inhibition of phytoplankton primary productivity by herbicides in the Bohai Sea and the Yellow Sea was obtained by substituting the triazine herbicide concentration data obtained from the survey into the concentration-response equation of the inhibition rate of atrazine on the phytoplankton chlorophyll *a* concentration at the community level, as described in the section above.

**Reporting summary**

Further information on research design is available in the Nature Portfolio Reporting Summary linked to this article.

## Data availability

The raw sequencing data generated in this study have been deposited in the National Center for Biotechnology Information (NCBI) Sequence Read Archive (SRA, http://www.ncbi.nlm.nih.gov/Traces/sra/sra.cgi) under accession number PRJNA913270 (https://www.ncbi.nlm.nih.gov/). The data of global georeferenced, crop-specific, annual herbicide application rates were obtained from the PEST-CHEMGRIDSv1 global database (https://doi.org/10.7927/weq9-pv30). Herbicide data are available in Tables 1, 2, and 5. Source data are provided with this paper.

## Code availability

The code used to calculate pesticide risk scores is provided as a MATLAB file available via Figshare at https://doi.org/10.6084/m9.figshare.23735130.

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

## Acknowledgements

This work was supported by the NSFC project grant No.42188102 (Y.Z.), No. 31600399 (L.Y.), No. U1906216(Y.Z.) and the Natural Science Foundation of Shandong Province grant ZR2022MD021(L.Y.).

## Author contributions

L.Y. designed the study, conducted the data analysis, and wrote the manuscript. X.H. collected the data of the spatiotemporal distribution of herbicide residues in typical bay and gulf worldwide. S.R. reviewed the manuscript and acquired funding. Y.Z. designed the study, interpreted the results, wrote and reviewed the manuscript. All authors commented on the manuscript.

## Competing interests

The authors declare no competing interests.
