## [Peer Review File · Nature Communications]

Reviewers' Comments:

Reviewer #1:

Remarks to the Author:

The authors have submitted a comprehensive work discussing the impact of herbicides on coastal environment. I have the following comments/question for the authors to address:

1. The authors narrowed their study to bay/gulf areas. However the reason for that is unclear. I can understand that the authors would want to avoid truly offshore areas, but why not general coastal zones or national coastal zones? The current selection of gulfs/bays does limit the applicability of the study as these areas generally are less mixed and very dependent on the geology and hydrology of the bay and pollution may therefore be a worst-case scenario which is not applicable to a broader context. At the same time, because of this, care should be taken in overgeneralizing the results to all coastal waters.
2. Why was *P. tricornutum* used as to determine the dose response effect given? Is it that representative for bay/gulf areas globally? Yet, the dominant species in the microcosm was *C. tenuissimus*, why was this one not used for the dose response curves?
3. For the section 2.2.3, I could not find the information on quality control and blanks for DNA sequencing? Where blank filters analyzed to account for potential contamination? Where the primers confirmed to work on all species that you expect to detect? Also, DNA is not always a good proxy for cell density and cell concentration.
4. Given that DNA based are also used to determine effects on the microzooplankton, it is unclear how the authors accounted for bias in the sequencing in not picking up cryptic species or larval species above a certain DNA threshold and therefore incorrectly reporting this as a change in species density. Additionally, biomass determines DNA quantity. Larger animals, larger larvae will lead to more DNA without being necessarily more dominant in terms of species number. This is particularly relevant for the larvae stage of zooplankton where there can be large variations in growth rate and size across species taxa.

The overall methodology on the concentration response curves and risk assessment approach of the paper is clearly detailed, executed according to current statistical methodology and risk assessment practices and well reported.

The overall resolution of the figures in the pdf is low and should be improved prior to publication.

Reviewer #2:

Remarks to the Author:

This publication from Yang et al. aims at predicting the impact of the most used herbicides on marine primary productivity at a global scale. The study is based on (i) a meta-analysis to collect concentrations of herbicides all around the world and (ii) some experimental work about the effect of herbicide concentrations on phytoplankton and microzooplankton growth and community composition. Then a combination of both approaches is used to predict the environmental effect at a global scale. Authors detected a sound effect of herbicide concentration (namely atrazine) on phytoplankton, from nano- to micro-sized community (mainly due to change in a few species dominance) and a strong negative effect on pico-plankton growth rate. There is also a measured effect on micro-zooplankton with a copepod-dominated community to a heterotrophic protist-based community (ciliates). Prediction on primary production inhibition under current herbicide stress is expected to be measurable for 25% of analyzed sites (at least a 5% inhibition).

The subject of the paper is fascinating: I would like to thank the authors for having dealt with this very interesting question that is sometimes complicated to deal with because of the food industry lobbies. I like this work a lot because the research plan is well thought and a lot of experiments and analyses have been done to decipher all the potential confounding effects. It is hard to measure and predict the effects of herbicides in the environment and the authors have been able to couple lab experiments and meta-analysis to look at the big picture, which makes this study pretty original. However, I think there is still a lot of work to improve the quality of the paper in term of writing to make it clearer, more concise and stronger. The flaws of the paper are more in the interpretation (which is pretty short compared to the other sections) and referencing than in

the data analysis and methodology that have been well done. The work is overall reproducible which is another good point.

Main concerns:

(i) the introduction needs to be revised to improve the references and sometimes clarify the statements.

(ii) the discussion section length is weirdly balanced compared to the results section, the latter being way longer. This section also needs to be improved with more discussion related to the existing literature.

(iii) I do not think the figures in the current version are of good quality in term of clarity and information conveyed.

Minor comments:

Line 55- the reference Yang et al. (2019) might not be the correct one to cite here.

Line 67- the reference Suttle et al. is absolutely not referring to the fact that half of the photosynthesis comes from phytoplankton. Field et al. 1998 is a better choice here. Be careful, this kind of bad referencing could be considered as a red flag.

Line 76- 'in situ in seawater' sounds weird, keep in situ or in seawater

Line 78- here and along the manuscript, be sure that you talk about primary productivity and not primary production.

Line 81- Once again, not sure reference 7 is suited here.

Line 87- Protozoa is kind of an old term, maybe better to use the term 'heterotrophic protists'

Line 88- Please, specify what classic food chain means (I guess this is the classic phytoplankton->heterotrophic protists->zooplankton). And maybe cite also a review a more classical paper to reference this ecological statement (e.g., Worden et al. 2015 Science)

Line 95-99- Please, re-phrase the sentence as this is not proper English. Maybe something like: 'Therefore, exploring the response of phytoplankton to environmental stress caused by herbicides using sensitive species as case studies is expected to explain the effects of modern intensive agriculture on marine primary production.'

Line 110- 'in global bays' is not proper English. Discard the word globally at the end of the sentence, it is unnecessary.

Line 159- quotation from what reference??

Line 177-179- Please clarify what are the three stages you are talking about.

Line 184- From my experience, these safety standards in official documents are expressed in another unit (g/L) than the one used in the paper. Maybe a correspondence table might interest some of the readers to understand the concentration levels discussed in this study.

Line 196-202- It should be admitted here this is a bias in the analysis and if discussed somewhere else or used in other publications, please cite.

Line 209- particles instead of particulates.

Line 209-213- very complicated sentence while it is understood since the beginning that there are four different treatments corresponding to 4 different concentrations.

Line 221- there is a discrepancy between Menzel et al. and the reference Yentsch and Mezel in the

reference list.

Line 270- Is there a pre-filtration applied to the collected seawater?

Line 287- what is the reference for these numbers?

Line 305-329- Nice experiment.

Line 335- Please add a reference to the SPSS17.0, is it a program, a pipeline, a software??

Line 354- "blown dry" ?? Really?

Line 362- In the supplementary Table S5 I have, most of these values (columns D/H/K/T/Y/Z/AB/AC) do not exist and I have a little warning sign. Maybe it is better to put the numbers and explain in the supplementary materials how these numbers have been obtained (e.g. equation).

Line 459- Fig. 2a, are you sure?

Line 494- What % are talking about? Reads? OTUs?

Line 500-501- First main result of your study, I would highlight it a little bit more.

Line 506-507- Second main result of your study, I would highlight it a little bit more.

Line 504- I guess you want to reference Fig. 4 instead of 3.

Line 509- Here and in the previous paragraph, it is not very clear if the results are from molecular or microscopic observations, please help the readers to follow the results.

Line 551-555- Third main result of your study, I would highlight it a little bit more.

Line 558- replace μ by "daily growth rates".

Line 563- Not sure about what you are mentioning to by "the production cycle" here and some other places in the ms...do you mean growth rate or cell cycle ?

Line 566-571- Does it refer to a figure? Table?

Line 579- Fig. S2b is absolutely not referring to diversity. Please, carefully proof-read the paper so there is no annoying mistakes as this one.

Line 579-582 - not sure the figure is the correct one, if yes to what protozoa correspond to? please specify. Not clear if we are talking about molecular or microscopic analyses here. Please, try as much as possible to improve the clarity of the results you present.

Line 697-699 - Not sure the authors present data from adjacent waters of Ukraine (as there is no data in the Black Sea), so not sure about what the authors are talking about?

Line 740- reference?

Line 776- reference?

Line 779- immeasurable or hard/complicated to measure?

Line 798- "larger specific areas": what does this mean? That tiny cells have a larger surface area to volume ratio than bigger ones...please, be specific and clear.

Line 800-803- why do you state here that molecular approach is not good enough to accurately describe species lower than genus level? Is it because of the approach you use of clustering reads

at 97%

Line 799 - Not sure about this statement, or not clear about what sizes you are comparing: with the reference you cite, are you talking about cells compared with macro-organisms? Or nano-plankton compared to pico-plankton? please be more specific about what you want to say here.

Line 848-850- Not sure this is true. The change in size of particle might modify the flux of matters from the main and traditional food chain to the microbial loop, but not sure what does "prolong the transmission process" and "reduce the transmission efficiency" mean? For example, heterotrophic protists are very efficient to bioremineralize matter from primary production. Please, clarify, be more specific or give more details and references about your ecological statements.

Figures and legends:

Overall, I think the figures are not very clear or easy to read. With figures with maps, I do not understand the units of the bar graphs that are plotted (e.g. figures 1 and 2). Font should be bigger, especially for figure S2a for example, where it is very hard to read the numbers that then should be connected (not easily) with herbicides. It should be easier for the readers to understand the figures. Figures 3 and 5, change the direction of the legend squares so it is easier for the readers to connect taxonomy and the bar graphs. Figure 6, I am colour-blind, so the choice of colors does not fit me at all. Figure 7, you mention "chronological order" but it is not explained anywhere in the main text. Figure 9 and Figure S5, and some others, the unit of the bar legends on the right are not explained. Figure S6, not sure about what panels are what letters?

Overall, if I think the subject is brave, the approach is original and the study is of good quality in term of research design and analyses, I am a little bit disappointed by the choice of references, clarity of results and development of the discussion. If publication is accepted it would need a deep revision in the writing so it can suit the editorial requirements of a Nature Communications paper.

Reviewer #3:

Remarks to the Author:

The scope and purpose of the paper is very commendable - to provide a global assessment of the risk posed by herbicides, although it is only the risk posed by 12 triazine herbicides. The authors have conducted a very thorough set of experiments to compliment the data acquisition and analyses that they have conducted.

I did not read the whole manuscript because I had so many questions and concerns about the methodology that I could not be confident that they are accurate and therefore it was not possible to review the results, discussion and conclusions. It is not necessary for the answers to some of my questions to be in the text of the manuscript but at the very least they need to be provided in the Supplementary Material section. I do not have expertise in genomics, so I cannot make any comment on the appropriateness of the information provided, nor the methods that were used. I do not believe that sufficient information is provided to permit readers to repeat the assessment or to use the methods for their own similar analysis.

The sources of pesticide concentration data for the 15 bays must be provided.

I recommend that the manuscript be rejected. If this had been submitted for a lesser quality journal I would have recommended very major revision.

I have written numerous comments on the manuscript which I have attached with this review. I would strongly encourage the authors to consider my comments and questions and to submit a revised manuscript to another journal - as I believe that it could be a valuable contribution to pollution science and environmental protection.

We are very grateful to the reviewers for their constructive comments and suggestions, which greatly helped us to improve the manuscript. During the revision, all the questions and suggestions raised by the reviewers were resolved, and a professional English editing service refined the language. The detailed point-by-point responses are as follows:

Reviewer 1

The authors have submitted a comprehensive work discussing the impact of herbicides on coastal environment. I have the following comments/question for the authors to address:

Q1 The authors narrowed their study to bay/gulf areas. However, the reason for that is unclear. I can understand that the authors would want to avoid truly offshore areas, but why not general coastal zones or national coastal zones? The current selection of gulfs/bays does limit the applicability of the study as these areas generally are less mixed and very dependent on the geology and hydrology of the bay and pollution may therefore be a worst-case scenario which is not applicable to a broader context. At the same time, because of this, care should be taken in overgeneralizing the results to all coastal waters.

Answer: Thank you very much for your valuable comments. In fact, we did try to obtain information on herbicide pollution in coastal waters worldwide. However, as far as we know, most of the published data are in gulf and bay regions, except for a few coastal regions (such as the Great Barrier Reef in Australia). We agree with the reviewer that the concentrations of herbicides in gulfs should be higher than those in coastal zones. In our own large-scale herbicide investigation in the coastal zones of the Bohai Sea and the Yellow Sea, we observed that the concentrations of triazine herbicides at the sampling stations were negatively correlated with their distances from the estuaries, showing a decreasing trend in concentration along the estuary-bay-coastal zone. However, even in the coastal zones more than 50 nautical miles offshore (S11, for example), the total concentration of 10 triazine herbicides was still as high as 1.99 nmol L^{-1} , which was equivalent to or even higher than that at some sites in the Bohai Bay area. These results indicate that even in coastal areas, herbicide pollution is quite serious.

According to the reviewer's comment, to avoid overgeneralizing the results to all coastal waters, we have added the corresponding discussion and clarification as follows:

P₂L₃₅-P₂L₃₇

Our field surveys revealed that the concentrations of triazine herbicide in coastal waters, although

relatively low, were also within this range.

P27L597-P28L607

As herbicides are considered a typical non-point-source agricultural pollutant, previous studies on the ecological risks of herbicides mainly focused on farmland and freshwater ecosystems. However, the global coastal pollution status of herbicides and their negative impact on marine life (especially phytoplankton) in natural environmental concentrations are poorly understood except for few special environments (e.g. the Great Barrier Reef^{61, 62}, Australia). Due to the hydrodynamic diffusion and self-purification effects of the oceans^{63, 64}, the in situ impact of herbicides on primary productivity in seawater may be lower than that estimated by microcosmic experiments. To clarify this issue, we conducted a large-scale field investigation of herbicide pollution in the Bohai Sea and the Yellow Sea, observing the spread of herbicides along the estuary, bay, and open sea (Supplementary Fig. S5). The results confirmed that the concentration of triazine herbicide at each sampling station was negatively correlated with its distance from the estuaries (Supplementary Fig. S5). However, all ten triazine herbicides were detected at nearly all the 64 stations. Even in sea areas more than 50 nautical miles offshore (S11, for example), the total concentration of 10 triazine herbicides was still as high as 1.99 nmol L⁻¹, which was equivalent to or higher than the concentrations at some sites in the Bohai Sea (Supplementary Fig. S5). These indicate that even in coastal areas, herbicide pollution is also quite serious.

References:

61. Kroon FJ, et al. River loads of suspended solids, nitrogen, phosphorus and herbicides delivered to the Great Barrier Reef lagoon. *Marine pollution bulletin* 65, 167-181 (2012).
62. Kroon FJ, Thorburn P, Schaffelke B, Whitten S. Towards protecting the Great Barrier Reef from land-based pollution. *Global change biology* 22, 1985-2002 (2016).

Supplementary Fig. S5. The distribution of the three typical herbicide classes (triazine, phenylurea, and amide herbicides) in the surface waters of the Bohai Sea and Yellow Sea of China in autumn 2017 (a-c) and spring 2018 (d-f). (a, d) Triazine herbicides; (b, e) phenylurea herbicides; (c, f) amide herbicides. The colors represent high (red/green/blue) and low (gray) concentrations of the three typical herbicide classes. The units are nmol L^{-1} .

Q2. Why was *P. tricorutum* used as to determine the dose response effect given? Is it that representative for bay/gulf areas globally? Yet, the dominant species in the microcosm was *C. tenuissimus*, why was this one not used for the dose reponse curves?

Answer: Indeed, *Phaeodactylum tricorutum* is one of the most cosmopolitan diatom species⁶⁶, especially in nutrient-rich coastal waters^{67,68}. It is also a model organism among diatoms and is widely used in research fields such as global climate change^{69, 70}, eutrophication^{71, 72}, and marine pollution^{73, 74}.

As the reviewers pointed out, *Chaetoceros*, *Thalassiosira*, and *Skeletonema* are also dominant diatom taxa. In fact, in our previous research, five dominant diatom species (*Chaetoceros muelleri* CCMP1316, *Phaeodactylum tricorutum* Pt-1, *Thalassiosira pseudonana* CCMP1335, *Nitzschia*

closterium, and *Skeletonema costatum*) were selected to analyze the sensitivity of different algal species to atrazine.

Fig. S4. Concentration–response curves showing the effects of atrazine on the five tested *Bacillariophyceae* (solid lines) and five tested *Dinophyceae* (dashed lines) after 72 hours. The numbers 1–10 represent the ten algal species in the following order: *Chaetoceros muelleri* CCMP1316, *Phaeodactylum tricorutum* Pt-1, *Thalassiosira pseudonana* CCMP1335, *Nitzschia closterium*, *Skeletonema costatum*, *Prorocentrum donghaiense* CCMA-129, *Gymnodinium* sp. CCMA-167, *Amphidinium carterae* CCMA-279, *Heterocapsa circularisquama* CCMA-128, and *Gyrodinium* sp.

Yang L, Mou S, Li H, et al. Terrestrial input of herbicides has significant impacts on phytoplankton and bacterioplankton communities in coastal waters[J]. Limnology and Oceanography, 2021, 66(11): 4028-4045.

Considering that *Chaetoceros* is the most sensitive species to atrazine, the degree of inhibition of

atrazine on coastal primary productivity may be overestimated if calculated based on the dose–response curve of *Chaetoceros*. In contrast, *P. tricornutum* is moderately sensitive to atrazine and was therefore used for the toxicity equivalent conversion of triazine herbicides in this study.

Based on the reviewer’s comment, we have supplemented the description of the reasons for using *P. tricornutum* to determine the concentration–response effects and added the corresponding references.

P₃₀L₆₆₁-P₃₀L₆₇₀

Selection of model organism: *Phaeodactylum tricornutum* was used in this study for the toxicity equivalent conversion of triazine herbicides for three reasons: 1) it is one of the best-known cosmopolitan diatom species⁶⁹, especially in nutrient-rich coastal ecosystems^{70, 71}; 2) it is a model organism among diatoms, widely used in research on global climate change^{72, 73}, eutrophication^{74, 75}, marine pollution^{76, 77}, etc.; 3) among five dominant diatom species (*Chaetoceros muelleri* CCMP1316, *Phaeodactylum tricornutum* Pt-1, *Thalassiosira pseudonana* CCMP1335, *Nitzschia closterium*, and *Skeletonema costatum*), it is moderately sensitive to atrazine¹⁰ and is therefore suitable for the calculation of the inhibition rate of atrazine on diatoms in offshore waters.

References

10. Yang L, Mou S, Li H, Zhang Z, Jiao N, Zhang Y. Terrestrial input of herbicides has significant impacts on phytoplankton and bacterioplankton communities in coastal waters. *Limnology and Oceanography* **66**, 4028-4045 (2021).
69. Gao K, *et al.* Rising CO₂ and increased light exposure synergistically reduce marine primary productivity. *Nature Climate Change* **2**, 519-523 (2012).
70. Malviya S, *et al.* Insights into global diatom distribution and diversity in the world’s ocean. *Proceedings of the National Academy of Sciences* **113**, E1516-E1525 (2016).
71. Buitenhuis ET, *et al.* MAREDAT: towards a world atlas of MARine Ecosystem DATA. *Earth Syst Sci Data* **5**, 227-239 (2013).
72. Buck JM, *et al.* Lhcx proteins provide photoprotection via thermal dissipation of absorbed light in the diatom *Phaeodactylum tricornutum*. *Nature Communications* **10**, 4167 (2019).
73. Baker KG, Geider RJ. Phytoplankton mortality in a changing thermal seascape. *Global Change Biology* **27**, 5253-5261 (2021).
74. You Y, *et al.* Trypsin is a coordinate regulator of N and P nutrients in marine phytoplankton. *Nature Communications* **13**, 4022 (2022).
75. Cáceres C, Spatharis S, Kaiserli E, Smeti E, Flowers H, Bonachela JA. Temporal phosphate

- gradients reveal diverse acclimation responses in phytoplankton phosphate uptake. *The ISME Journal* **13**, 2834-2845 (2019).
76. Huang R, *et al.* Physiological and molecular responses to ocean acidification among strains of a model diatom. *Limnology and Oceanography* **65**, 2926-2936 (2020).
77. Xie Z, *et al.* Organophosphate ester pollution in the oceans. *Nature Reviews Earth & Environment* **3**, 309-322 (2022).

Q3. For the section 2.2.3, I could not find the information on quality control and blanks for DNA sequencing? Where blank filters analyzed to account for potential contamination? Where the primers confirmed to work on all species that you expect to detect? Also, DNA is not always a good proxy for cell density and cell concentration.

Answer: We apologize for the ambiguous description. In the process of DNA extraction and PCR amplification of samples, blank quality control was carried out simultaneously. According to the reviewer's suggestion, we modified the relevant sentences and marked them in red in the revised manuscript as follows:

P₃₃L₇₂₆-P₃₃L₇₂₉

For DNA extraction, the phytoplankton in 1 L of each water sample was filtered onto a 0.22 μm pore-size polycarbonate membrane (47 mm diameter, Millipore, MA, USA) and stored at -80°C for analysis. Blank filters were treated in the same manner as the samples.

P₃₃L₇₄₀-P₃₄L₇₄₄

The PCR samples were sent to Shanghai Majorbio Bio-Pharm Technology Co., Ltd. (Shanghai, China) and sequenced (2×300) on an Illumina MiSeq platform. The DNA concentrations in parallel incubations of blank filters were below the detection limit, and no detectable amplification of 18S rRNA gene products was observed.

Information about the coverage and specificity of the primer set is provided as follows:

P₃₃L₇₃₃-P₃₃L₇₄₀

The V1-V3 hypervariable region of the phytoplankton 18S rRNA gene was amplified with the eukaryote-specific primers 18S-82F (5'-GAAACTGCGAATGGCTC-3') and Ek-516R (5'-ACCAGACTTGCCCTCC-3')⁸³. The coverage and specificity of this primer set for the major taxonomic groups of phytoplankton and micro-zooplankton were evaluated using SILVA TestPrime

1.0 with version 138.1 of the SILVA SSU Ref database⁸⁴, with no mismatches allowed. This primer set was previously shown to theoretically amplify the 18S rDNA region from all major taxonomic groups (Supplementary Fig. S7).

Supplementary Figure S7: Screenshot of the Taxonomy Browser showing TestPrime results for the universal primers 82F-516R.

Reference

84. Quast C, et al. The SILVA ribosomal RNA gene database project: improved data processing and web-based tools. *Nucleic Acids Research* 41, D590-D596 (2012).

We agree with the reviewer that DNA is not always a good proxy for cell density and cell concentration due to the differences in gene copy number and primer preference among phytoplankton. The results based on high-throughput sequencing reflect the relative proportions of different phytoplankton groups or the trend of changes in the community structure in the experimental group relative to the control group, rather than absolute quantification. To reflect the effect of herbicide stress on the phytoplankton community structure more accurately, we also

determined the size fractions of chlorophyll *a* to complement the high-throughput sequencing results (Section 2.2.2).

Q4. Given that DNA based are also used to determine effects on the microzooplankton, it is unclear how the authors accounted for bias in the sequencing in not picking up cryptic species or larval species above a certain DNA threshold and therefore incorrectly reporting this as a change in species density. Additionally, biomass determines DNA quantity. Larger animals, larger larvae will lead to more DNA without being necessarily more dominant in terms of species number. This is particularly relevant for the larvae stage of zooplankton where there can be large variations in growth rate and size across species taxa.

Answer: Thank you for the constructive comments. As noted, large differences in the growth rate and size of taxa across species do introduce some bias in DNA-sequencing-based results for microzooplankton communities, which is an inevitable challenge in the current metabarcoding approach. Even so, due to the advantages of time savings, high throughput, low cost, high sensitivity and low destructiveness of this method, it is increasingly used to determine the taxonomic composition of higher trophic levels. Here, to reduce the bias caused by this method, we also incorporated microscopic morphological observation, which can reflect the number of species of different sizes in different growth stages.

Based on the reviewer's comments, we have pointed out the potential bias of the DNA-based method in the revised manuscript as follows:

P₂₆L₅₆₁-P₂₆L₅₆₉

It should be noted that large differences in growth rate and taxon size across species do introduce some bias in DNA-sequencing-based results for micro-zooplankton communities, which is an inevitable challenge of the current metabarcoding approach. Even so, due to the advantages of time savings, high throughput, low cost, high sensitivity and minimal destructiveness of this method, it is increasingly being used to determine the taxonomic composition of higher trophic levels^{53, 54, 55}. Here, to reduce the bias caused by this method, we also included the method of microscopic morphological observation (Supplementary Table S6), which can reflect the number of species of different sizes at different growth stages.

References:

- 53. Pearman JK, Casas L, Merle T, Michell C, Irigoien X. Bacterial and protist community changes during a phytoplankton bloom. *Limnology and Oceanography* **61**, 198-213 (2016).
- 54. Djurhuus A, *et al.* Evaluation of marine zooplankton community structure through environmental DNA metabarcoding. *Limnology and Oceanography: Methods* **16**, 209-221 (2018).
- 55. Aylagas E, Borja Á, Irigoien X, Rodríguez-Ezpeleta N. Benchmarking DNA metabarcoding for biodiversity-based monitoring and assessment. *Frontiers in Marine Science* **3**, 96 (2016).

Q5 . The overall methodology on the concentration response curves and risk assessment approach of the paper is clearly detailed, executed according to current statistical methodology and risk assessment practices and well reported. The overall resolution of the figures in the pdf is low and should be improved prior to publication.

Answer: Thank you for the suggestion. All the figures have been improved in the revised manuscript accordingly.

Fig. 1. Geographical distribution of the total concentration (nM) of 12 triazine herbicides in typical bays worldwide. The box-scatter plots show the residual concentrations of herbicides. Each point in the scatter plots corresponds to a survey site, arranged from left to right in chronological order. The box chart on the left shows the first quartile, median and third quartile of 12 triazine herbicides at all sites in the corresponding sea area. The solid circle in each sea area represents the median value

of the total concentration of 12 the triazine herbicides at each station in that area, with a larger circle size representing a larger median value. **Source data are provided as a Source Data file.**

Fig. 2. Geographical distribution of equieffective concentrations of atrazine in typical bays worldwide. The box-scatter plots show the equieffective concentrations of atrazine. Each point in the scatter plot on the right corresponds to a survey site, arranged from left to right in chronological order. The box chart on the left shows the first quartile, median and third quartile of the equieffective concentrations of atrazine at all sites in the corresponding sea area. The solid circle in each sea area represents the median value of the equieffective concentrations of atrazine at each station in that area, with a larger circle size representing a larger median value. **Source data are provided as a Source Data file.**

Fig. 3. Comparison of phytoplankton community structure between the control and atrazine-treated groups at the phylum/class level. CK, 0.5, 5 and 50 nmol L⁻¹ represent the control and the treatment groups dosed with 0.5–50 nmol L⁻¹ of atrazine, respectively, on the 21st day (with three replicates). A total of 566,933,791 high-quality sequences (average length of 464) from 12 samples were obtained and clustered into 99 OTUs (97% cutoff). Phyla or classes that did not represent at least 1% of the sequences in at least one sample were regrouped as “Others.” Source data are provided as a Source Data file.

Fig. 4. Size-fractionated chlorophyll *a* concentrations of the phytoplankton communities in the control and atrazine-treated groups. Source data are provided as a Source Data file.

Fig. 5. Comparison of the micro-zooplankton community structure between the control and atrazine-treated groups at the phylum/class level. CK, 0.5, 5 and 50 nmol L⁻¹ represent the control and treatment groups dosed with 0.5–50 nmol L⁻¹ of atrazine, respectively, on the 21st day (with three replicates). A total of 566,933,791 high-quality sequences (average length of 464) from 12 samples

were obtained and clustered into 99 OTUs (97% cutoff). Phyla or classes that did not represent at least 1% of the sequences in at least one sample were regrouped as “Others.” Source data are provided as a Source Data file.

Fig. 6. Correlation patterns between phytoplankton and their associated micro-zooplankton at the genus level. Correlations were calculated for phytoplankton (*Bacillariophyta*, *Dinophyceae*, and *Cryptophyceae*) and zooplankton groups with an abundance $\geq 1\%$ in at least one sample. The groups marked unclassified in the high-throughput sequencing results were manually blast-searched against the NCBI nucleotide collection and EzBioCloud Database. Sequences with the same taxonomic assignment at the genus level were combined. The relative abundances of phytoplankton and micro-zooplankton OTUs in the control and atrazine-treated groups (0.5, 5, and 50 nmol L^{-1}) were employed to construct Bray–Curtis similarity matrices. The analyses were based on Spearman’s rank correlation coefficients. Source data are provided as a Source Data file.

Fig. 7. Inhibition levels of environmental concentrations of triazine herbicides on phytoplankton primary productivity in typical bays worldwide. The box-scatter plots show the inhibition levels. Each point in the scatter plot on the right corresponds to a survey site, which is arranged from left to right in chronological order. The box chart on the left shows the first quartile, median and third quartile of the inhibition level of triazine herbicides on phytoplankton primary productivity at all sites in the corresponding sea area. The solid circle in each sea area represents the median value of the inhibition level at each station in that area, with a larger circle size indicating a larger median value. Source data are provided as a Source Data file.

Fig. 8. Global map of herbicide risk scores (RSs). The map has a spatial resolution of 5 arcmin, which is approximately 10 km × 10 km at the equator. The pie charts represent the fraction of agricultural land classed under different RSs in each region, and the values in parentheses above the pie charts denote the total amount of agricultural land in those regions. Source data are provided as a Source Data file.

Fig. 9. Global map of herbicide risk scores in agricultural areas and 22 herbicide residues in 3 categories in adjacent sea areas. The map has a spatial resolution of 5 arcmin, which is

approximately 10 km × 10 km at the equator. The box-scatter plots show the residual concentration of herbicides. Each point in the scatter plots corresponds to a survey site, which is arranged from left to right in chronological order. The box chart on the left shows the first quartile, median and third quartile of 22 triazine herbicides at all sites in the corresponding sea area. The solid circle in each sea area represents the median value of the total concentration of 22 triazine herbicides at each station in that area, with a larger circle size representing a larger median value. Source data are provided as a Source Data file.

Reviewer 2

This publication from Yang et al. aims at predicting the impact of the most used herbicides on marine primary productivity at a global scale. The study is based on (i) a meta-analysis to collect concentrations of herbicides all around the world and (ii) some experimental work about the effect of herbicide concentrations on phytoplankton and microzooplankton growth and community composition. Then a combination of both approaches is used to predict the environmental effect at a global scale. Authors detected a sound effect of herbicide concentration (namely atrazine) on phytoplankton, from nano- to micro-sized community (mainly due to change in a few species dominance) and a strong negative effect on pico-plankton growth rate. There is also a measured effect on micro-zooplankton with a copepod-dominated community to a heterotrophic protist-based community (ciliates). Prediction on primary production inhibition under current herbicide stress is expected to be measurable for 25% of analyzed sites (at least a 5% inhibition).

The subject of the paper is fascinating: I would like to thank the authors for having dealt with this very interesting question that is sometimes complicated to deal with because of the food industry lobbies. I like this work a lot because the research plan is well thought and a lot of experiments and analyses have been done to decipher all the potential confounding effects. It is hard to measure and predict the effects of herbicides in the environment and the authors have been able to couple lab experiments and meta-analysis to look at the big picture, which makes this study pretty original. However, I think there is still a lot of work to improve the quality of the paper in term of writing to

make it clearer, more concise and stronger. The flaws of the paper are more in the interpretation (which is pretty short compared to the other sections) and referencing than in the data analysis and methodology that have been well done. The work is overall reproducible which is another good point.

Answer: We appreciate the reviewer's positive comments.

According to the reviewer's suggestion, the Introduction and Discussion sections have been completely rewritten, corresponding references have been corrected and supplemented, and a comparative analysis with existing studies has been added and marked in red in the revised manuscript. To avoid duplication, the relevant modification information is provided in the "Main concerns" section. Please review the revised text.

Main concerns:

(i) the introduction needs to be revised to improve the references and sometimes clarify the statements.

Answer: According to the reviewer's suggestion, the Introduction section has been revised, and appropriate references have been added. The main changes are listed below:

P₃L₅₄-P₃L₄₆

Millions of tonnes of herbicides are used annually worldwide ^{1,2}, 70% of which eventually enter the ocean through runoff ³,

P₄L₈₆-P₄L₉₈

The smaller pico-phytoplankton mainly enter the complex microbial loop through ingestion by heterotrophic protists ¹⁹, while the larger nano- and micro-phytoplankton mainly enter the classic food chain (phytoplankton->heterotrophic protists->zooplankton), with higher energy conversion efficiency ¹⁹. Therefore, the inhibitory effects of herbicides on high-abundance and sensitive groups (e.g., *P. tricornutum* and *Chaetoceros* sp.) may not only lead to a decrease in overall primary productivity but may increase the proportion of phytoplankton groups that are resistant to herbicides, which may trigger succession of the community structure and changes in the algal cell particle size composition, thus changing the growth rate and energy conversion efficiency. Therefore, exploring the effects of the suppression of sensitive algae on the abundance, community composition, and particle size of phytoplankton is expected to explain the effects of modern intensive agriculture on marine primary production.

P₅L₁₀₇-P₅L₁₁₂

To quantify the impact of current herbicide pollution on offshore primary productivity on a larger scale, we first collected survey data published from 1995 to 2022 on herbicide pollution in bays around the world. The temporal and spatial distribution patterns and background values of herbicides in the coastal waters of typical bay areas on all continents were determined. This approach aimed to help answer key scientific questions such as the scope of herbicide pollution in offshore waters and how large of an area of phytoplankton primary productivity is affected.

P₅L₁₂₀-P₆L₁₂₂

Chemical Survey and Risk Assessment of 56 Pesticides in the Sado River Estuary. There is an additive toxicity effect among herbicides with the same mode of action.

P₆L₁₃₉-P₆L₁₄₂

Agricultural production is the main source of herbicides in coastal waters, and the types and dosages of herbicides used on farmlands largely determine the herbicide pollution status in the adjacent seas 22, 25. At present, there are more than 1,500 herbicides on the market worldwide, with more than 300 active ingredients^{26}.

References

1. Sharma A, et al. Worldwide pesticide usage and its impacts on ecosystem. *SN Applied Sciences* 1, 1446 (2019).
2. Varah A, et al. The costs of human-induced evolution in an agricultural system. *Nature Sustainability* 3, 63-71 (2020).
3. Agrawal A, Pandey RS, Sharma B. Water Pollution with Special Reference to Pesticide Contamination in India. *Journal of Water Resource and Protection* Vol.02No.05, 17 (2010).
19. Worden AZ, Follows MJ, Giovannoni SJ, Wilken S, Zimmerman AE, Keeling PJ. Rethinking the marine carbon cycle: Factoring in the multifarious lifestyles of microbes. *Science* 347, 1257594 (2015).
22. Tyohemba RL, Pillay L, Humphries MS. Herbicide residues in sediments from Lake St Lucia (iSimangaliso World Heritage Site, South Africa) and its catchment areas: Occurrence and ecological risk assessment. *Environmental Pollution* 267, 115566 (2020).
25. Sun B, Zhang L, Yang L, Zhang F, Norse D, Zhu Z. Agricultural non-point source pollution in China: causes and mitigation measures. *Ambio* 41, 370-379 (2012).

26. Heap I. *Herbicide resistant weeds*. Springer (2014).

(ii) the discussion section length is weirdly balanced compared to the results section, the latter being way longer. This section also needs to be improved with more discussion related to the existing literature.

Answer: According to the reviewer's suggestion, the Discussion section has been improved, and comparative analysis with existing studies has been added. The main changes are as follows:

P₁₉L₄₂₇-P₂₀L₄₃₂

Agricultural practices determine levels of food production and largely determine the state of the global environment^{31, 32}. In recent years, water pollution caused by herbicides has been spreading rapidly from fresh water to seawater, due to the increasing food demand caused by population growth and the acceleration of global urbanization^{33, 34, 35}. Various herbicide residues have been detected frequently in coastal waters worldwide.

P₂₂L₄₈₃-P₂₂L₄₈₈

Although the coastal ocean accounts for only 7–8% of the ocean area, it contributes more than 25–28% of the global ocean primary productivity⁴². If herbicide pollution causes a 5% drop in coastal primary productivity, the annual carbon fixation amounts of phytoplankton will decrease by 3.75–8.75 × 10⁸ tons, which is equivalent to the carbon fixation by the Amazon rainforest.

P₂₃L₅₀₆-P₂₄L₅₁₉

Diatoms are the most dominant phytoplankton in coastal oceans, contributing up to 40% of the total marine primary production. Moreover, it is well known that the production originating from larger phytoplankton, such as diatoms, is the portion most efficiently transferred to higher levels of the food web⁴⁵. The inhibitory effect of herbicides on diatoms and the changes in the phytoplankton community will inevitably have a profound impact on the higher trophic levels of natural ecosystems. Changes in the community structure inevitably lead to changes in the particle size structure. Tiny cells have a higher surface area to volume ratio than larger cells^{46, 47}. Thus, for the same biomass, nano-phytoplankton contributes more to primary productivity than micro-phytoplankton⁴⁸. However, it is still unclear how herbicides change primary productivity by changing the particle size of phytoplankton. As the reads from high-throughput sequencing techniques are relatively short, it is difficult to obtain information such as the full-length 18S/ITS

rDNA sequences in phytoplankton, and these techniques cannot provide particle size information at the species level.

P₂₅L₅₅₅-P₂₆L₅₆₁

This result is unexpected considering the results of a single species-based micro-zooplankton toxicity test. Herbicides mainly target the photosynthesis system, which does not exist in micro-zooplankton, and therefore are generally considered to have low toxicity to micro-zooplankton. According to existing literature reports, the herbicide concentration needs to reach the milligram level to have a significant impact on some micro-zooplankton, under pure culture conditions 51, 52.

P₂₆L₅₈₁-P₂₈L₆₀₇

Marine organisms of different particle sizes and different nutrient levels form a complex network structure through predatory relationships, and changes in any one population may affect the entire offshore ecosystem 58. The change in particle size might modify the flux of materials from the main and traditional food chain to the microbial loop, which might prolong the transmission process of the marine food chain and reduce the transmission efficiency of primary productivity, since the microbial loop was the most inefficient among multiple routes for algal primary production to transfer to higher trophic levels in terms of respiratory carbon losses 19, 59, 60. As herbicides are considered a typical non-point-source agricultural pollutant, previous studies on the ecological risks of herbicides mainly focused on farmland and freshwater ecosystems. However, the global coastal pollution status of herbicides and their negative impact on marine life (especially phytoplankton) in natural environmental concentrations are poorly understood except for few special environments (e.g. the Great Barrier Reef 61, 62, Australia). Due to the hydrodynamic diffusion and self-purification effects of the oceans 63, 64, the in situ impact of herbicides on primary productivity in seawater may be lower than that estimated by microcosmic experiments. To clarify this issue, we conducted a large-scale field investigation of herbicide pollution in the Bohai Sea and the Yellow Sea, observing the spread of herbicides along the estuary, bay, and open sea (Supplementary Fig. S5). The results confirmed that the concentration of triazine herbicide at each sampling station was negatively correlated with its distance from the estuaries (Supplementary Fig. S5). However, all ten triazine herbicides were detected at nearly all the 64 stations. Even in sea areas more than 50 nautical miles offshore (S11, for example), the total concentration of 10 triazine herbicides was still as high as 1.99 nmol L⁻¹, which was equivalent to or higher than the concentrations at some sites in the Bohai Sea

(Supplementary Fig. S5). These indicate that even in coastal areas, herbicide pollution is also quite serious.

References:

19. Worden AZ, Follows MJ, Giovannoni SJ, Wilken S, Zimmerman AE, Keeling PJ. Rethinking the marine carbon cycle: Factoring in the multifarious lifestyles of microbes. *Science* **347**, 1257594 (2015).
31. Tilman D, Cassman KG, Matson PA, Naylor R, Polasky S. Agricultural sustainability and intensive production practices. *Nature* **418**, 671-677 (2002).
32. Xu R, *et al.* Global ammonia emissions from synthetic nitrogen fertilizer applications in agricultural systems: Empirical and process - based estimates and uncertainty. *Global change biology* **25**, 314-326 (2019).
33. Ouyang W, *et al.* Occurrence, transportation, and distribution difference of typical herbicides from estuary to bay. *Environment international* **130**, 104858 (2019).
34. Gallen C, *et al.* Integrated chemical exposure assessment of coastal green turtle foraging grounds on the Great Barrier Reef. *Science of the Total Environment* **657**, 401-409 (2019).
35. Topaz T, Egozi R, Eshel G, Chefetz B. Pesticide load dynamics during stormwater flow events in Mediterranean coastal streams: Alexander stream case study. *Science of the Total Environment* **625**, 168-177 (2018).
45. Calbet A, Landry MR. Phytoplankton growth, microzooplankton grazing, and carbon cycling in marine systems. *Limnology and Oceanography* **49**, 51-57 (2004).
46. Verity PG, Robertson CY, Tronzo CR, Andrews MG, Nelson JR, Sieracki ME. Relationships between cell volume and the carbon and nitrogen content of marine photosynthetic nanoplankton. *Limnology and Oceanography* **37**, 1434-1446 (1992).
47. Rodriguez J, *et al.* Mesoscale vertical motion and the size structure of phytoplankton in the ocean. *Nature* **410**, 360-363 (2001).
48. Chen B, Liu H. Relationships between phytoplankton growth and cell size in surface oceans: Interactive effects of temperature, nutrients, and grazing. *Limnology and Oceanography* **55**, 965-972 (2010).
51. Chen CY, Hathaway KM, Folt CL. Multiple stress effects of Vision® herbicide, pH, and food on zooplankton and larval amphibian species from forest wetlands. *Environmental Toxicology and*

Chemistry: An International Journal **23**, 823-831 (2004).

52. Filimonova V, Goncalves F, Marques JC, De Troch M, Goncalves AM. Biochemical and toxicological effects of organic (herbicide Primextra® Gold TZ) and inorganic (copper) compounds on zooplankton and phytoplankton species. *Aquatic Toxicology* **177**, 33-43 (2016).

58. Smriga S, Fernandez VI, Mitchell JG, Stocker R. Chemotaxis toward phytoplankton drives organic matter partitioning among marine bacteria. *Proceedings of the National Academy of Sciences* **113**, 1576-1581 (2016).

59. Sswat M, *et al.* Food web changes under ocean acidification promote herring larvae survival. *Nature Ecology & Evolution* **2**, 836-840 (2018).

60. Berglund J, Müren U, Båmstedt U, Andersson A. Efficiency of a phytoplankton-based and a bacterial-based food web in a pelagic marine system. *Limnology and Oceanography* **52**, 121-131 (2007).

61. Kroon FJ, *et al.* River loads of suspended solids, nitrogen, phosphorus and herbicides delivered to the Great Barrier Reef lagoon. *Marine pollution bulletin* **65**, 167-181 (2012).

62. Kroon FJ, Thorburn P, Schaffelke B, Whitten S. Towards protecting the Great Barrier Reef from land-based pollution. *Global change biology* **22**, 1985-2002 (2016).

(iii) I do not think the figures in the current version are of good quality in term of clarity and information conveyed.

Answer: Thank you for the comments. All the figures have been improved in the revised manuscript as indicated above (Reviewer 1, Q5).

Minor comments:

Q1. Line 55- the reference Yang et al. (2019) might not be the correct one to cite here.

Answer: We apologize for this mistake. In the revised MS, the new references are provided as follows:

P₃L₅₄-P₃L₅₅

Millions of tonnes of herbicides are used annually worldwide ^{1,2}, 70% of which eventually enter the ocean through runoff ³,

References:

1. Sharma A, et al. Worldwide pesticide usage and its impacts on ecosystem. *SN Applied Sciences* 1, 1446 (2019).

(Presently, throughout the globe, approximately 2 million tonnes of pesticides are utilized, of which 47.5% are herbicides, 29.5% are insecticides, 17.5% are fungicides and 5.5% are other pesticides...by the year 2020, the global pesticide usage has been estimated to increase up to 3.5 million tonnes.)

2. Varah A, et al. The costs of human-induced evolution in an agricultural system. *Nature Sustainability* 3, 63-71 (2020).

(...an estimated 4 million tonnes of pesticides are applied worldwide each year.)

3. Agrawal A, Pandey RS, Sharma B. Water Pollution with Special Reference to Pesticide Contamination in India. *Journal of Water Resource and Protection* Vol.02No.05, 17 (2010).

(The major part of the pesticides applied in any area for a specific reason (about 99%) remain unused and it gets mixed with air, soil, water and plants which by several means causes harmful effects on the people, pets, and the environment.)

Q2. Line 67- the reference Suttle et al. is absolutely not referring to the fact that half of the photosynthesis comes from phytoplankton. Field et al. 1998 is a better choice here. Be careful, this kind of bad referencing could be considered as a red flag.

Answer: We apologize for this carelessness and thank the reviewer for the guidance. In the revised manuscript, the inappropriate reference has been replaced by the reference recommended by the reviewer.

P₃L₆₆-P₃L₆₈

Marine phytoplankton contribute approximately 50% of the total global primary productivity and play a vital role in global carbon cycling ¹¹.

References:

11. Field CB, Behrenfeld MJ, Randerson JT, Falkowski P. Primary Production of the Biosphere: Integrating Terrestrial and Oceanic Components. *Science* **281**, 237-240 (1998).

Q3. Line 76- “in situ in seawater” sounds weird, keep in situ or in seawater

Answer: Thank you for the suggestion. The sentence has been rewritten accordingly.

P₃L₇₆-P₄L₇₇

However, it is difficult to truly understand the ecological effects of herbicides in situ from the individual or population response level of algae alone,

Q4. Line 78- here and along the manuscript, be sure that you talk about primary productivity and not primary production.

Answer: We carefully checked the descriptions involving primary productivity in the full text and made the following amendments to some ambiguous expressions.

P₄L₉₃-P₄L₉₈

which may trigger succession of the community structure and changes in the algal cell particle size composition, thus changing the growth rate and energy conversion efficiency. Therefore, exploring the effects of the suppression of sensitive algae on the abundance, community composition, and particle size of phytoplankton is expected to explain the effects of modern intensive agriculture on marine primary production.

P₃₄L₇₆₂-P₃₄L₇₆₃

2.2.4 Potential effects of environmental concentrations of triazine herbicides on the growth rate and energy flow transfer of phytoplankton

P₁₃L₂₉₆-P₁₄L₂₉₇

3.3 Secondary effects of environmental concentrations of triazine herbicides due to structural changes in phytoplankton community

Q5. Line 81- Once again, not sure reference 7 is suited here.

Answer: In the revised manuscript, the inappropriate reference has been replaced by the new reference as follows:

P₄L₇₉-P₄L₈₂

Marine primary productivity is the aggregate of various phytoplankton groups, and the community structure and particle size of phytoplankton are two key indicators of primary productivity^{15, 16}.

References:

16. Mouw CB, Yoder JA. Primary production calculations in the Mid-Atlantic Bight, including

effects of phytoplankton community size structure. *Limnology and Oceanography* 50, 1232-1243 (2005).

(...a change in community structure can greatly influence the overall primary production; This is evident from the simulated changes in the phytoplankton size composition. Changing the cell size spatial and temporal composition from full ecological variability to all picoplankton cells increased the overall production by approximately 16%, while changing the composition to all microplankton cells reduced the overall production by approximately 70%.)

Q6. Line 87- Protozoa is kind of an old term, maybe better to use the term “heterotrophic protists”

Answer: Thank you for the suggestion. The relevant phrases have been modified accordingly throughout the text.

P₄L₈₆-P₄L₈₇

The smaller pico-phytoplankton mainly enter the complex microbial loop through ingestion by **heterotrophic protists**,

P₁₄L₃₀₇-P₁₄L₃₀₇

...while the proportion of **heterotrophic protists** was only 3%.

P₁₄L₃₁₀-P₁₄L₃₁₁

The relative abundance of **heterotrophic protists** significantly increased to 10%

P₁₄L₃₁₄-P₁₄L₃₁₅

and the proportion of **heterotrophic protists** (*Holosticha diademata*) significantly increased to 40.0%

Q7. Line 88- Please, specify what classic food chain means (I guess this is the classic phytoplankton->heterotrophic protists->zooplankton). And maybe cite also a review a more classical paper to reference this ecological statement (e.g., Worden et al. 2015 Science)

Answer: We appreciate the reviewer’s valuable advice. The description of the classic food chain and appropriate references have been added in the revised manuscript.

P₄L₈₇-P₄L₈₉

while the larger nano- and micro-phytoplankton mainly enter the classic food chain (phytoplankton->heterotrophic protists->zooplankton), with higher energy conversion efficiency¹⁹.

References:

19. Worden AZ, Follows MJ, Giovannoni SJ, Wilken S, Zimmerman AE, Keeling PJ. Rethinking the marine carbon cycle: Factoring in the multifarious lifestyles of microbes. *Science* 347, 1257594 (2015).

Q8. Line 95-99- Please, re-phrase the sentence as this is not proper English. Maybe something like: ‘Therefore, exploring the response of phytoplankton to environmental stress caused by herbicides using sensitive species as case studies is expected to explain the effects of modern intensive agriculture on marine primary production.’

Line 110- ‘in global bays’ is not proper English. Discard the word globally at the end of the sentence, it is unnecessary.

Answer: We apologize for these inappropriate descriptions. These sentences have been rephrased in the revised MS and marked in red as follows:

P₄L₉₅-P₄L₉₈

Therefore, exploring the effects of the suppression of sensitive algae on the abundance, community composition, and particle size of phytoplankton is expected to explain the effects of modern intensive agriculture on marine primary production.

P₅L₁₀₈-P₅L₁₁₂

we first collected survey data published from 1995 to 2022 on herbicide pollution in bays around the world. The temporal and spatial distribution patterns and background values of herbicides in the coastal waters of typical bay areas on all continents were determined.

Q9. Line 159- quotation from what reference??

Answer: We apologize for our ambiguous description. The phrase “land–sea coordination” is an expression of China’s marine ecological civilization construction. No suitable reference was found.

To prevent ambiguity, this sentence has been rephrased in the revised MS as follows:

P₇L₁₅₇-P₇L₁₅₉

This work will provide suggestions for reducing emissions, preventing herbicide overuse on the premise of ensuring food security, and ensuring coastal ecological health.

Q10. Line 177-179- Please clarify what are the three stages you are talking about.

Answer: Thank you for the suggestion.

In addition, from 1990 to 2022, the agricultural consumption of herbicides worldwide showed an overall increasing trend: the level was relatively flat from 1990 to 2000, increased rapidly from 2001 to 2012, and then stabilized (FAOSTAT (Jun 16, 2023) <https://www.fao.org/faostat/en/#data/RP/visualize>). The survey data of each sea area were roughly divided into three chronological stages (1990-2000, 2001-2011, and 2012-2022) to analyze the trend of the changes in herbicide residue concentration in each sea area over time.

Agricultural consumption of herbicides worldwide from 1990 to 2020 (in million metric tons)

FAOSTAT (Jun 16, 2023)

According to the reviewer's comment, we added the description of the three stages in the revised MS.

P₂₉L₆₄₁-P₂₉L₆₄₆

In addition, from 1990 to 2022, the agricultural consumption of herbicides worldwide showed an overall increasing trend: the level was relatively flat from 1990 to 2000, increased rapidly from 2001 to 2012, and then stabilized (FAOSTAT (Jun 16, 2023) <https://www.fao.org/faostat/en/#data/RP/visualize>). The survey data of each sea area were roughly divided into three chronological stages (1990-2000, 2001-2011, and 2012-2022) to analyze the

changes in herbicide residue concentration in each sea area over time.

Q11. Line 184- From my experience, these safety standards in official documents are expressed in another unit (g/L) than the one used in the paper. Maybe a correspondence table might interest some of the readers to understand the concentration levels discussed in this study.

Answer: Thank you for the comments. To facilitate the equivalent conversion of herbicide congeners, the concentration unit used in this article is moles, which may not be convenient for readers to compare with the safety standards in official documents. According to the reviewer's suggestion, a list of concentration data expressed in units of ng/L has been added in Source data. Fig. 1.

The quality standards are also expressed in units of ng/L as follows:

P₃₀L₆₅₁-P₃₀L₆₅₃

with the highest determined concentration (several times higher than the water quality safety standard ⁶⁸, which sets a maximum allowable concentration of 0.1 µg L⁻¹ for any single pesticide and 0.5 µg L⁻¹ for the total pesticide concentration

References:

68. Jess S, Kildea S, Moody A, Rennick G, Murchie AK, Cooke LR. European Union policy on pesticides: implications for agriculture in Ireland. *Pest Management Science* **70**, 1646-1654 (2014).

Q12. Line 196-202- It should be admitted here this is a bias in the analysis and if discussed somewhere else or used in other publications, please cite.

Answer: Thank you for the suggestion. The references have been supplied, and the phrasing has been revised as follows:

P₃₁L₆₈₄-P₃₁L₆₉₁

Although this toxicity equivalent conversion based on the concentration addition model⁷⁸ has been widely used in the study of various pollutants ^{79, 80, 81} and makes it possible to assess the combined toxicity of multiple herbicides in natural seawater, it is also necessary to admit that it may introduce some bias due to the varying susceptibility of different species to herbicides. To minimize this effect, we selected the most representative species possible for the single-substance concentration–response experiments, as mentioned above.

References:

78. Drescher K, Boedeker W. Assessment of the Combined Effects of Substances: The Relationship between Concentration Addition and Independent Action. *Biometrics* 51, 716-730 (1995).
79. Silva E, Rajapakse N, Kortenkamp A. Something from “Nothing” – Eight Weak Estrogenic Chemicals Combined at Concentrations below NOECs Produce Significant Mixture Effects. *Environmental Science & Technology* 36, 1751-1756 (2002).
80. Belden JB, Gilliom RJ, Lydy MJ. How well can we predict the toxicity of pesticide mixtures to aquatic life? *Integrated Environmental Assessment and Management* 3, 364-372 (2007).
81. Faust M, et al. Predicting the joint algal toxicity of multi-component s-triazine mixtures at low-effect concentrations of individual toxicants. *Aquatic Toxicology* 56, 13-32 (2001).

Q13. Line 209- particles instead of particulates.

Answer: We apologize for this mistake. It has been corrected accordingly.

P₄₂L₉₂₂-P₄₂L₉₂₃

and filtered through a 200 µm-mesh net to remove large particles.

Q14. Line 209-213- very complicated sentence while it is understood since the beginning that there are four different treatments corresponding to 4 different concentrations.

Answer: According to the reviewer’s suggestion, the redundant descriptions have been removed as follows:

P₃₂L₆₉₈-P₃₂L₇₀₁

In the laboratory, four atrazine doses (0, 0.5, 5 and 50 nmol L⁻¹) were applied to establish the following treatments: 1) control check (CK), consisting of uncontaminated seawater; 2) 0.5 nmol L⁻¹ atrazine; 3) 5 nmol L⁻¹ atrazine; and 4) 50 nmol L⁻¹ atrazine.

Q15. Line 221- there is a discrepancy between Menzel et al. and the reference Yentsch and Mezel in the reference list.

Answer: We apologize for mistakenly spelling Mezel as Menzel. The error has been corrected.

P₃₂L₇₁₀-P₃₂L₇₁₁

Chlorophyll a was extracted and measured with the method described by Yentsch and Mezel

et al ⁷⁹.

Q16. Line 270- Is there a pre-filtration applied to the collected seawater?

Answer: Yes, the water samples mentioned here were collected from each treatment group mentioned above (2.1.1) on the fourth day. Therefore, they were prefiltered through a 200 µm mesh net during the collection phase.

Q17. Line 287- what is the reference for these numbers?

Answer: According to the reviewer's suggestion, the reference has been added as follows:

P₁₄L₃₀₀-P₁₄L₃₀₁

A total of 45–78% of the fixed carbon of phytoplankton enters the food chain through ingestion by zooplankton, especially micro-zooplankton ³⁰.

Reference:

30. Huang B, et al. Phytoplankton growth and microzooplankton grazing in a subtropical coastal upwelling system in the Taiwan Strait. *Continental Shelf Research* 31, S48-S56 (2011).

Q18. Line 305-329- Nice experiment.

Answer: We appreciate the positive comment.

Q19. Line 335- Please add a reference to the SPSS17.0, is it a program, a pipeline, a software??

Answer: Yes, SPSS (Statistical Product and Service Solutions) is a statistical software program.

According to the reviewer's suggestion, the reference has been added as follows:

P₃₇L₈₂₆-P₃₇L₈₂₈

The 96 h LC50 value of atrazine for each zooplankton species was obtained by performing a probit regression analysis and a chi squared goodness of fit test on the experimental data with SPSS17.0

⁹³.

Reference:

93. Duncan C. *Quantitative Data Analysis with IBM SPSS 17, 18 & 19. Quantitative Data Analysis with IBM SPSS 17, 18 & 19: A Guide for Social Scientists* (2011).

Q20. Line 354- ‘blown dry’ ?? Really?

Answer: We apologize for this ambiguous description. To eliminate potential interference from methanol in the experimental results, the methanol (1 mL) in each flask was blown dry with N₂ in a fume hood before seawater was added. In the revised MS, we rephrased this sentence for clarity as follows:

P₃₈L₈₄₉-P₃₈L₈₅₁

To prevent methanol from interfering with the experimental results, the methanol in each flask was blown dry with N₂ in a fume hood, and then one liter of the clean seawater mentioned above was added

Q21. Line 362- In the supplementary Table S5 I have, most of these values (columns D/H/K/T/Y/Z/AB/AC) do not exist and I have a little warning sign. Maybe it is better to put the numbers and explain in the supplementary materials how these numbers have been obtained (e.g. equation).

Answer: Thank you for the valuable suggestion. Table S5 has been revised accordingly (Source data Figs. 1, 2, 7, and 9.), and the concentration-response functions for each individual triazine herbicide are provided in Table S2.

Supplementary Table S2. The concentration–response functions for each individual triazine herbicide.

	EC50	Model	A1	A2	X0	p	R ² adj
Atrazine	141.3 ± 12.5	L	1.7142	104.2656	0.1349	0.9173	0.989
DEA	580.3 ± 21.8	L	1.5695	101.0427	0.6049	1.1381	0.995
Propazine	147.1 ± 15.2	L	4.3068	103.3116	0.1584	1.1995	0.989
Simazine	28.7 ± 1.6	L	0.3833	103.3089	0.0309	0.8911	0.996
Terbutryn	8.5 ± 1.1	L	2.1441	101.7772	0.0091	1.1025	0.997
Ametryn	25.1 ± 1.2	L	-0.5755	102.6998	0.0262	0.9461	0.996

Dipropetryn	37.0 ± 3.1	L	-1.0603	101.592	0.0375	0.8534	0.993
Cyanazine	280.3 ± 11.7	L	1.1414	109.9918	0.36235	0.66913	0.995
Cybutryne	4.3 ± 0.3	L	0.678	100.21925	0.00445	1.08514	0.999
	EC50	Model	A1	A2	d	k	R ² adj
DIA	732.1 ± 36.5	W	4.0148	99.4471	0.8182	0.8223	0.989
Prometryn	849.1 ± 21.7	W	1.9596	99.3406	1.0341	0.8109	0.997
Prometon	76.2 ± 3.3	W	5.80E-46	96.4563	0.643	0.2094	0.996
Atrazine		L	3.1467	101.122	0.103	1.0778	0.995

Q22. Line 459- Fig. 2a, are you sure?

Answer: We apologize for this mistake! It should be Fig. S2a. We have corrected the error in the revised MS.

P₈L₁₈₆-P₉L₁₈₉

The concentrations of atrazine that had an equivalent toxic effect to that of the in situ concentrations (equieffective concentrations) of all 12 detected triazine residues were calculated to be 0-47.58 nmol L⁻¹ (Supplementary Fig. S1, Supplementary Table S2),

Q23. Line 494- What % are talking about? Reads? OTUs?

Answer: We apologize for this unclear description. The relative proportions were calculated based on the count data of the corresponding OTU for each species. The phrase has been revised as follows:

P₁₀L₂₁₀-P₁₀L₂₁₁

with relative proportions (OTU count data) of 77.4% and 17.8%, respectively (Table S6).

Q24. Line 500-501- First main result of your study, I would highlight it a little bit more.

Answer: Thank you for the valuable comment. We added the description of community succession

in the revised MS as follows:

P₁₀L₂₁₇-P₁₀L₂₂₈

At the same time, *Dinophyceae* increased from 17.8% to 47.5%, 53.6% and 79.2%, respectively, and the phytoplankton community changed from *Bacillariophyta*-dominated to *Dinophyceae*-dominated (Supplementary Table S3). At genus or species level (Supplementary Table S4), the genus *Chaetoceros*, which was the most common representative of the *Bacillariophyta*, suffered the most significant inhibitory effects. Under the stress of low, medium and high doses of atrazine, the proportion of *Chaetoceros* decreased from 73.0% to 5.2%, 2.9% and 0.3%, respectively. Moreover, most members of the *Dinophyceae* (*Gyrodinium jinhaense*, *Adenoides eludens*, *Ankistrodinium semilunatum* and *Euduboscquella* sp.) were more resistant to atrazine stress. The relative abundances of *Gyrodinium jinhaense* significantly increased from 9.8% to 17.2%, 14.0% and 60.5%, respectively, under exposure to low, medium and high doses of atrazine.

Q24. Line 506-507- Second main result of your study, I would highlight it a little bit more.

Answer: Thank you for the valuable comment. We added the description of the particle size of the phytoplankton community in the revised MS as follows:

P₁₀L₂₂₉-P₁₁L₂₄₁

In terms of particle size, based on high-throughput sequencing data, the control group was dominated by nano-phytoplankton (*Chaetoceros tenuissimus* and *Goniomonas avonlea*, $\geq 73.1\%$), while the low-dose atrazine treatment group was dominated by micro-phytoplankton (*Gyrodinium jinhaense*, *Adenoides eludens*, *Euduboscquella* sp. JMC-2019a, and *Ankistrodinium semilunatum*, 31.9%) and pico-phytoplankton (*Chrysochromulina rotalis* and *Chrysochromulina leadbeateri*, 27.3%) (Fig. 3, Supplementary Table S4). The dominance of micro-phytoplankton (*Gyrodinium jinhaense*, *Euduboscquella* sp. JMC-2019a, *Woloszynskia halophila* and *Fibrocapsa japonica*, 69.4%) was more obvious in the treatment group that received a high dose of atrazine. The phytoplankton community changed from nano-phytoplankton-dominated to micro-phytoplankton-

dominated. Even within the class *Bacillariophyta*, the dominant taxa showed a transition from small-sized *Chaetoceros* (from 73% to 1%) to larger-sized *Thalassiosira* (from 2.7% to 8.5%) sp. (Supplementary Table S4).

Q25. Line 504- I guess you want to reference Fig. 4 instead of 3.

Answer: We apologize for this mistake! It should be Fig. 4. We have corrected it in the revised MS accordingly.

P₁₁L₂₃₁-P₁₁L₂₃₅

while the low-dose atrazine treatment group was dominated by micro-phytoplankton (*Gyrodinium jinhaense*, *Adenoides eludens*, *Euduboscquella* sp. JMC-2019a, and *Ankistrodinium semilunatum*, 31.9%) and pico-phytoplankton (*Chrysochromulina rotalis* and *Chrysochromulina leadbeateri*, 27.3%) (Fig. 3, Supplementary Table S4).

Q26. Line 509- Here and in the previous paragraph, it is not very clear if the results are from molecular or microscopic observations, please help the readers to follow the results.

Answer: We apologize for our unclear description and the resulting misunderstanding. In fact, morphological identification was only used to analyze the community structure of micro-zooplankton, and the community structure analysis of phytoplankton only used high-throughput sequencing. According to the reviewer's suggestion, we rephrased the sentence as follows:

P₁₀L₂₂₉-P₁₀L₂₃₁

In terms of particle size, based on high-throughput sequencing data, the control group was dominated by nano-phytoplankton (*Chaetoceros tenuissimus* and *Goniomonas Avonlea*, $\geq 73.1\%$),

Q27. Line 551-555- Third main result of your study, I would highlight it a little bit more.

Answer: Thank you for the valuable comments. We added the description of the intrinsic growth rate of the phytoplankton community in the revised MS as follows:

P₁₂L₂₇₁-P₁₃L₂₈₀

The daily growth rates of micro-, nano- and pico-phytoplankton in the control group were 0.65 d⁻¹, 0.74 d⁻¹ and 1.14 d⁻¹, respectively, which indicated that the larger the size fraction was, the slower the growth rate. Atrazine significantly reduced the intrinsic growth rate (μ) of phytoplankton

(Supplementary Table S5). Especially in the groups treated with intermediate and high doses of atrazine, the intrinsic growth rates of micro, nano- and pico-phytoplankton decreased by 18.5% and 32.3%, 36.5% and 52.7%, and 14.9% and 71.9%, respectively, which meant that atrazine slowed the growth rate of phytoplankton. Compared with micro- and pico-phytoplankton, the intrinsic growth rate of nano-phytoplankton was more inhibited under low, medium and high atrazine stress (Supplementary Table S5).

Indeed, for acute toxicological experiments, an inhibition rate of 13% is not very high. However, the inhibition rate obtained in this experiment was based on a field investigation, which indicated that the pollution of herbicides in offshore waters was very serious. As the reviewer mentioned, this conclusion was made based on the assumption of additive concentration. To make the manuscript more rigorous, we deleted conclusions such as “our results revealed a serious situation of herbicide pollutions in coastal ocean and uncovered their destructive role and mechanism to marine phytoplankton cells, both of which were significantly underestimated in the past”. Moreover, we pointed out that “herbicide pollution might represent a neglected global-change threat to marine life and deserves more attention” based on our results.

Q28. Line 558- replace μ by ‘daily growth rates’.

Answer: Thanks for the reviewer’s suggestion. This has been corrected in the revised MS as follows:

P₁₂L₂₇₁-P₁₂L₂₇₂

The daily growth rates of micro, nano- and pico-phytoplankton in the control group were 0.65 d⁻¹, 0.74 d⁻¹ and 1.14 d⁻¹,

Q29. Line 563- Not sure about what you are mentioning to by ‘the production cycle’ here and some other places in the ms...do you mean growth rate or cell cycle?

Answer: We apologize for this ambiguous description. We carefully checked the descriptions involving “the production cycle” in the full text and corrected them to “growth rate” accordingly.

Q30. Line 566-571- Does it refer to a figure? Table?

Answer: Yes, the data on the grazing rates of zooplankton are from Supplementary Table S5. We have supplemented the information in the revised MS as follows:

P13L289-P13L292

On the fourth day, the grazing rates of zooplankton on micro, nano- and pico-phytoplankton under intermediate and high doses of atrazine decreased by 18.1% and 37.7%, 16.9 and 11.9, and 2.9% and 24.5% (Supplementary Table S5)

Q31. Line 579- Fig. S2b is absolutely not referring to diversity. Please, carefully proof-read the paper so there is no annoying mistakes as this one.

Answer: We apologize for the carelessness and have carefully checked the paper to avoid any similar mistakes. In the revised manuscript, the inappropriate citation has been replaced as follows:

P14L307-P14L308

The alpha and beta diversity (Supplementary Table S3, Supplementary Fig. S2b) of the micro-zooplankton community were both significantly changed under low and intermediate doses of atrazine exposure.

Q32. Line 579-582 - not sure the figure is the correct one, if yes to what protozoa correspond to? please specify. Not clear if we are talking about molecular or microscopic analyses here. Please, try as much as possible to improve the clarity of the results you present.

Answer: We apologize for this ambiguous description. The figure is correct, and the protozoa (heterotrophic protists) in the article correspond to the phylum Ciliophora. To facilitate the distinction between molecular results and morphological identification results, two corresponding headings have been added as follows:

P14L304-P15L323

High-throughput sequencing

The results of high-throughput sequencing showed that metazoans such as copepod larvae were the dominant group (97%) of micro-zooplankton in the control group, while the proportion of **heterotrophic protists** was only 3%. The alpha and beta diversity (Supplementary Table S3, Supplementary Fig. S2b) of the micro-zooplankton community were both significantly changed under low and intermediate doses of atrazine exposure. The relative abundance of **heterotrophic**

protists (Ciliophora) significantly increased to 10%, although metazoan larvae were still predominant (Fig. 5, Supplementary Table S4). With a further increase in atrazine concentration, the dominant phyla of microplankton changed drastically, and the proportion of heterotrophic protists (*Holosticha diademata*) significantly increased to 40.0% (Supplementary Table S4, S6), becoming the most dominant group. In contrast, the relative abundance of arthropod larvae decreased from 74.4% in the intermediate-concentration group to 21.1% in the high-concentration group. Concomitantly, the proportion of Platyhelminthes (*Paraplehnia seisuiae*) also significantly increased, from <1% in the intermediate-concentration group to 19.9% in the high-concentration group (Fig. 5, Supplementary Table S4, S6).

Morphological identification

The morphological identification results were consistent with the trend of the high-throughput sequencing (Supplementary Table S6):

Q33. Line 697-699 - Not sure the authors present data from adjacent waters of Ukraine (as there is no data in the Black Sea), so not sure about what the authors are talking about?

Answer: We apologize for this mistake. To our knowledge, no herbicide residue data were found in the Black Sea region. The presentation here was mainly based on data from the Mediterranean region. To ensure the rigor of the conclusion, the relevant statement has been deleted from the revised MS accordingly.

Q34. Line 740- reference?

Line 776- reference?

Answer: The references have been provided according to the reviewer's comments.

P20L441-P20L443

Based on this, scientists predict that such widespread herbicide contamination could have

immeasurable effects on oceanic productivity^{36,37}.

P22L482-P22L483

The primary production of marine phytoplankton worldwide is estimated to be $3-7 \times 10^{10}$ tons of carbon/year^{11, 40, 41}.

References:

11. Field CB, Behrenfeld MJ, Randerson JT, Falkowski P. Primary Production of the Biosphere: Integrating Terrestrial and Oceanic Components. *Science* 281, 237-240 (1998).

36. Johnston EL, Mayer-Pinto M, Crowe TP. REVIEW: Chemical contaminant effects on marine ecosystem functioning. *Journal of Applied Ecology* 52, 140-149 (2015).

37. Bester K, Hühnerfuss H, Brockmann U, Rick HJ. Biological effects of triazine herbicide contamination on marine phytoplankton. *Archives of Environmental Contamination and Toxicology* 29, 277-283 (1995).

40. Mattei F, Scardi M. Collection and analysis of a global marine phytoplankton primary-production dataset. *Earth Syst Sci Data* 13, 4967-4985 (2021).

41. Carr M-E, et al. A comparison of global estimates of marine primary production from ocean color. *Deep Sea Research Part II: Topical Studies in Oceanography* 53, 741-770 (2006).

Q35. Line 779- immeasurable or hard/complicated to measure?

Answer: Thank you for the suggestion. The sentence has been rephrased as follows:

P22L488-P22L489

These losses will be **difficult to measure** if the secondary effects on marine organisms at different trophic levels throughout the food chain are considered.

Q36. Line 798- ‘larger specific areas’: what does this mean? That tiny cells have a larger surface area to volume ratio than bigger ones...please, be specific and clear.

Answer: We apologize for the ambiguous description. The sentence has been rephrased as follows:

P23L512-P23L513

Tiny cells have a higher surface area to volume ratio than larger cells^{46, 47}.

Q37. Line 800-803- why do you state here that molecular approach is not good enough to

accurately describe species lower than genus level? Is it because of the approach you use of clustering reads at 97%

Answer: As the reads from high-throughput sequencing techniques are relatively short, it is difficult to obtain information such as full-length 18S/ITS rDNA sequences in phytoplankton, and this approach cannot provide particle size information at the species level. The description has been provided in the revised MS accordingly.

P23L516-P24L519

As the reads from high-throughput sequencing techniques are relatively short, it is difficult to obtain information such as the full-length 18S/ITS rDNA sequences in phytoplankton, and these techniques cannot provide particle size information at the species level.

Q38. Line 799 - Not sure about this statement, or not clear about what sizes you are comparing: with the reference you cite, are you talking about cells compared with macro-organisms? Or nano-plankton compared to pico-plankton? please be more specific about what you want to say here.

Answer: We apologize for the ambiguous description. The sentence has been rephrased as follows:

P23L513-P23L515

Thus, for the same biomass, nano-phytoplankton contributes more to primary productivity than micro-phytoplankton ⁴⁸.

Reference:

48. Chen B, Liu H. Relationships between phytoplankton growth and cell size in surface oceans: Interactive effects of temperature, nutrients, and grazing. *Limnology and Oceanography* 55, 965-972 (2010).

Q39. Line 848-850- Not sure this is true. The change in size of particle might modify the flux of matters from the main and traditional food chain to the microbial loop, but not sure what does ‘prolong the transmission process’ and ‘reduce the transmission efficiency’ mean? For example, heterotrophic protists are very efficient to bioremineralize matter from primary production. Please, clarify, be more specific or give more details and references about your ecological statements.

Answer: Thank you for the suggestion. The description has been rephrased, and new references have been provided as follows:

P27L584-P2L589

The change in particle size might modify the flux of matter from the main and traditional food chain to the microbial loop, which might prolong the transmission process of the marine food chain and reduce the transmission efficiency of primary productivity, since the microbial loop was the most inefficient among multiple routes for algal primary production to transfer to higher trophic levels in terms of respiratory carbon losses ^{19, 59, 60}.

References:

19. Worden AZ, Follows MJ, Giovannoni SJ, Wilken S, Zimmerman AE, Keeling PJ. Rethinking the marine carbon cycle: Factoring in the multifarious lifestyles of microbes. *Science* 347, 1257594 (2015).

59. Sswat M, et al. Food web changes under ocean acidification promote herring larvae survival. *Nature Ecology & Evolution* 2, 836-840 (2018).

60. Berglund J, Müren U, Båmstedt U, Andersson A. Efficiency of a phytoplankton-based and a bacterial-based food web in a pelagic marine system. *Limnology and Oceanography* 52, 121-131 (2007).

Q40. Figures and legends:

Overall, I think the figures are not very clear or easy to read. With figures with maps, I do not understand the units of the bar graphs that are plotted (e.g. figures 1 and 2). Font should be bigger, especially for figure S2a for example, where it is very hard to read the numbers that then should be connected (not easily) with herbicides. It should be easier for the readers to understand the figures. Figures 3 and 5, change the direction of the legend squares so it is easier for the readers to connect taxonomy and the bar graphs. Figure 6, I am colour-blind, so the choice of colors does not fit me at all. Figure 7, you mention "chronological order" but it is not explained anywhere in the main text. Figure 9 and Figure S5, and some others, the unit of the bar legends on the right are not explained. Figure S6, not sure about what panels are what letters?

Answer: Thank you for the comments. All the figures have been improved in the revised manuscript

accordingly (Reviewer 1, Q5).

Supplementary Fig. S5. The distribution of the three typical herbicide classes (triazine, phenylurea, and amide herbicides) in the surface waters of the Bohai Sea and Yellow Sea of China in autumn 2017 (a-c) and spring 2018 (d-f). (a, d) Triazine herbicides; (b, e) phenylurea herbicides; (c, f) amide herbicides. The colors represent high (red/green/blue) and low (gray) concentrations of the three typical herbicide classes. The units are nmol L^{-1} .

In addition, the information about “chronological order” has been added in the revised MS as follows:

P29L641-P29L647

In addition, from 1990 to 2022, the agricultural consumption of herbicides worldwide showed an overall increasing trend: the level was relatively flat from 1990 to 2000, increased rapidly from 2001 to 2012, and then stabilized (FAOSTAT (Jun 16, 2023) <https://www.fao.org/faostat/en/#data/RP/visualize>). The survey data of each sea area were roughly divided into three chronological stages (1990–2000, 2001–2011, and 2012–2022) to analyze the changes in herbicide residue concentration in each sea area over time.

Q41. Overall, if I think the subject is brave, the approach is original and the study is of good quality in term of research design and analyses, I am a little bit disappointed by the choice of references, clarity of results and development of the discussion. If publication is accepted it would need a deep revision in the writing so it can suit the editorial requirements of a Nature Communications paper.

Answer: Thank you very much for your constructive comments on our manuscript. We apologize for these errors and unclear descriptions in the references, results, and discussion sections. In the revision, the language of the whole manuscript has been refined by a professional editing service, and all the comments and suggestions from you and the other reviewers have been addressed and listed point by point.

Reviewer 3

The scope and purpose of the paper is very commendable - to provide a global assessment of the

risk posed by herbicides, although it is only the risk posed by 12 triazine herbicides. The authors have conducted a very thorough set of experiments to compliment the data acquisition and analyses that they have conducted.

I did not read the whole manuscript because I had so many questions and concerns about the methodology that I could not be confident that they are accurate and therefore it was not possible to review the results, discussion and conclusions. It is not necessary for the answers to some of my questions to be in the text of the manuscript but at the very least they need to be provided in the Supplementary Material section. I do not have expertise in genomics, so I cannot make any comment on the appropriateness of the information provided, nor the methods that were used. I do not believe that sufficient information is provided to permit readers to repeat the assessment or to use the methods for their own similar analysis.

Answer: Thank you very much for ascertaining the merits of this study, pointing out the shortcomings of our manuscript directly, and giving the valuable comments and suggestions, which are of great help to the improvement of our manuscript. We regret that the previous unclear description of the methodology caused trouble for the reviewers. We have now substantially revised the Methods section. Meanwhile, the other parts of the manuscript have been revised accordingly based on the comments and suggestions from the other reviewers. In addition, we invited the professional English editing service to improve language. The detailed point-to-point responses are as follows:

The sources of pesticide concentration data for the 15 bays must be provided.

Answer: As suggested, the raw data on herbicide concentrations for the 15 bays has been submitted as Source File (XLSX, Fig S1, S2, S7 and S9).

Q1. Insert "the extent of"

Answer: The sentence has been rephrased accordingly.

P2L30-P2L30

because **the extent of** worldwide herbicide pollution in coastal waters

Q2. As you are talking about aquatic concentrations of herbicides this should be concentration-response relationship.

Answer: Following the reviewer's suggestion, "dose-response relationship" has been replaced throughout the text with "concentration–response relationship".

Q3. Calculated or determined?

Answer: The word “detected” has been replaced with “determined”.

P2L33-P2L34

we **determined** the median, third quartile and maximum concentrations of 12 triazine herbicides.

Q4. Not a SI unit, which is tonnes

Answer: We apologize for this mistake. The word “tons” has been replaced with “tonnes” accordingly.

PEL54-P3L54

Millions of tonnes of herbicides are used annually worldwide

Q5. Not sure about that.

Answer: New references have been added in the revised version to support this viewpoint.

P3L58-P3L61

The inhibition of photosynthesis in coral symbiotic algae (zooxanthellae) by herbicides is thought to be an important factor contributing to the bleaching of coral reefs in Australia's Great Barrier Reef^{7, 8}.

References

7. Lewis SE, et al. Herbicides: a new threat to the Great Barrier Reef. *Environmental Pollution* 157, 2470-2484 (2009).

8. Brodie JE, et al. Terrestrial pollutant runoff to the Great Barrier Reef: an update of issues, priorities and management responses. *Marine pollution bulletin* 65, 81-100 (2012).

Q6. Not sure I'd agree with that, but it is definitely a significant potential problem. I think this comment should be tone down or some justification provided to support this claim

Answer: Thanks to the reviewer for the valuable suggestion. The expression has been rephrased to avoid possible disputes. In particular, we removed the description “lead to offshore desertification” and changed the title of this manuscript as well based on the reviewer’s comments.

P3L68-P3L70

Whether the inhibition of sensitive phytoplankton communities by herbicides will seriously affect offshore primary productivity and thus lead to offshore desertification is an **important potential problem**.

The title of this manuscript has been changed to “**Will herbicide pollution weaken coastal primary production and disturb micro-zooplankton in the food chain?**”

Q7. Dashes that replace "to" as in 8 to 10 should be a en dash.

Answer: The punctuation has been revised accordingly.

P3L73-P3L74

nutrient uptake rate and the expression of key carbon sequestration enzyme-encoding genes of planktonic algae ¹²⁻¹⁴.

Q8. I'm not sure what this means. Could you rephrase or place in parentheses an example of what you mean?

Answer: We apologize for our ambiguous description. The phrase “production periods” has been replaced with “**growth rates**”.

P4L82-P4L83

Different phytoplankton groups have different preferences for substrates (such as NO_3^- and NH_4^+) and different **growth rates**

Q9. Refs to support this are needed

Answer: The reference about the microbial loop has been provided in the revised MS.

P4L86-P4L87

The smaller pico-phytoplankton mainly enter the complex microbial loop through ingestion by **heterotrophic protists** ¹⁹,

Reference:

19. Worden AZ, Follows MJ, Giovannoni SJ, Wilken S, Zimmerman AE, Keeling PJ. Rethinking the marine carbon cycle: Factoring in the multifarious lifestyles of microbes. *Science* 347, 1257594 (2015).

Q10. Replaced “response law” with “responses”

Answer: The text has been revised accordingly.

Q11. I don't understand what you mean. Rephrase or delete

Answer: According to the reviewer's suggestion, the sentence has been rephrased as follows:

P5L103-P5L105

However, most of the current research on herbicide toxicity is based on risk assessments of a **single herbicide in a small region**^{20, 21}

Q12. Some more references to support this claim is required

Answer: According to the reviewer's suggestion, new references to support this viewpoint have been added as follows:

P5L117-P5L120

In contrast to the single herbicide types typically found in farmland soils, the ocean is the final sink for nearly all herbicides, and the composition of herbicides in seawater is extremely complex^{10, 21-23}.

References:

22. Tyohemba RL, Pillay L, Humphries MS. Herbicide residues in sediments from Lake St Lucia (iSimangaliso World Heritage Site, South Africa) and its catchment areas: Occurrence and ecological risk assessment. *Environmental Pollution* 267, 115566 (2020).

23. Haynes D, Müller J, Carter S. Pesticide and Herbicide Residues in Sediments and Seagrasses from the Great Barrier Reef World Heritage A

Q13. Mode or mechanisms of action would be a better and more accurate phrase to use

Answer: The phrase has been revised accordingly.

P6L121-P6L122

There is an additive toxicity effect among herbicides with the same **mode of action**²⁴.

Q14. References to support this are required

Answer: References to support this viewpoint have been supplemented as suggested:

P6L139-P6L141

Agricultural production is the main source of herbicides in coastal waters, and the types and dosages of herbicides used on farmlands largely determine the herbicide pollution status in the adjacent seas^{22, 25}.

References:

22. Tyohemba RL, Pillay L, Humphries MS. Herbicide residues in sediments from Lake St Lucia (iSimangaliso World Heritage Site, South Africa) and its catchment areas: Occurrence and ecological risk assessment. *Environmental Pollution* 267, 115566 (2020).

25. Sun B, Zhang L, Yang L, Zhang F, Norse D, Zhu Z. Agricultural non-point source pollution in China: causes and mitigation measures. *Ambio* 41, 370-379 (2012).

Q15. This is not really an appropriate reference to use as it is only available upon purchase

Answer: Thank you for the comment. A more appropriate reference has been provided as follows:

P6L141-P6L142

At present, there are more than 1,500 herbicides on the market worldwide, with more than 300 active ingredients²⁶.

Reference:

26. Heap I. *Herbicide resistant weeds*. Springer (2014).

Q16. I don't understand this. There are 760 water samples? Surely not. Do you mean 760 bays included in the study? What are 4253 survey data points. From this I have no idea of how much data/sites/regions were included.

Now we learn there is data for 15 bays. This information should have been provided earlier.

Answer: We apologize for this misunderstanding. To quantify the impact of current herbicide pollution on offshore primary productivity on a larger scale, we first collected survey data published from 1995 to 2022 on herbicide pollution in bays worldwide and built a dataset that included 760 water samples and 4253 herbicide concentration data from 15 bays. According to the reviewer's suggestion, the sentence has been rephrased as follows:

P29L636-P29L637

which included 760 water samples and 4253 herbicide concentration data from 15 bays.

Q17. Surely the references used to provide the data for these 15 bays must be cited!

Answer: As mentioned above, the references and raw data for herbicide concentrations in these 15 bays have been submitted as Source File (Source data Fig. 1, Fig. 2, Fig. 7, and Fig. 9).

Q18. Greater consistency in the spatial precision is needed. Australia is a continent and the 7th largest country in the world. Vilaine Bay is, in comparison, incredibly precise. Maybe state the countries where the bays are located then have a table or map showing the actual locations?

Answer: Thank you for the comment. The 15 bays have been reclustered as follows in the revised MS:

P29L637-P29L639

The 15 bays were clustered into seven sea areas, including the United States east coast, Mexico Bay, France, the Mediterranean Sea, South Africa, East Asia and Australia.

Q19. What water quality safety standard?

Answer: We apologize for the ambiguous description. The “water quality safety standard” here means the water quality requirements imposed by the Water Framework Directive 2000/60/EC (WFD) and the Drinking Water Directive 1998/83/EC (DWD), which set a maximum allowable concentration (MAC) of $0.1 \mu\text{g L}^{-1}$ for any single pesticide and $0.5 \mu\text{g L}^{-1}$ for the total pesticide concentration. The reference and the details about the water quality safety standard have been added in the revised MS as follows:

P30L652-P30L655

with the highest determined concentration (several times higher than the water quality safety standard⁶⁸, which sets a maximum allowable concentration of $0.1 \mu\text{g L}^{-1}$ for any single pesticide and $0.5 \mu\text{g L}^{-1}$ for the total pesticide concentration) and targeted inhibitory effects on phytoplankton photosynthesis

Reference:

65. Jess S, Kildea S, Moody A, Rennick G, Murchie AK, Cooke LR. European Union policy on pesticides: implications for agriculture in Ireland. *Pest Management Science* 70, 1646-1654 (2014).

Q20. This work is fundamental to the outcomes of the whole study and yet there is no information provided on this. The Supplementary information section, at least, must show all of these concentration-response relationships and how the Teq calculations were conducted. We (readers) are being asked to trust the authors.

Answer: Thank you for the comments. The details of the toxicity equivalent conversion of triazine herbicides are provided in the Supplementary Information section as follows:

The toxicity equivalent conversion of triazine herbicides

P. tricornutum Pt-1 (CCMP 2561) cells, purchased from the National Center for Marine Algae and Microbiota (NCMA), were cultivated in f/2 medium⁹⁸ at 20°C under 60 $\mu\text{mol photons m}^{-2} \text{ s}^{-1}$ irradiance and a 12 L:12 D photoperiod until they reached the exponential phase. Cell concentrations were measured microscopically using a Sedgewick-Rafter (SR) counting chamber (Phycotech, MI, USA) to monitor the growth of these cultures⁹⁹. Each sample was homogenized gently and briefly before analysis. One of the fully mixed sample was carefully dispensed into the counting cells. The average cell number of *P. tricornutum* Pt-1 in 20 subcells of the Sedgewick-Rafter counting chamber was counted. The algae were harvested by centrifugation at 4°C (5000 \times g, 10 min). The cell pellets were washed twice and suspended in f/2 liquid medium to an OD₇₃₀ of 3.0.

The short-term toxicity effects of the ten triazine herbicides on *P. tricornutum* Pt-1 were tested using an Infinite M200 Pro plate reader (Tecan, Zurich, Switzerland) with excitation and emission wavelengths of 440 and 680 nm¹⁰⁰, respectively. Five micromoles of each herbicide was weighed accurately and diluted in 5 mL of methanol as a stock solution (1 mmol L⁻¹). A series of twofold dilutions of the stock solution were performed. The concentration series from 0.1 nmol L⁻¹ to 32 $\mu\text{mol L}^{-1}$ was arranged in 48-well microtiter plates (NEST Biotech, Shanghai, China) with three replicates. The procedure was as follows. Ten microliters of herbicide solution was added to the corresponding well and dried on a clean bench. Then, 1 mL of algae suspension prepared as

described in section 2.1 was added. After being shaken for 10 min, the plates were incubated at 20°C as described above. The fluorescence intensity was measured every 12 hours, and the data at 72 hours were selected for the determination of the inhibitory effects of each herbicide on the chlorophyll *a* fluorescence of *P. tricornutum* Pt-1 cells. Toxicity data were recorded for the description of the complete effect range (0–100%) for all individual triazine herbicides. The best-fitting model for each concentration response relationship was chosen as described by Scholze et al.⁹⁴

The mixture toxicity of triazine herbicides to strain *P. tricornutum* Pt-1 was calculated based on the concentration addition (CA) model and the single-substance concentration–response curves. The average concentration ratio of individual components (p_i) at the 64 stations was employed in the following calculation according to equation (1)⁷⁸:

$$ECx_{\text{Mix}} = \left(\sum_{i=1}^n \frac{p_i}{ECx_i} \right)^{-1} \quad (1)$$

In this equation, ECx_{Mix} is the total concentration of the mixture provoking $x\%$ effects; p_i denotes the fraction of component i in the mixture; and ECx_i denotes the equivalent effective concentration of a single substance, *i.e.*, the concentration–response curve of the mixture can be predicted by the percentage and toxicity data of a single substance¹⁰¹. The corresponding concentrations of the mixed triazine herbicides in the Yellow Sea and the Bohai Sea having 1 to 99% effects were calculated in steps of 1%. Based on the predicted concentration–response curve, the joint effects ($x\%$) of the triazine herbicides in the Yellow Sea and the Bohai Sea were obtained. Using equation (1), the ecotoxicity of triazine herbicides in the Yellow Sea and the Bohai Sea can be indicated by the equieffective concentration of atrazine, which makes it possible to accurately assess the joint effects of multiple herbicides in natural seawaters.

The concentration–response functions for each single triazine herbicide and the equieffective concentrations of atrazine that can cause a toxic effect equivalent to that of all 12 detected triazine residues at in situ concentrations have been added in Supplementary Table S2 and the Source data accordingly.

Supplementary Table S2. The concentration–response functions for each single triazine herbicide.

	EC50	Model	A1	A2	X0	p	R ² adj
Atrazine	141.3 ± 12.5	L	1.7142	104.2656	0.1349	0.9173	0.989
DEA	580.3 ± 21.8	L	1.5695	101.0427	0.6049	1.1381	0.995
Propazine	147.1 ± 15.2	L	4.3068	103.3116	0.1584	1.1995	0.989
Simazine	28.7 ± 1.6	L	0.3833	103.3089	0.0309	0.8911	0.996
Terbutryn	8.5 ± 1.1	L	2.1441	101.7772	0.0091	1.1025	0.997
Ametryn	25.1 ± 1.2	L	-0.5755	102.6998	0.0262	0.9461	0.996
Dipropetryn	37.0 ± 3.1	L	-1.0603	101.592	0.0375	0.8534	0.993
Cyanazine	280.3 ± 11.7	L	1.1414	109.9918	0.36235	0.66913	0.995
Cybutryne	4.3 ± 0.3	L	0.678	100.21925	0.00445	1.08514	0.999
	EC50	Model	A1	A2	d	k	R ² adj
DIA	732.1 ± 36.5	W	4.0148	99.4471	0.8182	0.8223	0.989
Prometryn	849.1 ± 21.7	W	1.9596	99.3406	1.0341	0.8109	0.997
Prometon	76.2 ± 3.3	W	5.80E-46	96.4563	0.643	0.2094	0.996
Atrazine		L	3.1467	101.122	0.103	1.0778	0.995

Note: EC50: 50% effective concentration, units are nmol L⁻¹; L: Logistic; W:

Weibull; A1, A2, X0, p, d and k represent the parameters of the functions.

Q21. Some explanation of why this species was chosen as the indicator species for all marine phytoplankton is required

What does this mean (PT-1)?

Answer: Indeed, *Phaeodactylum tricorutum* is one of the most cosmopolitan diatom species⁶⁶, especially in nutrient-rich coastal waters^{67,68}. It is also a model organism among diatoms and is widely used in research fields such as global climate change^{69, 70}, eutrophication^{71, 72}, and marine pollution^{73, 74}.

As the reviewers pointed out, *Chaetoceros*, *Thalassiosira*, and *Skeletonema* are also dominant diatom taxa. In fact, in our previous research, five dominant diatom species (*Chaetoceros muelleri* CCMP1316, *Phaeodactylum tricorutum* Pt-1, *Thalassiosira pseudonana* CCMP1335, *Nitzschia closterium*, and *Skeletonema costatum*) were selected to analyze the sensitivity of different algal species to atrazine.

Fig. S4. Concentration–response curves showing the effects of atrazine on the five tested *Bacillariophyceae* (solid lines) and five tested *Dinophyceae* (dashed lines) after 72 hours. The

numbers 1–10 represent the ten algal species in the following order: *Chaetoceros muelleri* CCMP1316, *Phaeodactylum tricorutum* Pt-1, *Thalassiosira pseudonana* CCMP1335, *Nitzschia closterium*, *Skeletonema costatum*, *Prorocentrum donghaiense* CCMA-129, *Gymnodinium* sp. CCMA-167, *Amphidinium carterae* CCMA-279, *Heterocapsa circularisquama* CCMA-128, and *Gyrodinium* sp.

Yang L, Mou S, Li H, et al. Terrestrial input of herbicides has significant impacts on phytoplankton and bacterioplankton communities in coastal waters[J]. Limnology and Oceanography, 2021, 66(11): 4028-4045.

Considering that *Chaetoceros* is the most sensitive species to atrazine, the degree of inhibition of atrazine on coastal primary productivity may be overestimated if calculated based on the dose–response curve of *Chaetoceros*. In contrast, *P. tricorutum* is moderately sensitive to atrazine and was therefore used for the toxicity equivalent conversion of triazine herbicides in this study.

Based on the reviewer’s comment, we have supplemented the description of the reasons for using *P. tricorutum* to determine the concentration–response effects and added the corresponding references.

P₃₀L₆₆₁-P₃₀L₆₇₀

Selection of model organism: *Phaeodactylum tricorutum* was used in this study for the toxicity equivalent conversion of triazine herbicides for three reasons: 1) it is one of the best-known cosmopolitan diatom species⁶⁹, especially in nutrient-rich coastal ecosystems^{70, 71}; 2) it is a model organism among diatoms, widely used in research on global climate change^{72, 73}, eutrophication^{74, 75}, marine pollution^{76, 77}, etc.; 3) among five dominant diatom species (*Chaetoceros muelleri* CCMP1316, *Phaeodactylum tricorutum* Pt-1, *Thalassiosira pseudonana* CCMP1335, *Nitzschia closterium*, and *Skeletonema costatum*), it is moderately sensitive to atrazine¹⁰ and is therefore suitable for the calculation of the inhibition rate of atrazine on diatoms in offshore waters.

References

10. Yang L, Mou S, Li H, Zhang Z, Jiao N, Zhang Y. Terrestrial input of herbicides has

- significant impacts on phytoplankton and bacterioplankton communities in coastal waters. *Limnology and Oceanography* **66**, 4028-4045 (2021).
69. Gao K, *et al.* Rising CO₂ and increased light exposure synergistically reduce marine primary productivity. *Nature Climate Change* **2**, 519-523 (2012).
 70. Malviya S, *et al.* Insights into global diatom distribution and diversity in the world's ocean. *Proceedings of the National Academy of Sciences* **113**, E1516-E1525 (2016).
 71. Buitenhuis ET, *et al.* MAREDAT: towards a world atlas of MARine Ecosystem DATA. *Earth Syst Sci Data* **5**, 227-239 (2013).
 72. Buck JM, *et al.* Lhex proteins provide photoprotection via thermal dissipation of absorbed light in the diatom *Phaeodactylum tricornutum*. *Nature Communications* **10**, 4167 (2019).
 73. Baker KG, Geider RJ. Phytoplankton mortality in a changing thermal seascape. *Global Change Biology* **27**, 5253-5261 (2021).
 74. You Y, *et al.* Trypsin is a coordinate regulator of N and P nutrients in marine phytoplankton. *Nature Communications* **13**, 4022 (2022).
 75. Cáceres C, Spatharis S, Kaiserli E, Smeti E, Flowers H, Bonachela JA. Temporal phosphate gradients reveal diverse acclimation responses in phytoplankton phosphate uptake. *The ISME Journal* **13**, 2834-2845 (2019).
 76. Huang R, *et al.* Physiological and molecular responses to ocean acidification among strains of a model diatom. *Limnology and Oceanography* **65**, 2926-2936 (2020).
 77. Xie Z, *et al.* Organophosphate ester pollution in the oceans. *Nature Reviews Earth & Environment* **3**, 309-322 (2022).

Q22. CK what does this acronym stand for? Control?

Answer: We apologize for this unclear description. Yes, CK means Control here. The sentence has been revised as follows:

P₃₂L₆₉₈-P₃₂L₇₀₃

In the laboratory, four atrazine doses (0, 0.5, 5 and 50 nmol L⁻¹) were used in the treatment groups: 1) control check (CK), consisting of uncontaminated seawater; 2) 0.5 nmol L⁻¹ atrazine; 3) 5 nmol L⁻¹ atrazine; and 4) 50 nmol L⁻¹ atrazine. Each treatment sample consisted of 80 L of seawater in a transparent polycarbonate bottle (100 L), and each treatment had three replicates.

Q23. This is pseudo-replication. Really, there should have been separate test vessels which were sampled at each sampling time. One set samples at each sampling time.

Answer: Thank you for the comments. The continuous sampling method used in this experiment is

mainly due to the following two considerations. 1) to ensure the continuity of various indicators (phytoplankton community structure and other physical/chemical indicators) and to compare the changes in these indicators in a time series. Although the succession of community structure under the stress of environmental factors is regular, it is not static. Although destructive sampling can ensure the independence of experimental data, it does not allow convenient comparison of the continuous changes in indicators in a time series. 2) Considering the operability of the experiment. In our experiment, each treatment sample consisted of 80 L of seawater in a transparent polycarbonate bottle (100 L), and each treatment had three replicates. For 4 concentration gradients and 8 sampling time points, destructive sampling would require 96 such 100 L containers. With so many 100 L containers, we don't have space to place them and it would be difficult to ensure consistent environmental parameters such as temperature and light during the cultivation process. We hope to gain your understanding.

Q24. What about light intensity and cycle (hours of daylight:dark)? What other quality assurance and quality control measures were implemented? In this journal they shouldn't be provided in the text but should be provided in the Supplementary Info

Answer: We apologize for this unclear description. To simulate the in situ environment, natural light was used in this experiment, as described in the text . According to the reviewer's suggestion, information about the temperature, light intensity and other quality assurance has been added as follows:

P₃₂L₇₀₄-P₃₂L₇₀₇

The bottles were incubated at room temperature (25°C ± 3°C) for 30 d in the laboratory under natural light conditions, and subsamples were collected from each bottle on days 0, 1, 2, 4, 7, 14, 21 and 30 for subsequent analyses.

Q25. OK, but why were the samples on the filters re-suspended and a particle size analysis or flow cytometer analysis conducted. I think a reason for the approach used is needed.

Answer: We are sorry for causing the reviewer to misunderstand this experimental procedure. Indeed, the filters were used for DNA extraction and high-throughput sequencing only. The particle

size analysis mentioned here was carried out after aligning these sequences with the NCBI BLAST database to obtain species information, which is helpful to further analyze the relationship between the changes in the particle size structure and the community structure of phytoplankton under different environmental concentrations of triazine herbicides. And the flow cytometry analysis mentioned by the reviewer is the content of section 2.5 of this paper.

Accordingly, we have made modifications for clearer description as follows:

P₃₄L₇₅₅-P₃₄L₇₆₁

By aligning these sequences with the NCBI BLAST database, the phytoplankton species with the highest sequence similarity were identified, and the particle size information of the corresponding species of each taxon was obtained. The corresponding results were helpful to further analyze the relationship between the changes in the particle size structure and the community structure of phytoplankton under different environmental concentrations of triazine herbicides.

Q26. But how were these measured?

Answer: The dilution experiments were performed based on the methods of Worden and Binder. According to the reviewer's comments, we added descriptions of the specific methods of determining the apparent growth rate of phytoplankton, the intrinsic growth rate of phytoplankton, and the feeding rate of zooplankton in the Supplementary Info as follows:

P₃₅L₇₆₄-P₃₅L₇₇₆

The effect of atrazine on the apparent or intrinsic growth rate of phytoplankton and the feeding rate of zooplankton in natural seawater subsamples were measured based on the methods of Worden and Binder⁸⁵ with a few modifications. Briefly, subsamples of each treatment group that were collected on the fourth day were considered the initial seawater (ISW) samples, and particle-free water (PFW) samples were prepared by filtering the ISW samples through Millipore filters (0.2 μm). Then, the ISW was diluted with PFW to five target dilutions of 100%, 80%, 60%, 40%, and 20% in 2.8 L transparent polycarbonate bottles. The incubation volume was 2.5 L with triplicates. All the bottles were incubated in a water incubator for 24 hours, and then the subsample of each dilution gradient was filtered sequentially through 20, 2, and 0.2 μm polycarbonate filters and stored in the dark at -

20°C for further analysis. The phytoplankton retained on the 20, 2, and 0.2 µm filters were designated as micro-phytoplankton, nano-phytoplankton and pico-phytoplankton, respectively.

References:

85. Worden AZ, Binder BJ. Application of dilution experiments for measuring growth and mortality rates among *Prochlorococcus* and *Synechococcus* populations in oligotrophic environments. *Aquatic Microbial Ecology* **30**, 159-174 (2003).

Q27. Where have the zooplankton come from? They have not been mentioned before or that they were introduced into the phytoplankton cultures. What species? Or were they a mix of what species? Where were they obtained from? What density or number of zooplankton were introduced? So many questions.

Answer: We apologize for this misunderstanding. The water samples used for the dilution experiments were collected from the herbicide microcosm system in section 2.1.1 on the fourth day. Therefore, the zooplankton mentioned here are not artificially introduced but natural communities in seawater, as are the phytoplankton. To prevent ambiguity, we rephrased the sentences in the revised MS as follows:

P₃₅L₇₆₄-P₃₅L₇₇₆

The effect of atrazine on the apparent or intrinsic growth rate of phytoplankton and the feeding rate of zooplankton in natural seawater subsamples were measured based on the methods of Worden and Binder⁸⁵ with a few modifications. Briefly, subsamples of each treatment group that were collected on the fourth day were considered the initial seawater (ISW) samples, and particle-free water (PFW) samples were prepared by filtering the ISW samples through Millipore filters (0.2 µm). Then, the ISW was diluted with PFW to five target dilutions of 100%, 80%, 60%, 40%, and 20% in 2.8 L transparent polycarbonate bottles. The incubation volume was 2.5 L with triplicates. All the bottles were incubated in a water incubator for 24 hours, and then the subsample of each dilution gradient was filtered sequentially through 20, 2, and 0.2 µm polycarbonate filters and stored in the dark at -

20°C for further analysis. The phytoplankton retained on the 20, 2, and 0.2 µm filters were designated as micro-phytoplankton, nano-phytoplankton and pico-phytoplankton, respectively.

References:

85. Worden AZ, Binder BJ. Application of dilution experiments for measuring growth and mortality rates among *Prochlorococcus* and *Synechococcus* populations in oligotrophic environments. *Aquatic Microbial Ecology* **30**, 159-174 (2003).

Q28. Yes, but this is the methods section and there are no methods here. This is not appropriate here. Some methods (see my previous comment) are required on the experimental design of the secondary effects of atrazine on zooplankton.

Answer: Thank you for the comments. This paragraph has been moved from the methods section to the results accordingly. Methods for the experimental design to analyze the secondary effects of atrazine on zooplankton are provided in the supplementary information, as mentioned above.

Q29. What microzooplankton samples? No collecting of these has been mentioned? How were they collected/ how many? etc

Answer: Both the phytoplankton and micro-zooplankton communities were obtained from the herbicide microcosm system in 2.2.1. According to the reviewer's comments, we rephrased the description as follows:

P₃₅L₇₇₉-P₃₆L₇₈₃

2.3.1 Effects of triazine herbicides at environmental concentrations on micro-zooplankton community structure in natural seawater

With reference to the high-throughput sequencing of phytoplankton, the specific primer pair 82F/Ek-516R was used to amplify the 18S rRNA gene of micro-zooplankton in the herbicide microcosm system (2.2.1).

Q30. Using what?

Answer: The micro-zooplankton in at least 20 random fields were identified and counted by using a Sedgwick-Rafter counting chamber under a light inverted microscope (magnification x100).

Q31. How often? What was the stocking density of the copepods and ciliates? Conditions for culturing these organisms?

What bacteria? What density? How often? How were they cultured?

How long were they acclimated to the laboratory culturing conditions?

What previous articles? Include some appropriate citations

How many concs?

The bait? What is this? More detail on the copepod experimental design is needed. Ditto for the ciliate tests

Answer: We apologize for our unclear description. The details of the toxicity test and references have been expanded as follows:

P₃₆L₇₉₄-P₃₈L₈₃₀

2.3.3 *Direct toxicity of atrazine to sensitive/tolerant microplankton groups*

Based on the observed changes in the micro-zooplankton community structure under atrazine exposure, atrazine at environmental concentrations can cause significant responses in some copepod larvae and ciliates. To distinguish whether this response is caused directly by the toxicity of atrazine to microplankton or indirectly by the influence of atrazine on the community structure and particle size composition of phytoplankton, thereby affecting microplankton foraging, we selected the larvae of two copepods (*Oithona similis* and *Paracalanus parvus*) and two ciliates (*Euplotes* sp. and *Strombidium* sp.), respectively, to assess their sensitivity to atrazine. The four experimental micro-zooplankton were isolated from the coastal waters of Shilaoren Bay (120°49'E, 36°09'N) and cultured in artificial seawater medium on 17 July 2021. Before the toxicity test, the copepods and ciliates stored in the laboratory were transferred with a micropipette to a new culture medium. Individuals of a uniform size and with good activity levels were selected and precultured at 25°C

for 48 hours.

Copepods: The toxicity tests on the two copepods were carried out with 20 acclimatized healthy individual copepods in 50 mL beakers with 20 mL of artificial seawater (S= 28–30‰; pH= 8.2±0.1) for 96 hours. The two copepods were fed daily with *Isochrysis galbana* (1.0×10^5 ind mL⁻¹). The stock cultures were maintained in a climate-controlled room/chamber at 20±1° C and a 14:10 h light:dark cycle. Other conditions were consistent with those of the ciliate experiments^{89, 90}.

Ciliates: The operational steps of this experiment were slightly modified from those described in previous articles^{91, 92}. Briefly, the ciliate toxicity tests were conducted in a 24-well plate, and 1 mL of boiled rice and wheat grain culture solution was added to each well. No food was added during the subsequent toxicity tests. A serial 2-fold dilution of atrazine concentrations (0.1 nmol L⁻¹ to 32 µmol L⁻¹) was prepared by adding the appropriate amount of atrazine stock solution. Both the atrazine-treated and control groups were established with 3 replicates. Twenty precultured ciliates were added to each microwell with a micropipette and incubated in a constant-temperature incubator at 25°C, oxygen saturation >45%, and photoperiod of 14:10 h light:dark. After 96 hours, the number of surviving ciliates in each group was counted under a stereomicroscope and recorded. Individuals that were incapacitated or had significant morphological changes were considered dead.

The 96 h LC50 value of atrazine for each zooplankton species was obtained by performing a probit regression analysis and a chi squared goodness of fit test on the experimental data with SPSS17.0⁹³. Using the logarithm of the atrazine concentration as the x value and the mortality of the test species as the y value, a regression curve was drawn and a regression equation was fitted to the data.

References:

86. Willis KJ, Ling N. The toxicity of emamectin benzoate, an aquaculture pesticide, to planktonic marine copepods. *Aquaculture* **221**, 289-297 (2003).

87. Willis KJ, Ling N. Toxicity of the aquaculture pesticide cypermethrin to planktonic marine copepods. *Aquaculture Research* **35**, 263-270 (2004).

88. Mortimer M, Kasemets K, Kahru A. Toxicity of ZnO and CuO nanoparticles to ciliated protozoa *Tetrahymena thermophila*. *Toxicology* **269**, 182-189 (2010).

89. Madoni P. The acute toxicity of nickel to freshwater ciliates. *Environmental Pollution* **109**, 53-59 (2000).

Q32. Were solvent controls established?

Answer: The solvent used for the standard solutions was blown dry with N₂, which eliminated the potential influence of the solvent on the results. To make the description clearer, we rephrased the sentence as follows:

P₃₈L₈₄₉-P₃₈L₈₅₀

To prevent methanol from interfering with the experimental results, the methanol in each flask was blown dry with N₂ in a fume hood,

Q33. Each? How many such relationships were developed? What were the biological endpoints used in these relationships?

Answer: We apologize for the inaccurate description. It has been revised as follows:

P₃₉L₈₅₆-P₃₉L₈₅₈

The best-fitting model for the concentration–response relationship between chlorophyll *a* and atrazine concentration was chosen as described by Scholze et al ⁹⁴.

The biological endpoint used in this test is the decrease in chlorophyll *a* concentration.

Q34. Was any pre-test analysis conducted to determine the density? Type of zooplankton present? Did it vary from test vessel to test vessel or between the treatments?

Answer: Natural seawater was employed for this test. So the phytoplankton mentioned here are not artificially introduced but natural communities in seawater.

Q35. Estimated. And there are heaps of assumptions being made in doing that.

Surely, phytoplankton species composition would differ in the 15 bays? Might not their sensitivity to atrazine or other PSII herbicides vary? How are any of these issues addressed in

this project?

Answer: Thank you very much for the valuable comments. Just as the reviewer mentioned, the composition of the phytoplankton community in different bays is not completely consistent due to the influences of temperature, nutrient salts and other factors. However, in general, offshore phytoplankton are dominated by diatoms and dinoflagellates. For example, *Chaetoceros* is a dominant genus of marine planktonic diatoms with worldwide distribution.

In our previous research, five dominant diatom species (*Chaetoceros muelleri* CCMP1316, *Phaeodactylum tricornutum* Pt-1, *Thalassiosira pseudonana* CCMP1335, *Nitzschia closterium*, and *Skeletonema costatum*) and five dominant dinophyceae were selected to analyze the sensitivity of different algal species to atrazine. The results indicated that the diatoms were generally more sensitive to herbicides than dinoflagellates.

Fig. S4. Concentration–response curves showing the effect of atrazine on the five tested *Bacillariophyceae* (solid lines) and five tested *Dinophyceae* (dashed lines) after 72 hours. The numbers 1–10 represent the ten algal species in the following order: *Chaetoceros muelleri* CCMP1316, *Phaeodactylum tricornutum* Pt-1, *Thalassiosira pseudonana* CCMP1335, *Nitzschia closterium*, *Skeletonema costatum*, *Prorocentrum donghaiense* CCMA-129, *Gymnodinium* sp.

CCMA-167, *Amphidinium carterae* CCMA-279, *Heterocapsa circularisquama* CCMA-128, and *Gyrodinium* sp.

Therefore, even if there are differences in the phytoplankton community structure in different bays, the results of this experiment can still largely reflect the current herbicide threats facing offshore areas.

However, as we mentioned in the discussion section, despite our efforts to simulate the in situ environment, there are still many factors that were not considered in this study (such as hydrodynamic diffusion and other herbicide types). In future research, we will do our best to improve them.

Q36. Where was this information obtained from?

Citations needed

Answer: This information was based on the CHENGRIDS database by Maggi et al. According to the reviewer's suggestions, we added the information and reference in the revised MS as follows:

P₃₉L₈₇₀-P₃₃₉L₈₇₆

The top 20 herbicides (based on the CHENGRIDS database²⁷) used for each crop class in 2015 were selected as the statistical range of the global herbicide geographical distribution.

The global georeferenced, crop specific, annual herbicide application rates were obtained from the PEST-CHENGRIDS database²⁷ and constrained against the country-specific pesticide use data reported in the FAOSTAT database²⁸ on chlorophyll a content and phytoplankton abundance.

Reference:

27. Maggi F, Tang FH, la Cecilia D, McBratney A. PEST-CHEMGRIDS, global gridded maps of the top 20 crop-specific pesticide application rates from 2015 to 2025. *Scientific data* 6, 1-20 (2019).

28. FAOSTAT. FAOSTAT: Database Collection of the Food and Agriculture Organization of the United Nations (FAO, 2019). (2019).

Q37. But how were these two crucial values calculated?

Answer: The methods to calculate the values have been provided as follows:

P₄₀L₈₈₀-P₄₀L₈₉₅

PECs

Due to the lack of historical data on herbicide use, the PEC in this study refers to the noncumulative environmental concentration. The spatially explicit approach of the Environmental Potential Risk Indicator for Pesticide version 2.1⁹⁵ was employed to estimate the PEC value of herbicide i on crop j ($PEC_{i,j}$) as follows:

$$S_{i,j} = VDT_j \frac{AR_{i,j}}{100} \quad (1)$$

$$PEC_{i,j} = \frac{S_{i,j}}{4500} \int_0^t \frac{C^{(t)} dt}{100} \quad (2)$$

Note: $S_{i,j}$ is the absorption amount of herbicide i on crop j ; $AR_{i,j}$ is the usage amount of herbicide i on crop j ; VDT_j is the percentage of soil coverage found in the coverage table (%)²⁸; $C^{(t)} dt$ is the residual proportion of herbicide after t days of application (based on DT₅₀ and DT₉₀ values).

PNECs

The PNEC values of herbicide i in soil ($PNEC_i^{SL}$) and water ($PNEC_i^{SW}$) were estimated from the LC50 values of earthworms and fish, respectively, as follows:

$$PNEC_i^{SL} = \frac{LC50_i^{earthworms}}{1000} \quad (3)$$

$$PNEC_i^{SW} = \frac{LC50_i^{fishes}}{1000} \quad (4)$$

Q38. From here on you seem to not be allocating a reference number to each citation. Nor is this citation in the reference list.

I'm confused this sentence implies Nagai et al developed these RP classes. But a quick search of this paper did not find any mention of Risk point or such classes.

Answer: We apologize for this mistake. The reference has been provided in the right format as follows:

P₄₁L₈₉₉-P₄₁L₉₀₀

According to the species sensitivity distribution data of 59 pesticides reported by Nagai et al⁹⁶,

Reference:

96. Nagai T. Ecological effect assessment by species sensitivity distribution for 68 pesticides used in Japanese paddy fields. *Journal of pesticide science* 41, 6-14 (2016).

In this study, the species sensitivity distributions (SSDs) of 68 commonly used pesticides were analyzed based on collected acute toxicity data as a higher-tier ecological effect assessment. Then, based on each SSD, the 5% hazardous concentration (HC5) values were calculated as the predicted no-effect concentrations for aquatic ecosystems.

Please review this information.

Q39. This will be rather important later in this paper. Some explanation of how such % species affected values were determined for various RP values is absolutely necessary.

As the RP is essentially the same as RQsum or HQsum I do not understand how these % species affected values were obtained.

Answer: We apologize for this unclear description. The relationship between the RP value and species fluorescence rate is mainly based on the species sensitivity distribution data of 59 pesticides reported by Nagai et al.

Fig. 1. Conceptual diagram of SSD. The variability of toxicity values (EC_{50} or LC_{50}) of 6 species is fitted to log-normal distribution. Arrow 1 indicates the derivation of HC_5 , and arrow 2 indicates the calculation of PAF from pesticide concentration.

It has been clarified in the revised MS as follows:

According to the species sensitivity distribution data of 59 pesticides reported by Nagai et al ⁹⁶, RP values are divided into four grades: $RP \leq 0$, negligible; $0 < RP \leq 1$, low risk; $1 < RP \leq 3$, medium risk; and $RP > 3$, high risk. $RP \leq 0$ means that the probability of any species being affected is less than 5%; $RP > 3$ means that the probability of a randomly selected species being affected by the herbicide exceeds 90% ⁹⁶.

Reference:

96. Nagai T. Ecological effect assessment by species sensitivity distribution for 68 pesticides used in Japanese paddy fields. *Journal of pesticide science* **41**, 6-14 (2016).

Q40. Is this referring to the flow cytometry mentioned in the next sentence?

Answer: We apologize for this carelessness. The reference has been deleted here.

Q41. Not correct cross reference

Answer: We apologize for this carelessness. Fig. 1a and Fig. 1b should be Fig. S1a and Fig. S1b, respectively. The text has been corrected as follows:

Environmental surveys were carried out in autumn (August 29 to September 24) 2017 and spring (March 26 to April 16) 2018 to determine the spatial distribution and seasonal variation in 22 herbicides that are widely used in upstream agricultural areas in the Bohai Sea and Yellow Sea (Supplementary Fig. S8a). The distribution of the 64 monitoring stations is shown in Supplementary Fig. S8b,

Q42. Being pedantic now, but desethylatrazine and deisopropylatrazine are degradation products of triazine herbicides. They are not triazine herbicides.

Answer: Thank you for the reviewer's comments. We have changed the description of 10 triazine herbicides to 10 triazine herbicides and their derivatives as follows:

P₄₂L₉₂₄-P₄₂L₉₂₄

The concentrations of 10 triazine herbicides **and their derivatives**

Q43. What are these 12. Only 10 have been named earlier.

Answer: We are sorry for the unclear description. We detected only 10 triazine herbicides and their derivatives detected in the Bohai Sea and Yellow Sea indeed. But in section 3.1, the data of twelve triazine herbicides and their derivatives are from the 15 bays mentioned above. The details of 12 herbicides and their derivatives in the 15 bays reported between 1990 and 2022 are listed in Table S1.

Q44. Is that its usual name? We always refer to it as the Gulf of Mexico, but it is hard from the map to know the exact location

Answer: Thank you for the reminder. The Gulf of Mexico should be a more acceptable name. We have changed this term throughout the text.

Q44. I'd certainly prefer ug/L units

Answer: Thanks for your suggestion. In order to facilitate the conversion of toxic equivalents between herbicide congeners, the unit of herbicide concentration used in this paper is moles rather

than a mass unit, which may cause some trouble for reviewers. For ease of comparison, we have added the mass concentration values corresponding to each molar concentration.

Q45. What is the residual conc?

But what summary statistic of the concentration data for a site is it?

Answer: The specific value of the residual concentrations of each herbicide in each sampling site can be seen in the Supplementary information (Table S1, Table S2 and Table S5).

Q46.

These box and scatter plots are so small that they are essentially useless. They need to be much bigger.

Answer: We apologize for the difficulty of reading the figures. The figures have been improved as mentioned above.

Reviewers' Comments:

Reviewer #1:

Remarks to the Author:

I have no further comments as the authors have thoroughly addressed all my comments and made changes that improved the quality of the work significantly.

Reviewer #2:

Remarks to the Author:

I reviewed the rebuttal and changes made in the first version of the paper. I would like to thank the authors for careful answers to three reviewers and all their comments. I think the paper is suited for publication as it is.

To be honest, I did not get the same amount of time for this second review but I carefully checked the changes and it seems they improve and clarify a lot the paper.

Reviewer #4:

Remarks to the Author:

This study a priori seems interesting and authors make many improvements of the Ms following referee suggestions. Despite that I still find the study quite difficult to understand. There is no an apparent link between aims, methods and results. Figures are not numbered. Tables and Figs in Suppl information and also in the Main are far away to be self-explanatory. In many occasions methods are incomplete and so it is impossible to interpret correctly the results.

Authors need to mention with more details the particular aims of the study and how this aims will be reached using appropriated methodology.

Other issues

The authors describe correctly how to model mixture joint effects of all measured atrazine residues in the different studied sites but it is still unclear to me how they deviate TEQ for atrazine. Where is the data for Yellow Sea and the Bohai Sea?

Where are the model equations of Table S2, what is the second atrazine model whose EC50 is empty? Authors should include an additional column with the equivalent EC50 of atrazine for each of the 12 triazines or explain that is just calculate by the EC50 ratios between atrazine and the other triazines

I agree with the referee that time points taken from the same replicate have to be consider a "repeated measurement" but I do not know how this data is used in the Ms. Where is this data in results?

Copepod experimentation is incomplete: exposure concentrations and their preparation are missing, number of replicates, measured endpoint. How ingestion rates were measured? Results in Table S4 are confused. It says % relative abundance but values reported seem P values ?

L211 table S3 does not show the data mentioned in this sentence. It shows only diversity indexes.

L229-241. Results reported in this section are not seen in Fig 3 neither in Table S4. It is unclear Again section 3.2.3 is unclear since Fig 4 is unclear since no atrazine concentrations appear but time and Table S4 only show p values

It is unclear in methods how algae growth rates μ and NGR and zooplankton grazing rates were determined in Table S4.

Table S4 legend needs to specify parameters and units.

L289 "it decreased significantly ($P \leq 0.01$)" where is this test? with increasing atrazine dosage. On the

L290 "fourth day, the grazing rates of zooplankton on micro, nano- and pico-phytoplankton" From where come from this data

Fig 6 is difficult to read. Better to limit that figure to correlations without hierarchical analysis and changing OTUS by taxonomic names of interest
I do not understand from where come Fig S1b and so results depicted in L370 are unclear to me

We are very grateful for the reviewer 4's further valuable comments and suggestions to help us improve the manuscript.

As mentioned by the reviewer 4, a remaining problem with the previous manuscript is that the description of the method was not very clear. Therefore, in this round of revisions, we have taken great care to describe the method section in more details and to better connect the aims, methods and results sections in the revised manuscript.

The detailed point-to-point responses to the reviewer 4's comments are as follows:

Reviewer 4

Q1 This study a priori seems interesting and authors make many improvements of the Ms following referee suggestions. Despite that I still find the study quite difficult to understand. There is no an apparent link between aims, methods and results. Figures are not numbered. Tables and Figs in Suppl information and also in the Main are far away to be self-explanatory. In many occasions methods are incomplete and so it is impossible to interpret correctly the results.

Authors need to mention with more details the particular aims of the study and how this aims will be reached using appropriated methodology.

Answer: Thank you very much for helping to point out these very important issues or shortcomings that we had not noticed before. We also apologize for our carelessness and the still unclear description of the method during the last revision. As suggested, in this round of revisions, we have taken great care to describe the method section in more detail and to better connect the aims, methods and results sections in the revised manuscript.

Since there are too many methods used in the study and the amount of information is large, in order to better connect the research purpose, results and methods, we made a brief overview at the beginning of the results section and drew an overall diagram (Figure S1) in supplementary materials to more clearly describe the purpose of this study and the methods used to obtain the results (Lines:161-175). Meanwhile, we have also added some introduction describing the several knowledge gaps faced by this study in the supplementary materials. The details are as follows:

Line 161-175: This study aims to reveal the current global status of marine herbicide pollution and

evaluate its impacts on marine primary productivity and secondary effects on higher trophic levels (Supplementary Fig. S1). By analyzing the spatiotemporal distribution of herbicides at 661 gulf stations worldwide from 1990 to 2022, an overall picture of the current status of herbicide pollution in global coastal waters was obtained; by establishing the toxicity equivalent database of each herbicide and the dose-response relationship between the concentration of atrazine and chlorophyll a in seawater at the phytoplankton community level, the overall inhibition effect of 12 triazine herbicides on phytoplankton primary productivity was quantified; by analyzing the effects of herbicides on phytoplankton community structure, particle size composition, production cycle, and energy transfer process, the potential mechanism of herbicides inhibiting phytoplankton primary productivity was elucidated; Moreover, the effect of herbicides on higher trophic levels and the possibility of predicting marine herbicide pollution through indicators of herbicide use on land were explored. Detailed results for each aspect are as follows.

Line 133-154 in supplementary materials:

Fig. S1

Supplementary Fig. S1. Diagram showing the aim of the study and experimental methods used to reveal the results. Silver, green, blue and orange boxes represent the main purpose of the study, knowledge gaps, problem-solving methods and phased results respectively. The number in each box

corresponds to the Experimental Methods and Experimental Results sections in the manuscript. K1-K4 corresponds to the four knowledge gaps faced in this study.

Knowledge gap 1 (K1): Current herbicide toxicity studies are mostly based on risk assessment of a single herbicide in a small region, lacking a comprehensive understanding of the current status and impact of herbicide pollution in global coastal waters.

Knowledge gap 2 (K2): Due to the large differences in the types and concentrations of herbicides between different sites (Table S1), there is no unified prediction scale and comparison benchmark for the effects of herbicides on primary productivity.

Knowledge gap 3 (K3): There are joint toxicity effects between herbicides with the same target of action. Thus, assessing the ecotoxicity of individual herbicides does not reflect the combined toxic effects of herbicides under *in situ* conditions.

Knowledge gap 4 (K4): It is difficult to truly understand the ecological effects of herbicides *in situ* only from the responses of individual algae or groups, and the impact of herbicides on the structure of the phytoplankton community cannot be based on changes in photosynthetic physiological indicators alone.

Meanwhile, the detailed description of relevant research methods and results has been supplemented (See the specific supplement below); tables and figs in the main manuscript or suppl information have been revised and numbered to make them more self-explanatory. We believe that the above modifications will help reviewers understand and evaluate this study more easily.

Line 161-175

By comparing the changes in the Chl *a* concentration of each particle size (<2 μm , Pico-; 2-20 μm , Nano-; 20-200 μm , Micro-) in the control group and the experimental group, the potential effects of environmental concentrations of triazine herbicides on the particle size structure of phytoplankton was characterized. We revealed that the contribution of nano-phytoplankton decreased significantly after two days of atrazine exposure.

Line 266-270

Based on the observed changes in the micro-zooplankton community structure under atrazine exposure, atrazine at environmental concentrations can cause significant

responses in some copepod larvae and ciliates. To distinguish whether this response is caused directly by the toxicity of atrazine to microplankton or indirectly by the influence of atrazine on the community structure and particle size composition of phytoplankton, thereby affecting microplankton foraging, we selected the larvae of two copepods (*Oithona similis* and *Paracalanus parvus*) and two ciliates (*Euplotes* sp. and *Strombidium* sp.), respectively, to assess their sensitivity to atrazine.

Line 355-361

Changes in micro-zooplankton community structure in response to atrazine exposure revealed that atrazine at environmental concentrations can cause significant responses in some copepod larvae and ciliates (Fig. 5). To distinguish whether these responses are caused by the direct toxicity of atrazine to microplankton, or whether they are caused indirectly by atrazine altering the phytoplankton community structure and particle size composition, thereby affecting microplankton foraging, the larvae of two copepods (*Oithona similis* and *Paracalanus parvus*) and two ciliates (*Euplotes* sp. and *Strombidium* sp.) were employed to evaluate their susceptibility to atrazine.

Line 685-695

Due to the large differences in herbicide types and concentrations between locations (Supplementary Table S1), there is no unified prediction scale and comparison benchmark for the effects of herbicides on primary productivity. To truly reflect the stress effects on phytoplankton under the current herbicide pollution levels, the dose-effect relationship curves and toxicity equivalent database of the 12 triazine herbicides on a representative population (i.e., Supplementary Fig. S2, Source data. Fig. 2) of phytoplankton were firstly established. Then, according to the relative toxicity of the typical atrazine herbicide and its homologs, the concentrations of 12 triazine herbicides were converted into equi-effective concentrations of atrazine, which is uniformly used to measure the degree of herbicide pollution at each site in this experiment.

Line 797-837

2.2.4 Potential effects of environmental concentrations of triazine herbicides on phytoplankton growth rate and energy flow transfer

The changes in community structure and particle size of phytoplankton can modify the overall growth rate and even the energy flow of phytoplankton to higher trophic levels. Based on the classical model (dilution method)^{85, 86, 87} of energy flow transfer between phytoplankton and microzooplankton, the effect of atrazine on the net (NGR) or intrinsic (μ) growth rate of phytoplankton and the grazing rate (g) of zooplankton in natural seawater were measured. Briefly, subsamples of each treatment group (mentioned above 2.2.1) that were collected on the fourth day (initial decline in total chlorophyll a concentration, Fig. 4) were considered as the initial seawater (ISW) samples, and particle-free water (PFW) samples were prepared by filtering the ISW samples through Millipore filters (pore size: 0.2 μm). Then, the ISW was diluted with PFW to five target dilutions of 100%, 80%, 60%, 40%, and 20% in 2.8 L transparent polycarbonate bottles. The incubation volume was 2.5 L with triplicates. All the bottles were incubated in a water incubator for 24 hours, and then the subsample of each dilution gradient was filtered sequentially through 20, 2, and 0.2 μm pore-size polycarbonate filters and stored in the dark at -20°C for further analysis. The phytoplankton retained on the 20, 2, and 0.2 μm pore-size filters were designated as micro-phytoplankton, nano-phytoplankton and pico-phytoplankton, respectively.

The dilution approach rests on two fundamental assumptions: (1) that dilution ratio has no effect on phytoplankton growth rate, and (2) that grazing impact is a linear function of the experimental dilution⁸⁸. Rates of phytoplankton growth and grazing mortality can be inferred from observed changes in population density following incubations of different dilutions of natural seawater. The changes in phytoplankton density over time for the above dilution series can be represented appropriately by the following exponential equations.

$$P_t = P_0 e^{(\mu-g)t} \quad (1)$$

$$P_t = P_0 e^{(\mu-0.8g)t} \quad (2)$$

$$P_t = P_0 e^{(\mu-0.6g)t} \quad (3)$$

$$P_t = P_0 e^{(\mu-0.4g)t} \quad (4)$$

$$P_t = P_0 e^{(\mu-0.2g)t} \quad (5)$$

where P_0 and P_t are the initial and post-24h incubation time (t , d^{-1}) phytoplankton biomass, respectively; μ (d^{-1}) and g (d^{-1}) are values of phytoplankton intrinsic growth rate in the absence of grazing and the phytoplankton mortality rate due to herbivory, respectively. Phytoplankton net

growth rate (d^{-1}) is related to grazing and mortality by Eq. 6 at each dilution level.

$$\left(\frac{1}{t}\right) \ln \frac{P_t}{P_0} = \mu - g \quad (6)$$

The negative slope of this relationship is the grazing mortality rate (g); the Y-axis intercept is the intrinsic growth rate (μ). Phytoplankton net growth rate (NGR) is subtracted by μ and g .

Line862-894

Atrazine toxicity to copepods: A 96-hour toxicity test was conducted on the two species of copepod in 50 mL beakers containing 20 mL of artificial seawater (S= 28-30‰; pH= 8.2±0.1). A serial 2-fold dilution of atrazine concentrations (0.1 nmol L⁻¹ to 32 μmol L⁻¹) was prepared by adding appropriate amount of atrazine stock solution, including 20 copepods in 3 replicates for each concentration. Copepods were transferred into test solutions using disposable Pasteur pipettes in a minimum of sea water to reduce dilution. The two species of copepods were fed daily with *Isochrysis galbana* (1.0×10⁵ ind mL⁻¹). The stock cultures were maintained in a climate-controlled room/chamber at 20±1° C and a 14:10 h light:dark cycle. After 96 hours, the animal's mobility was examined by stereomicroscopic observation⁹⁰. The observed immobility was determined by lack of movement when gently prodded or blown with water. By the end of the experiment, the survival rate of copepods in the control group should be greater than 80%^{92,93}. The data at 96 hours were recorded to calculate the mortality (0-100%) of each copepods across a range of atrazine exposures. The best-fitting model for each concentration response relationship was chosen according to Scholze *et al*^{94 94}.

Atrazine toxicity to ciliates: The toxicity test to ciliates was according to previous studies^{95, 96 95, 96}, but with slight modifications. Briefly, the tests were conducted in a 24-well plate, and 1 mL of boiled rice and wheat grain culture solution was added to each well. No more food was added during the subsequent toxicity tests. A serial 2-fold dilution of atrazine concentrations (0.1 nmol L⁻¹ to 32 μmol L⁻¹) was prepared by adding appropriate amount of atrazine stock solution. Both the atrazine-treated and control groups were established with 3 replicates. Twenty precultured ciliates were added to each microwell with a micropipette and incubated in a constant-temperature incubator at 25°C, oxygen saturation >45%, and photoperiod of 14:10 h light:dark. After 96 hours, the number

of surviving ciliates in each group was counted under a stereomicroscope and recorded. Individuals that were incapacitated or had significant morphological changes were considered dead.

The 96 h LC50 value of atrazine for each zooplankton species was obtained by performing a probit regression analysis and a chi squared goodness of fit test on the experimental data with SPSS17.0⁹⁷. Taking the logarithm of the atrazine concentration as the x value and the mortality of the test species as the y value, a regression curve was drawn and a regression equation was fitted to the data.

Other issues

Q2 The authors describe correctly how to model mixture joint effects of all measured atrazine residues in the different studied sites but it is still unclear to me how they deviate TEQ for atrazine.

Where is the data for Yellow Sea and the Bohai Sea?

Answer: We apologize for the previous unclear description. In this revised manuscript, the details about the toxicity conversion have been supplemented as follows.

Supplementary Methods

Line 319-336

Toxicity equivalent conversion of triazine herbicides: Based on the concentration addition (CA) model (Drescher & Boedeker, 1995) and the single-substance concentration-response curves, the residual distribution data of the 12 triazine herbicides in each bay in the 2.1 dataset were normalized according to equation (1):

$$\sum_{i=1}^n \frac{c_i}{ECx_i} = 1 \quad (1)$$

In this equation, n is the number of herbicide types contained in the water body; ECx_i is the concentration of the single substance i provoking $x\%$ effects; c_i denotes the concentration of component i in the mixture; and c_i/ECx_i is called the toxicity unit (TU) of component i , *i.e.* (Backhaus et al., 2004b). Combined with the best-fitting model mentioned above and the equation (1), the detected concentration of triazine herbicide i (c_i) in a specific sea area/station that was obtained from a marine survey can be converted into the concentration of atrazine with the same TU. Then the total concentration of various herbicides was expressed by the TEQ of atrazine to normalize the various herbicide homologs remaining in the water body, which made it possible to assess accurately

the joint effects of multiple herbicides in natural seawaters (Source data. Fig. 2). The joint toxicity of the twelve triazine was predicted based on the concentration-response curve of atrazine on the phytoplankton chlorophyll a concentration at the community level (See 2.4).

The data for Yellow Sea and the Bohai Sea are part of the collected data, which were attached in the source data (Source data. Fig. 1). We have removed the relevant description to avoid possible ambiguity.

Q3 Where are the model equations of Table S2, what is the second atrazine model whose EC50 is empty? Authors should include an additional column with the equivalent EC50 of atrazine for each of the 12 triazines or explain that is just calculate by the EC50 ratios between atrazine and the other triazines

Answer: We apologize for these errors and unclear descriptions, and thank you very much for the constructive suggestion

Accordingly, the Logistic (L) and Weibull (W) model equations have been supplemented in the revised Table S2.

Supplementary Table S2. The concentration-response functions for each single triazine herbicide.

Sensitive species level	EC50	Model	A1	A2	X0/d	p/k	R ² adj
Atrazine	141.3 ± 12.5	L	1.7142	104.2656	0.1349	0.9173	0.989
DEA	580.3 ± 21.8	L	1.5695	101.0427	0.6049	1.1381	0.995
Propazine	147.1 ± 15.2	L	4.3068	103.3116	0.1584	1.1995	0.989
Simazine	28.7 ± 1.6	L	0.3833	103.3089	0.0309	0.8911	0.996
Terbutryn	8.5 ± 1.1	L	2.1441	101.7772	0.0091	1.1025	0.997
Ametryn	25.1 ± 1.2	L	-0.5755	102.6998	0.0262	0.9461	0.996
Dipropetryn	37.0 ± 3.1	L	-1.0603	101.592	0.0375	0.8534	0.993
Cyanazine	280.3 ± 11.7	L	1.1414	109.9918	0.36235	0.66913	0.995
Cybutryne	4.3 ± 0.3	L	0.678	100.21925	0.00445	1.08514	0.999
DIA	732.1 ± 36.5	W	4.0148	99.4471	0.8182	0.8223	0.989
Prometryn	849.1 ± 21.7	W	1.9596	99.3406	1.0341	0.8109	0.997

Prometon	76.2 ± 3.3	W	5.80E-46	96.4563	0.643	0.2094	0.996
Atrazine (Community level)	55.8±3.25	L	3.1467	101.122	0.103	1.0778	0.995

Note: EC50: 50% effective concentration, the unit is nmol L⁻¹; L: Logistic; W: Weibull; A1, A2, X0, p, d and k represent the parameters of the functions.

Logistic

$$y = \frac{A_1 - A_2}{1 + (x/X_0)^p} + A_2$$

Weibull

$$y = A_2 - (A_2 - A_1)e^{-(kx)^d}$$

The first atrazine equation in Table S2 represents the dose-effect relationship of atrazine on a single species (*Phaeodactylum tricornutum*), and the second atrazine equation represents the effect of atrazine on the phytoplankton at community level *in situ*. The EC50 of the second atrazine equation has been supplemented.

As stated in the article, the toxicity equivalent conversion of triazine herbicides is not calculated simply by the EC50 ratio between atrazine and its homolog. To minimize the error, the corresponding concentrations of each triazine herbicide causing 1 to 99% effects on the chlorophyll *a* fluorescence of cell *P. tricornutum* Pt-1 were calculated in steps of 1%. Detailed steps for all these concentration-response relationships and the calculations were provided in the Supplementary Information (section ‘Individual concentration-response curves and mixture toxicity prediction’).

The EC50 of the first atrazine equation is the equivalent EC50 of atrazine for each of the 12 triazines.

Supplementary Methods

1. Individual concentration-response curves and mixture toxicity prediction

Toxicity equivalent conversion of triazine herbicides: Based on the concentration addition (CA) model (Drescher & Boedeker, 1995) and the single-substance concentration-response curves, the residual distribution data of the 12 triazine herbicides in each bay in the 2.1 dataset were normalized according to equation (1):

$$\sum_{i=1}^n \frac{c_i}{ECx_i} = 1 \quad (1)$$

In this equation, *n* is the number of herbicide types contained in the water body; *ECx_i* is the

concentration of the single substance i provoking $x\%$ effects; c_i denotes the concentration of component i in the mixture; and c_i/ECx_i is called the toxicity unit (TU) of component i , *i.e.* (Backhaus et al., 2004b). Combined with the best-fitting model mentioned above and the equation (1), the detected concentration of triazine herbicide i (c_i) in a specific sea area/station that was obtained from a marine survey can be converted into the concentration of atrazine with the same TU. Then the total concentration of various herbicides was expressed by the TEQ of atrazine to normalize the various herbicide homologs remaining in the water body, which made it possible to assess accurately the joint effects of multiple herbicides in natural seawaters (Source data. Fig. 2). The joint toxicity of the twelve triazine was predicted based on the concentration-response curve of atrazine on the phytoplankton chlorophyll a concentration at the community level (See 2.4).

Q4 I agree with the referee that time points taken from the same replicate have to be consider a “repeated measurement” but I do not know how this data is used in the Ms. Where is this data in results?

Answer: We apologize for our unclear description in the previous manuscript. Indeed, the time point samples were collected separately from three replicates of each treatment, not just one replicate. Specifically, the herbicide microcosm system consisted of twelve 100-liter transparent polycarbonate bottles, with 4 treatments (0, 0.5, 5 and 50 nmol L⁻¹) and three replicates. Sampling and subsequent parametric measurements were performed independently for each bottle at each time point. In order to facilitate readers to understand the design of the experiment, a schematic diagram showing the arrangement of 12 bottles (a) and the sampling time corresponding to each indicator (b) is provided in the revised MS (Supplementary Fig. S8).

Fig. S8

Fig. S8. Schematic diagram of the arrangement of 12 bottles (a) and the sampling time corresponding to each indicator (b)

In addition, the continuous sampling method used in this experiment is mainly due to the following consideration, i.e., ensuring the continuity of various indicators (phytoplankton community structure and other physical/chemical indicators) and comparing the changes of these indicators in time series. Although the succession of community structure under the stress of environmental factors is regular, it is not static. Separately investigating the dynamic changes of the phytoplankton community in three replicate groups of each treatment allows for a more statistically accurate assessment of the response of the phytoplankton community structure to herbicide exposure.

Q5 Copepod experimentation is incomplete: exposure concentrations and their preparation are missing, number of replicates, measured endpoint. How ingestion rates were measured?

Answer: According to the reviewer's comment, detailed description of the copepod experiment has been added as follows.

Line 862-894

Atrazine toxicity to copepods: The toxicity tests on the two species of copepod were carried out in 50 mL beakers with 20 mL of artificial seawater (S= 28-30‰; pH= 8.2±0.1) for 96 hours. A serial 2-fold dilution of atrazine concentrations (0.1 nmol L⁻¹ to 32 µmol L⁻¹) was prepared by adding the appropriate amount of atrazine stock solution, with three replicates of 20 animals per concentration. Copepods were transferred into test solutions using disposable Pasteur pipettes in a minimum of sea water to reduce dilution. The two copepods were fed daily with *Isochrysis galbana* (1.0×10⁵ ind mL⁻¹). The stock cultures were maintained in a climate-controlled room/chamber at 20 ± 1° C and a 14:10 h light:dark cycle. The test end-point was immobility, identified by lack of movement when gently prodded or blown with water. Animals were checked for mobility after 96 h by observation under a stereomicroscope⁹⁰. By the end of the experiment, the survival rate of copepods in the control group should be greater than 80%⁸⁹. The data at the 96 hours were recorded for the description of the mortality rate (0-100%) of each copepods over a range of atrazine exposures. The best-fitting model for each concentration response relationship was chosen as described by Scholze *et al.*

Atrazine toxicity to ciliates: The operational steps of this experiment were slightly modified from those described in previous studies^{91,92}. Briefly, the ciliate toxicity tests were conducted in a 24-well plate, and 1 mL of boiled rice and wheat grain culture solution was added to each well. No food was added during the subsequent toxicity tests. A serial 2-fold dilution of atrazine concentrations (0.1 nmol L⁻¹ to 32 µmol L⁻¹) was prepared by adding the appropriate amount of atrazine stock solution. Both the atrazine-treated and control groups were established with 3 replicates. Twenty precultured ciliates were added to each microwell with a micropipette and incubated in

a constant-temperature incubator at 25°C, oxygen saturation >45%, and photoperiod of 14:10 h light:dark. After 96 hours, the number of surviving ciliates in each group was counted under a stereomicroscope and recorded. Individuals that were incapacitated or had significant morphological changes were considered dead.

The 96 h LC50 value of atrazine for each zooplankton species was obtained by performing a probit regression analysis and a chi squared goodness of fit test on the experimental data with SPSS17.0⁹⁵. Using the logarithm of the atrazine concentration as the x value and the mortality of the test species as the y value, a regression curve was drawn and a regression equation was fitted to the data.

Q6 Results in Table S4 are confused. It says % relative abundance but values reported seem P values?

Answer: We are sorry for this mistake. The values are the mean relative abundance (%) for the respective group of samples indeed. But it's in the form of a ratio rather than a percentage, a hundred times different. In the revised MS, it has been corrected as follows.

Supplementary Table S4. Taxonomic assignment of highly abundant genera of phytoplankton and zooplankton in control and atrazine-treated groups.

OUT ID	Control	0.5 nmol L ⁻¹	5 nmol L ⁻¹	50 nmol L ⁻¹	Phylum	Class	Family	Genus	Similar species	NCBI No. (Similarity)	Cell size (µm)	Reference
OTU1 87	9.8	17.1***	14.3***	60.5***	Dinophyceae	Dinophyceae	Gymnodiniaceae	Gyrodinium	Gyrodinium jinhaense	MH665395.1 (100%)	Micro-(21.8-39.9)	Jiang et al, 2019
OTU3 42	<1.0	3.9*	1.3	<1.0	Dinophyceae	Dinophyceae	Gonyaulacaceae	Adenoides	Adenoides eludens	KY980212.1 (100%)	Micro-(28-35)	Hoppenrath et al, 2003
OTU3 86	<1.0	4.7*	3.0*	2.1*	Dinophyceae	Dinophyceae	Eudubosquellidae	Euduboscquella	Euduboscquella sp. JMC-2019a	MN388923.1 (88.7%)	Micro-(85-194)	Jung et al, 2016
OTU7 55	1.2	1.5	1.6	2.7**	Cryptophyceae	Cryptophyceae	Goniomonadaceae	Goniomonas	Goniomonas avonlea	JQ434475.1 (96.4%)	Nano-(7-11)	Kim et al, 2013
OTU6 55	<1.0	<1.0	<1.0	2.8*	Dinophyceae	Dinophyceae	Lophodiniaceae	Woloszynskia	Woloszynskia halophila	AY628430.1 (87.5%)	Micro-(32-35)	Kremp et al, 2005
OTU2 79	<1.0	<1.0	<1.0	1.6*	Dinophyceae	Dinophyceae	Kareniaceae	Karlodinium	Karlodinium veneficum	JF791048.1 (96.9%)	Nano-(14-18)	Bergholtz et al, 2006

OTU7 81	<1.0	<1.0	1.2	1.5*	Dinophyceae	Dinophyceae	Heterocapsaceae	Heterocapsa	Heterocapsa rotundata	KY980409.1 (97.6%)	Nano-(10-14)	Hansen et al, 1995
OTU2 85	<1.0	16.1***	4.4*	2.9*	Haptista	Prymnesiophyceae	Chrysochromulaceae	Chrysochromulina	Chrysochromulina rotalis	LT560338.1 (100%)	Pico-(2-6)	Eikrem et al, 1999
OTU3 19	<1.0	11.2**	2.8*	1.6*	Haptista	Prymnesiophyceae	Chrysochromulaceae	Chrysochromulina	Chrysochromulina leadbeateri	AM491017.2 (99.4%)	Pico-(2-8)	Estep et al, 1984
OTU1 7	<1.0	1.2*	13.3***	4.5***	Raphidophyceae	Raphidophyceae	Chattonellaceae	Fibrocapsa	Fibrocapsa japonica	JX026949.1 (100%)	Micro-(32-48)	Bowers et al, 2006
OTU5 33	<1.0	6.3*	1.8	<1.0	Dinophyceae	Dinophyceae	Gymnodiniaceae	Ankistrodinium	Ankistrodinium semilunatum	AF274256.1 (91.5%)	Micro-(40-60)	Hoppenrath et al, 2021
OTU3 55	<1.0	2.7	17.0***	8.1**	Dinophyceae	Dinophyceae	Kareniaceae	Gertia	Gertia stigmatica	LC490696.1 (97.4%)	Nano-(7.8-9.5)	Takahashi et al, 2019
OTU4 84	<1.0	1.1*	13.0***	4.7**	Pelagophyceae	Pelagophyceae	Aureococcus	Aureococcus	Aureococcus anophagefferens	KY980308.1 (95.2%)	Pico-(2.0-3.0)	Ma et al, 2020
OTU3 34	0.73	5.2***	2.9***	1.0***	Bacillariophyta	Coscinodiscophyceae	Chaetocerotaceae	Chaetoceros	Chaetoceros tenuissimus	MG972315.1 (100%)	Nano-(>4)	Meunier et al, 1913
OTU5 65	2.7	2.1	8.5*	<1.0	Bacillariophyta	Coscinodiscophyceae	Thalassiosiraceae	Thalassiosira	Thalassiosira nordenskioeldii	MW722947.1 (100%)	Nano-(13.2-44.6)	Durbin et al, 1977
OTU6 34	<1.0	1.3	4.3*	4.7*	Ciliophora	Spirotrichea	Euplotidae	Moneuplotes	Moneuplotes minuta	KX516699.1 (99.1%)	Micro-(40-70)	Song et al, 1997
OTU6 30	<1.0	<1.0	5.4**	46.1***	Ciliophora	Spirotrichea	Holostichidae	Holosticha	Holosticha diademata	KF306396.1 (100%)	Micro-(28-90)	Hu et al, 1999

OTU2 70	1.3	7.5**	12.0***	3.1*	Annelida	Polychaeta	Spionidae	Pseudopolydora	Pseudopolydora paucibranchiata	LC019991.1 (100%)	larvae	Wu et al, 1980
OTU1 3	8.4	2.7***	< 1.0***	< 1.0***	Annelida	Polychaeta	Chrysopetalidae	Paleanotus	Paleanotus bellis	EU555041.1 (99.58)	larvae	Milejkovskij et al, 1961
OTU2 82	<1.0	5.1***	<1.0	<1.0	Arthropoda	Hexanauplia	Acartiidae	Acartia	Acartia pacifica	GU969157.1 (100%)	larvae	Moon et al, 2008
OTU2 64	74.1	72.3	73.0	14.3***	Arthropoda	Hexanauplia	Oithonidae	Oithona	Oithona davisae	KT030258.1 (100%)	larvae	Ferrari et al, 1984
OTU2 31	<1.0	5.0*	<1.0	<1.0	Arthropoda	Hexanauplia	Pyrgomatidae	Pyrgoma	Pyrgoma cancellatum	KM217494.1 (99.8%)	larvae	Ross et al, 2002
OTU5 1	4.7	3.2*	<1.0**	<1.0**	Mollusca	Bivalvia	Ostreidae	Crassostrea	Crassostrea gigas	CP048848.1 (100%)	larvae	Escapa et al, 2004
OTU9 7	2.0	<1.0*	<1.0*	<1.0*	Mollusca	Bivalvia	Veneridae	Ruditapes	Ruditapes philippinarum	MZ227551.1 (100%)	larvae	Delgado et al, 2007
OTU6 11	<1.0	<1.0	<1.0	14.5***	Platyhelminthes	Rhabditophora	Plehnidae	Paraplehnia	Paraplehnia seisuiae	LC508167.1 (99.6%)	larvae	Oya et al, 2019

The mean relative abundance (%) for the respective group of samples is shown.

Darker colours indicate higher abundances of the respective features.

Differences were considered significant at $p < 0.05^*$, $p < 0.01^{**}$, and $p < 0.001^{***}$ according to Tukey's test.

Q7 L211 table S3 does not show the data mentioned in this sentence. It shows only diversity indexes.

Answer: We apologize for this carelessness. The above high-throughput sequencing content corresponds to the data in Figure 3 and Table S4. In the revised MS, the description has been corrected as follows.

with relative proportions (OTU count data) of 77.4% and 17.8%, respectively (Figure 3, Supplementary Table S4).

Q8 L229-241. Results reported in this section are not seen in Fig 3 neither in Table S4. It is unclear

Answer: We are sorry for the ambiguous description. The corresponding description in this part is indeed based on the data in Figure 3 and Appendix Table 4. However, it may be that Table S4 shows the specific particle size values of algal cells, while the main text uses general particle size categories (Pico-, Nano- and Micro-), which may cause barriers to understanding. In the revised manuscript, based on the reviewers' questions, we have added general particle size categories information to Table S4 to make it easier to compare the table data with the text content.

Supplementary Table S4. Taxonomic assignment of highly abundant genera of phytoplankton and zooplankton in control and atrazine-treated groups.

OUT ID	Cont rol	0.5 nmol L ⁻¹	5 nmol L ⁻¹	50 nmol L ⁻¹	Phylum	Class	Family	Genus	Similar species	NCBI No. (Similarity)	Cell size (µm)	Reference
OTU1 87	9.8	17.1***	14.3***	60.5***	Dinophyceae	Dinophyceae	Gymnodiniaceae	Gyrodinium	Gyrodinium jinhaense	MH665395.1 (100%)	Micro-(21.8-39.9)	Jiang et al , 2019
OTU3 42	<1.0	3.9*	1.3	<1.0	Dinophyceae	Dinophyceae	Gonyaulacaceae	Adenoides	Adenoides eludens	KY980212.1 (100%)	Micro- (28-35)	Hoppenrath et al , 2003
OTU3 86	<1.0	4.7*	3.0*	2.1*	Dinophyceae	Dinophyceae	Eudubosquellidae	Euduboscquella	Euduboscquella sp. JMC-2019a	MN388923.1 (88.7%)	Micro-(85-194)	Jung et al , 2016
OTU7 55	1.2	1.5	1.6	2.7**	Cryptophyceae	Cryptophyceae	Goniomonadaceae	Goniomonas	Goniomonas avonlea	JQ434475.1 (96.4%)	Nano-(7-11)	Kim et al , 2013
OTU6 55	<1.0	<1.0	<1.0	2.8*	Dinophyceae	Dinophyceae	Lophodiniaceae	Woloszynskia	Woloszynskia halophila	AY628430.1 (87.5%)	Micro-(32-35)	Kremp et al , 2005

OTU2 79	<1.0	<1.0	<1.0	1.6*	Dinophyceae	Dinophyceae	Kareniaceae	Karlodinium	Karlodinium veneficum	JF791048.1 (96.9%)	Nano-(14-18)	Bergholtz et al, 2006
OTU7 81	<1.0	<1.0	1.2	1.5*	Dinophyceae	Dinophyceae	Heterocapsaceae	Heterocapsa	Heterocapsa rotundata	KY980409.1 (97.6%)	Nano-(10-14)	Hansen et al, 1995
OTU2 85	<1.0	16.1***	4.4*	2.9*	Haptista	Prymnesiophyceae	Chrysochromulaceae	Chrysochromulina	Chrysochromulina rotalis	LT560338.1 (100%)	Pico-(2-6)	Eikrem et al, 1999
OTU3 19	<1.0	11.2**	2.8*	1.6*	Haptista	Prymnesiophyceae	Chrysochromulaceae	Chrysochromulina	Chrysochromulina leadbeateri	AM491017.2 (99.4%)	Pico-(2-8)	Estep et al, 1984
OTU1 7	<1.0	1.2*	13.3***	4.5***	Raphidophyceae	Raphidophyceae	Chattonellaceae	Fibrocapsa	Fibrocapsa japonica	JX026949.1 (100%)	Micro-(32-48)	Bowers et al, 2006
OTU5 33	<1.0	6.3*	1.8	<1.0	Dinophyceae	Dinophyceae	Gymnodiniaceae	Ankistrodinium	Ankistrodinium semilunatum	AF274256.1 (91.5%)	Micro-(40-60)	Hoppenrath et al, 2021
OTU3 55	<1.0	2.7	17.0***	8.1**	Dinophyceae	Dinophyceae	Kareniaceae	Gertia	Gertia stigmatica	LC490696.1 (97.4%)	Nano-(7.8-9.5)	Takahashi et al, 2019
OTU4 84	<1.0	1.1*	13.0***	4.7**	Pelagophyceae	Pelagophyceae	Aureococcus	Aureococcus	Aureococcus anophagefferens	KY980308.1 (95.2%)	Pico-(2.0-3.0)	Ma et al, 2020
OTU3 34	0.73	5.2***	2.9***	1.0***	Bacillariophyta	Coscinodiscophyceae	Chaetocerotaceae	Chaetoceros	Chaetoceros tenuissimus	MG972315.1 (100%)	Nano-(>4)	Meunier et al, 1913
OTU5 65	2.7	2.1	8.5*	<1.0	Bacillariophyta	Coscinodiscophyceae	Thalassiosiraceae	Thalassiosira	Thalassiosira nordenskioeldii	MW722947.1 (100%)	Nano-(13.2-44.6)	Durbin et al, 1977
OTU6 34	<1.0	1.3	4.3*	4.7*	Ciliophora	Spirotrichea	Euplotidae	Moneuplotes	Moneuplotes minuta	KX516699.1 (99.1%)	Micro-(40-70)	Song et al, 1997

OTU6 30	<1.0	<1.0	5.4**	46.1***	Ciliophora	Spirotrichea	Holostichidae	Holosticha	Holosticha diademata	KF306396.1 (100%)	Micro-(28- 90)	Hu et al, 1999
OTU2 70	1.3	7.5**	12.0***	3.1*	Annelida	Polychaeta	Spionidae	Pseudopolydora	Pseudopolydora paucibranchiata	LC019991.1 (100%)	larvae	Wu et al, 1980
OTU1 3	8.4	2.7***	<1.0***	<1.0***	Annelida	Polychaeta	Chrysopetalidae	Paleanotus	Paleanotus bellis	EU555041.1 (99.58)	larvae	Milejkovskij et al, 1961
OTU2 82	<1.0	5.1***	<1.0	<1.0	Arthropoda	Hexanauplia	Acartiidae	Acartia	Acartia pacifica	GU969157.1 (100%)	larvae	Moon et al, 2008
OTU2 64	74.1	72.3	73.0	14.3***	Arthropoda	Hexanauplia	Oithonidae	Oithona	Oithona davisae	KT030258.1 (100%)	larvae	Ferrari et al, 1984
OTU2 31	<1.0	5.0*	<1.0	<1.0	Arthropoda	Hexanauplia	Pyrgomatidae	Pyrgoma	Pyrgoma cancellatum	KM217494.1 (99.8%)	larvae	Ross et al, 2002
OTU5 1	4.7	3.2*	<1.0**	<1.0**	Mollusca	Bivalvia	Ostreidae	Crassostrea	Crassostrea gigas	CP048848.1 (100%)	larvae	Escapa et al, 2004
OTU9 7	2.0	<1.0*	<1.0*	<1.0*	Mollusca	Bivalvia	Veneridae	Ruditapes	Ruditapes philippinarum	MZ227551.1 (100%)	larvae	Delgado et al, 2007
OTU6 11	<1.0	<1.0	<1.0	14.5***	Platyhelminthes	Rhabditophora	Plehnidae	Paraplehnia	Paraplehnia seisuiae	LC508167.1 (99.6%)	larvae	Oya et al, 2019

The mean relative abundance (100%) for the respective group of samples is shown.

Darker colours indicate higher abundances of the respective features.

Differences were considered significant at $p < 0.05^*$, $p < 0.01^{**}$, and $p < 0.001^{***}$ according to Tukey's test.

Again section 3.2.3 is unclear since Fig 4 is unclear since no atrazine concentrations appear but time and Table S4 only show p values

Answer: Sorry for this mistake. The atrazine concentration of each group and the legend information have been supplemented in the Figure 4 in the revised MS as follows:

Fig. 4. Size-fractionated chlorophyll *a* concentrations of the phytoplankton communities in the control and atrazine-treated groups. The number 200, 20, 2 represented size-fraction (Micro-, Nano-, and Pico-) of the phytoplankton. **Source data are provided as a Source Data file.**

It is unclear in methods how algae growth rates μ and NGR and zooplankton grazing rates were determined in Table S4.

Answer: We apologize for the unclear description. In the revised manuscript, the details about the methods have been supplemented as follows.

Line 797-837

2.2.4 Potential effects of environmental concentrations of triazine herbicides on phytoplankton growth rate and energy flow transfer

The changes in community structure and particle size of phytoplankton can modify the overall growth rate and even the energy flow of phytoplankton to higher trophic levels. Based on the

classical model (dilution method)^{85, 86, 87} of energy flow transfer between phytoplankton and micro-zooplankton, the effect of atrazine on the net (NGR) or intrinsic (μ) growth rate of phytoplankton and the grazing rate (g) of zooplankton in natural seawater were measured. Briefly, subsamples of each treatment group (mentioned above 2.2.1) that were collected on the fourth day (initial decline in total chlorophyll a concentration, Fig. 4) were considered as the initial seawater (ISW) samples, and particle-free water (PFW) samples were prepared by filtering the ISW samples through Millipore filters (pore size: 0.2 μm). Then, the ISW was diluted with PFW to five target dilutions of 100%, 80%, 60%, 40%, and 20% in 2.8 L transparent polycarbonate bottles. The incubation volume was 2.5 L with triplicates. All the bottles were incubated in a water incubator for 24 hours, and then the subsample of each dilution gradient was filtered sequentially through 20, 2, and 0.2 μm pore-size polycarbonate filters and stored in the dark at -20°C for further analysis. The phytoplankton retained on the 20, 2, and 0.2 μm pore-size filters were designated as micro-phytoplankton, nano-phytoplankton and pico-phytoplankton, respectively.

The dilution approach rests on two fundamental assumptions: (1) that dilution ratio has no effect on phytoplankton growth rate, and (2) that grazing impact is a linear function of the experimental dilution⁸⁸. Rates of phytoplankton growth and grazing mortality can be inferred from observed changes in population density following incubations of different dilutions of natural seawater. The changes in phytoplankton density over time for the above dilution series can be represented appropriately by the following exponential equations.

$$P_t = P_0 e^{(\mu-g)t} \quad (1)$$

$$P_t = P_0 e^{(\mu-0.8g)t} \quad (2)$$

$$P_t = P_0 e^{(\mu-0.6g)t} \quad (3)$$

$$P_t = P_0 e^{(\mu-0.4g)t} \quad (4)$$

$$P_t = P_0 e^{(\mu-0.2g)t} \quad (5)$$

where P_0 and P_t are the initial and post-24h incubation time (t , d^{-1}) phytoplankton biomass, respectively; μ (d^{-1}) and g (d^{-1}) are values of phytoplankton intrinsic growth rate in the absence of grazing and the phytoplankton mortality rate due to herbivory, respectively. Phytoplankton net growth rate (d^{-1}) is related to grazing and mortality by Eq. 6 at each dilution level.

$$\left(\frac{1}{t}\right) \ln \frac{P_t}{P_0} = \mu - g \quad (6)$$

The negative slope of this relationship is the grazing mortality rate (g); the Y-axis intercept is the intrinsic growth rate (μ). Phytoplankton net growth rate (NGR) is subtracted by μ and g.

Q9 Table S4 legend needs to specify parameters and units.

Answer: We are sorry for causing you the misunderstanding. The values in Table S4 represent the relative abundance values of plankton genera. And the units of the cell size are provided in the header.

Q10 L289 “it decreased significantly (P≤0.01)” where is this test? with increasing atrazine dosage.

Answer: We are sorry for the ambiguous description. The corresponding significance test results are marked in the upper right corner of the corresponding value in Table S5. Differences were considered significant at $p < 0.05^*$, $p < 0.01^{**}$, and $p < 0.001^{***}$ according to Tukey's test.

Supplementary Table S5. Effects of atrazine exposure on phytoplankton growth rate and feeding pressure of zooplankton

Group	Partical Size (μm)	Intrinsic Growth Rate (μ)	Grazing Mortality Rate (g)	Net Growth Rate (NGR)	R ²
CK	20-200	0.65	0.61	0.04	0.8 7
	2-20	0.74	0.71	0.03	0.9 1
	<2	1.14	1.02	0.12	0.8 1
0.5 nmol L ⁻¹	20-200	0.67	0.62	0.05	0.9 3
	2-20	0.61 [*]	0.69	-0.08 ^{***}	0.8 5
	<2	1.01 [*]	0.94	0.07 ^{**}	0.8 8

5 nmol L ⁻¹	20-200	0.53*	0.5**	0.03	0.9 2
	2-20	0.47***	0.59**	-0.12***	0.8 7
	<2	0.97*	0.99	-0.02***	0.8 4
50 nmol L ⁻¹	20-200	0.44**	0.38***	0.06*	0.8 8
	2-20	0.35***	0.52***	-0.17***	0.9 2
	<2	0.82**	0.77**	-0.05***	0.8 5

Note: Differences were considered significant at $p < 0.05^*$, $p < 0.01^{**}$, and $p < 0.001^{***}$ according to Tukey's test.

Q11 On the L290 “fourth day, the grazing rates of zooplankton on micro, nano- and pico-phytoplankton” From where come from this data

Answer: We are sorry to causing this confusion to the reviewer. The relevant data are provided in Supplementary information Table 5 of the supplementary material, and detailed experimental methods and procedures are provided in 2.2.4. We have annotated the character abbreviations in Table 5 to enhance its self-explanation.

On the fourth day, the grazing rates of zooplankton on micro, nano- and pico-phytoplankton under intermediate and high doses of atrazine decreased by 18.1% and 37.7%, 16.9 and 11.9, and 2.9% and 24.5% (Supplementary Table S5)

Supplementary Table S5. Effects of atrazine exposure on phytoplankton growth rate and feeding pressure of zooplankton

Group	Partical Size (μm)	Intrinsic Growth Rate (μ)	Grazing Mortality Rate (g)	Net Growth Rate (NGR)	R ²
CK	20-200	0.65	0.61	0.04	0.8

					7
	2-20	0.74	0.71	0.03	0.9
					1
	<2	1.14	1.02	0.12	0.8
					1
0.5 nmol L ⁻¹	20-200	0.67	0.62	0.05	0.9
					3
	2-20	0.61*	0.69	-0.08***	0.8
					5
	<2	1.01*	0.94	0.07**	0.8
					8
5 nmol L ⁻¹	20-200	0.53*	0.5**	0.03	0.9
					2
	2-20	0.47***	0.59**	-0.12***	0.8
					7
	<2	0.97*	0.99	-0.02***	0.8
					4
50 nmol L ⁻¹	20-200	0.44**	0.38***	0.06*	0.8
					8
	2-20	0.35***	0.52***	-0.17***	0.9
					2
	<2	0.82**	0.77**	-0.05***	0.8
					5

Note: Differences were considered significant at $p < 0.05^*$, $p < 0.01^{}$, and $p < 0.001^{***}$ according to Tukey's test.**

Line 797-837

2.2.4 Potential effects of environmental concentrations of triazine herbicides on phytoplankton growth rate and energy flow transfer

The changes in community structure and particle size of phytoplankton can modify the overall growth rate and even the energy flow of phytoplankton to higher trophic levels. Based on the classical model (dilution method)^{85, 86, 87} of energy flow transfer between phytoplankton and micro-

zooplankton, the effect of atrazine on the net (NGR) or intrinsic (μ) growth rate of phytoplankton and the grazing rate (g) of zooplankton in natural seawater were measured. Briefly, subsamples of each treatment group (mentioned above 2.2.1) that were collected on the fourth day (initial decline in total chlorophyll a concentration, Fig. 4) were considered as the initial seawater (ISW) samples, and particle-free water (PFW) samples were prepared by filtering the ISW samples through Millipore filters (pore size: 0.2 μm). Then, the ISW was diluted with PFW to five target dilutions of 100%, 80%, 60%, 40%, and 20% in 2.8 L transparent polycarbonate bottles. The incubation volume was 2.5 L with triplicates. All the bottles were incubated in a water incubator for 24 hours, and then the subsample of each dilution gradient was filtered sequentially through 20, 2, and 0.2 μm pore-size polycarbonate filters and stored in the dark at -20°C for further analysis. The phytoplankton retained on the 20, 2, and 0.2 μm pore-size filters were designated as micro-phytoplankton, nano-phytoplankton and pico-phytoplankton, respectively.

The dilution approach rests on two fundamental assumptions: (1) that dilution ratio has no effect on phytoplankton growth rate, and (2) that grazing impact is a linear function of the experimental dilution⁸⁸. Rates of phytoplankton growth and grazing mortality can be inferred from observed changes in population density following incubations of different dilutions of natural seawater. The changes in phytoplankton density over time for the above dilution series can be represented appropriately by the following exponential equations.

$$P_t = P_0 e^{(\mu - g)t} \quad (1)$$

$$P_t = P_0 e^{(\mu - 0.8g)t} \quad (2)$$

$$P_t = P_0 e^{(\mu - 0.6g)t} \quad (3)$$

$$P_t = P_0 e^{(\mu - 0.4g)t} \quad (4)$$

$$P_t = P_0 e^{(\mu - 0.2g)t} \quad (5)$$

where P_0 and P_t are the initial and post-24h incubation time (t , d^{-1}) phytoplankton biomass, respectively; μ (d^{-1}) and g (d^{-1}) are values of phytoplankton intrinsic growth rate in the absence of grazing and the phytoplankton mortality rate due to herbivory, respectively. Phytoplankton net growth rate (d^{-1}) is related to grazing and mortality by Eq. 6 at each dilution level.

$$\left(\frac{1}{t}\right) \ln \frac{P_t}{P_0} = \mu - g \quad (6)$$

The negative slope of this relationship is the grazing mortality rate (g); the Y-axis intercept is the intrinsic growth rate (μ). Phytoplankton net growth rate (NGR) is subtracted by μ and g .

Q12 Fig 6 is difficult to read. Better to limit that figure to correlations without hierarchical analysis and changing OTUS by taxonomic names of interest.

Answer: Many thanks for your suggestion. The figure 6 has been improved in the revised manuscript accordingly.

Fig. 6. Correlation patterns between phytoplankton and their associated microzooplankton at the genus level. Correlations were calculated for phytoplankton (*Bacillariophyta*, *Dinophyceae*, and *Cryptophyceae*) and zooplankton groups with an abundance $\geq 1\%$ in at least one sample. The groups marked unclassified in the high-throughput sequencing results were manually blasted against the NCBI nucleotide collection and EzBioCloud Database. Sequences with the same taxonomic assignment at the genus level were combined. The relative abundances of phytoplankton and microzooplankton OTUs in the control and atrazine-treated groups (0.5, 5, and 50 nmol L^{-1}) were

employed to construct Bray–Curtis similarity matrices. The analyses were based on Spearman’s rank correlation coefficients.

Q13 I do not understand from where come Fig S1b and so results depicted in L370 are unclear to me

Answer: We are sorry for causing this confusion. The Fig S1b has been provided in the Supplementary information (P_{17}) as follows.

Fig. S2

Supplementary Fig. S2. Single-substance concentration–response curves of the 12 tested triazine herbicides (a) and the concentration–response curve of atrazine on the phytoplankton chlorophyll a concentration at the community level (b). From left to right, the curves intersecting the dashed lines represent the twelve triazine herbicides in the following order: Simazine, Dipropetryn, Cyanazine, Terbutryn, Cybutryne, Prometon, Prometryn, Desethylatrazine, Desisopropylatrazine, Propazine, Ametryn, and Atrazine; 95% LPL: 95% lower prediction limit; 95% UPL: 95% upper prediction limit.

To make it easier for reviewers to compare the contents, all the Tables and Figs in the main MS and Suppl information are numbered.

Reviewers' Comments:

Reviewer #4:

Remarks to the Author:

The authors implemented almost all suggested changes. To me there is only a minor issue remaining. Figure 4 is still confusing. I suggest to add a text in the legend (i.e. within each time period the four bars depict respectively the values for control, 0.5, 5 and 50 nmol/L of atrazine). There are four columns of legends . Left only one to identify the size fractions .

The detailed point-to-point responses to the reviewer 4's comments are as follows:

Reviewer 4

The authors implemented almost all suggested changes. To me there is only a minor issue remaining. Figure 4 is still confusing. I suggest to add a text in the legend (i.e. within each time period the four bars depict respectively the values for control, 0.5, 5 and 50 nmol/L of atrazine). There are four columns of legends. Left only one to identify the size fractions.

Answer: We are very grateful to the reviewer for his further valuable comments and suggestions to help us improve the manuscript.

Figure 4 and the legend are revised as follows.

Fig. 4. Size-fractionated chlorophyll *a* concentrations of the phytoplankton communities in the control and atrazine-treated groups. Within each time period the four bars depict respectively the values for control, 0.5, 5 and 50 nmol/L of atrazine. $n=3$ samples per group. Error bars represent mean \pm standard deviation (SD). Statistical significance was determined and comparisons used t-tests (two-sided) paired with FDR adjusted for multiple comparisons using the Benjamini and Hochberg method.

The exact p values were provided in the Source data. The asterisk represents a significant difference ($p < 0.05$) in the chlorophyll a concentration of phytoplankton between the treatment group and the control group. Source data are provided as a Source Data file.